# Adaptive Algorithms with Sharp Convergence Rates for Stochastic Hierarchical Optimization

**Xiaochuan Gong**
George Mason University
xgong2@gmu.edu

**Jie Hao**
George Mason University
jhao6@gmu.edu

**Mingrui Liu**
George Mason University
mingruil@gmu.edu

## Abstract

Hierarchical optimization refers to problems with interdependent decision variables and objectives, such as minimax and bilevel formulations. While various algorithms have been proposed, existing methods and analyses lack adaptivity in stochastic optimization settings: they cannot achieve optimal convergence rates across a wide spectrum of gradient noise levels without prior knowledge of the noise magnitude. In this paper, we propose novel adaptive algorithms for two important classes of stochastic hierarchical optimization problems: nonconvex-strongly-concave minimax optimization and nonconvex-strongly-convex bilevel optimization. Our algorithms achieve sharp convergence rates of $\widetilde{O}(1/\sqrt{T} + \sqrt{\bar{\sigma}}/T^{1/4})$ in $T$ iterations for the gradient norm, where $\bar{\sigma}$ is an upper bound on the stochastic gradient noise. Notably, these rates are obtained *without prior knowledge* of the noise level, thereby enabling automatic adaptivity in both low and high-noise regimes. To our knowledge, this work provides the *first* adaptive and sharp convergence guarantees for stochastic hierarchical optimization. Our algorithm design combines the momentum normalization technique with novel adaptive parameter choices. Extensive experiments on synthetic and deep learning tasks demonstrate the effectiveness of our proposed algorithms.

## 1 Introduction

Hierarchical optimization refers to a class of optimization problems characterized by nested structures in their objectives or constraints, such as minimax optimization [61, 64, 50] and bilevel optimization [5, 13]. These problems have wide applications in machine learning. For example, minimax optimization is the foundation for adversarial learning [27] and AUC maximization [75, 53], while bilevel optimization is central to meta-learning [19] and hyperparameter optimization [20, 18]. In this paper, we are interested in solving two classes of stochastic hierarchical optimization problems. The first class is the nonconvex-strongly-concave minimax problem in (1):

$$\min_{x \in \mathbb{R}^{d_x}} \max_{y \in \mathbb{R}^{d_y}} f(x,y) := \mathbb{E}_{\xi \sim \mathcal{D}}\left[F(x,y;\xi)\right], \tag{1}$$

where $\mathcal{D}$ is an unknown distribution where one can sample from, and $f(x,y)$ is nonconvex in $x$ and strongly concave in $y$. The second class is the nonconvex-strongly-convex bilevel problem in (2):

$$\min_{x \in \mathbb{R}^{d_x}} \Phi(x) := f(x, y^*(x)), \quad \text{s.t.,} \quad y^*(x) = \arg\min_{y \in \mathbb{R}^{d_y}} g(x,y), \tag{2}$$

where $\mathcal{D}_x$ and $\mathcal{D}_y$ are unknown distributions where one can sample from, $f(x,y) := \mathbb{E}_{\xi \sim \mathcal{D}_x}[F(x,y;\xi)]$ is nonconvex in $x$ and $g(x,y) := \mathbb{E}_{\xi \sim \mathcal{D}_y}[G(x,y;\zeta)]$ is strongly convex in $y$. We call $x$ the upper-level variable and $y$ the lower-level variable. Note that the bilevel problem in (2) degenerates to the minimax problem in (1) when $g = -f$ and then $\Phi(x) = \max_{y \in \mathbb{R}^{d_y}} f(x,y)$.

39th Conference on Neural Information Processing Systems (NeurIPS 2025).

There are various algorithms designed for the minimax problem (1) and the bilevel problem (2) in the stochastic setting [64, 50, 47, 22, 34, 40, 8, 33, 45, 29, 28]. However, existing algorithms and analyses lack adaptivity to various levels of stochastic gradient noise: their convergence rates remain suboptimal in various noise regimes unless the noise level is known a priori (see Appendix H for discussion and details). In contrast, such an adaptivity guarantee is achieved in single-level stochastic optimization [48, 67, 42, 17, 56, 2] by AdaGrad-type algorithms [60, 14]. This naturally motivates us to study the following question:

**How can we design adaptive gradient algorithms for stochastic hierarchical optimization problems** (1) **and** (2) **that achieve convergence rates automatically adapting to the level of stochastic gradient noise, without requiring prior knowledge of this noise?**

Designing such algorithms in stochastic hierarchical optimization presents significant challenges. In particular, applying AdaGrad-type algorithms (e.g., AdaGrad-Norm [67]) simultaneously to the upper- and lower-level variables will introduce complicated randomness dependency issues due to AdaGrad stepsizes. These dependencies are difficult to handle analytically without imposing strong assumptions such as bounded stochastic gradients or bounded function values [47]. However, such assumptions undermine the algorithm's ability to adapt effectively in various noise regimes.

In this paper, we address these challenges by developing novel adaptive algorithms for solving (1) and (2), respectively. Unlike standard AdaGrad-type algorithms [67], the key innovation of our approach lies in combining the momentum normalization technique [11] with novel adaptive parameter choices. A distinctive feature of our method is the dynamic adjustment of the momentum parameter based on online estimates of the stochastic gradient variance. This adaptive momentum directly informs our stepsize scheme, enabling improved convergence across both high- and low-noise regimes without requiring prior knowledge of the noise level. The primary challenge in analyzing the convergence of our proposed algorithms is simultaneously controlling the upper-level and lower-level errors under time-varying parameters, including adaptive momentum and stepsizes, while maintaining adaptivity in the presence of unknown stochastic noise. Our main contributions are summarized as follows.

- We propose two new adaptive algorithms, namely Ada-Minimax and Ada-BiO, for solving the nonconvex-strong-concave minimax optimization problem (1) and the nonconvex-strongly-convex bilevel optimization problem (2) respectively. Both algorithms leverage the momentum normalization technique and adaptively set the momentum parameter, along with carefully designed adaptive stepsizes for both upper- and lower-level variables. To our knowledge, adaptive algorithms of this type that distincts from standard AdaGrad approaches are novel within both stochastic single-level and hierarchical optimization problems.

- We obtain a high probability convergence rate of $\widetilde{O}(1/\sqrt{T} + \sqrt{\bar{\sigma}}/T^{1/4})$ in $T$ iterations for the gradient norm (here $\widetilde{O}(\cdot)$ compresses poly-logarithmic factors of $T$ and the failure probability $\delta \in (0,1)$), where $\bar{\sigma}$ denotes an upper-bound on the stochastic gradient noise. Notably, our algorithms automatically adapt to both high- and low-noise regimes without requiring prior knowledge of the noise levels.

- We empirically validate our theoretical results through a synthetic experiment and various deep learning tasks, including deep AUC maximization and hyperparameter optimization. Our results demonstrate that our proposed algorithms consistently outperform existing adaptive gradient methods as well as well-tuned non-adaptive baselines.

## 2   Related Work

**Minimax Optimization.** Early works on minimax optimization focused on convex-concave settings and developed first-order algorithms with convergence guarantees [61, 63, 41, 62]. The study of first-order algorithms for nonconvex-concave minimax optimization was pioneered by [64]. Subsequent works improved convergence rates under various assumptions [53, 68, 59], proposed single-loop algorithms [50, 29, 71], and relaxed the concavity requirement on the maximization variable [69, 51, 52, 4]. Some recent efforts have incorporated adaptive gradient methods into minimax optimization [51, 47, 38, 70]. However, none of these approaches provide convergence guarantees that adapt across different levels of stochastic gradient noise in nonconvex-strongly-concave settings.

**Bilevel Optimization.** Bilevel optimization [5, 13] is a type of hierarchical optimization problem where one optimization task (i.e., upper-level problem) is constrained by another optimization task

(i.e., lower-level problem). The first nonasymptotic convergence guarantees for bilevel optimization problems were established by [22], followed by a growing body of work that established improved complexity bounds under various assumptions [34, 40, 8, 43, 9, 12, 28, 72, 33, 25, 24, 12, 36, 45, 65, 57, 58]. More recently, a few studies have explored bilevel optimization algorithms with adaptive step sizes [15, 1, 73, 37]. However, these methods are either restricted to the deterministic setting [1, 73] or fail to adapt to a broad range of stochastic gradient noise levels [15, 37, 26] in nonconvex-strongly-convex bilevel optimization problems.

**Adaptive Gradient Algorithms.** Adaptive gradient algorithms [60, 14, 66, 44] refer to a class of first-order algorithms where the stepsizes are computed based on the historical stochastic gradients, and they are empirically effective for training deep neural networks. The theoretical guarantees of adaptive gradient algorithms for single-level optimization problems are extensively studied and well-understood in the literature [48, 67, 42, 17, 56, 2, 46, 16]. Extensions of adaptive methods to minimax [51, 47, 38, 70] and bilevel optimization [37, 26, 1, 73] have also been proposed. However, none of these works establish theoretical guarantees for adaptivity to unknown stochastic gradient noise levels in nonconvex-strongly-concave minimax or nonconvex-strongly-convex bilevel optimization problems, as achieved by our proposed algorithms in this work.

## 3 Preliminaries

**Notations.** Denote $\|\cdot\|$ as the Euclidean norm. We use the standard $O(\cdot), \Theta(\cdot), \Omega(\cdot)$ notations, with $\widetilde{O}(\cdot), \widetilde{\Theta}(\cdot), \widetilde{\Omega}(\cdot)$ hiding logarithmic factors. Throughout, with slight abuse of notation, we use $\mathcal{F}_t$ to denote the filtration (i.e., $\sigma$-algebra) generated by stochastic queries up to iteration $t$, and $\mathbb{E}_t[\cdot] = \mathbb{E}[\cdot \mid \mathcal{F}_t]$ to denote the conditional expectation with respect to $\mathcal{F}_t$, for all algorithms. A function $h$ is said to be $L$-smooth if $\|\nabla h(x) - \nabla h(y)\| \leq L\|x - y\|$ for all $x, y \in \mathbb{R}^d$. We additionally assume that all objective functions are bounded from below, i.e., $f^* := \inf_x f(x) > -\infty$ (Section 5.1) and $\Phi^* := \inf_x \Phi(x) > -\infty$ (Sections 3.1 and 3.2).

**Settings.** Let $y^*(x) = \arg\max_{y \in \mathbb{R}^{d_y}} f(x, y)$ for (1) and $y^*(x) = \arg\min_{y \in \mathbb{R}^{d_y}} g(x, y)$ for (2). Define the objective function $\Phi(x) = f(x, y^*(x))$ for both minimax and bilevel optimization. Recall from Section 1 that the bilevel problem (2) reduces to the minimax problem (1) when $g = -f$. The goal of this paper is to minimize $\Phi$.

### 3.1 Assumptions for Nonconvex-Strongly-Concave Minimax Optimization

**Assumption 3.1.** The objective function $f$ is $L$-smooth in $(x, y)$ and $f(x, \cdot)$ is $\mu$-strongly concave.

**Assumption 3.2.** (i) The gradient oracle is unbiased, i.e., $\mathbb{E}[\nabla F(x, y; \xi) \mid x, y] = \nabla f(x, y)$. (ii) With probability one, the following holds: $\underline{\sigma}_x \leq \|\nabla_x F(x, y; \xi) - \nabla_x f(x, y)\| \leq \bar{\sigma}_x$ with $\underline{\sigma}_x \geq 0$ and $\|\nabla_y F(x, y; \xi) - \nabla_y f(x, y)\| \leq \sigma_y$.

**Remark:** Assumptions 3.1 and 3.2(i) are standard in the minimax optimization literature [50, 74, 29]. The main extra assumption we make is Assumption 3.2(ii): the stochastic gradient noise is lower bounded and upper-bounded (with probability one), which may appear somewhat unusual. However, this assumption holds naturally in the additive noise setting used in certain nonconvex optimization scenarios, such as escaping saddle points with isotropic noise [21], where the stochastic gradient noise is sampled uniformly from the unit sphere and therefore has a nonzero magnitude with probability one. We also empirically validate this assumption, as shown in Appendix L. In the noiseless case, we have $\bar{\sigma}_x = \underline{\sigma}_x = 0$, and $\sigma_y = 0$.

### 3.2 Assumptions for Nonconvex-Strongly-Convex Bilevel Optimization

**Assumption 3.3.** The objective functions $f$ and $g$ satisfy: (i) $f$ is $L$-smooth in $(x, y)$; for every $x, \xi$, $\|\nabla_y f(x, y)\| \leq l_{f,0}$ and $\|\nabla_y F(x, y; \xi)\| \leq l_{f,0}$. (ii) For every $x$, $g(x, \cdot)$ is $\mu_g$-strongly convex for $\mu_g > 0$ and $g$ is $l_{g,1}$-smooth in $(x, y)$. (iii) $g$ is twice continuously differentiable, and $\nabla^2_{xy} g, \nabla^2_{yy} g$ are $l_{g,2}$-Lipschitz in $(x, y)$.

**Remark:** Assumption 3.3 is standard in the bilevel optimization literature [40, 45, 22, 33, 7]. Notably, the condition $\|\nabla_y F(x, y; \xi)\| \leq l_{f,0}$ is essential for deriving $\|\bar{\nabla} f(x, y; \bar{\xi}) - \mathbb{E}[\bar{\nabla} f(x, y; \bar{\xi})]\| \leq \bar{\sigma}_\phi$ in Lemma E.3 (see Appendix E.1 for the definition of $\bar{\nabla} f(x, y; \bar{\xi})$), where $\bar{\sigma}_\phi$ plays a similar role to

$\bar{\sigma}_x$ in Assumption 3.2 for minimax optimization. Under these assumptions, the objective function $\Phi$ is $L_F$-smooth; please refer to Lemma E.1 in Appendix E for the definition of $L_F$ and further details.

**Assumption 3.4.** All stochastic estimators are unbiased, and almost surely satisfy (i) $\|\nabla_x F(x, y; \xi) - \nabla_x f(x, y)\| \leq \sigma_f$; (ii) $\|\nabla_y F(x, y; \xi) - \nabla_y f(x, y)\| \leq \sigma_f$; (iii) $\|\nabla_y G(x, y; \zeta) - \nabla_y g(x, y)\| \leq \sigma_{g,1}$; (iv) $\|\nabla_{xy}^2 G(x, y; \zeta) - \nabla_{xy}^2 g(x, y)\| \leq \sigma_{g,2}$; (v) $\|\nabla_{yy}^2 G(x, y; \zeta) - \nabla_{yy}^2 g(x, y)\| \leq \sigma_{g,2}$; (vi) $\|\bar{\nabla} f(x, y; \bar{\xi}) - \mathbb{E}[\bar{\nabla} f(x, y; \bar{\xi})]\| \geq \underline{\sigma}_\phi$, where $\bar{\nabla} f(x, y; \bar{\xi})$ is defined in Equation (39).

**Remark:** Assumption 3.4(i)-(v) assumes the noise in the stochastic gradient and Hessian/Jacobian is bounded with probability one. This is a commonly used assumption to establish high probability guarantees or handle generalized-smooth objective functions in the single-level optimization literature [46, 2, 39, 78, 77], as well as for stochastic bilevel optimization under the unbounded smoothness setting [33, 25]. Assumption 3.4(vi) is a stochastic gradient noise lower bound for the bilevel optimization problem, sharing a similar spirit to Assumption 3.2(ii). Note that Assumption 3.2 is empirically verified in Appendix L. Under Assumption 3.4, we also have $\|\bar{\nabla} f(x, y; \bar{\xi}) - \mathbb{E}[\bar{\nabla} f(x, y; \bar{\xi})]\| \leq \bar{\sigma}_\phi$, where the definition of $\bar{\sigma}_\phi$ can be found in Equation (44). See the detailed proof in Lemma E.3.

**Additional Notations.** In the subsequent analysis, we denote $\kappa_\sigma := \bar{\sigma}/\underline{\sigma}$ in Section 5.1 (single-level optimization), $\kappa_\sigma := \bar{\sigma}_x/\underline{\sigma}_x$ in Section 4.2 (minimax optimization), and $\kappa_\sigma := \bar{\sigma}_\phi/\underline{\sigma}_\phi$ in Section 4.3 (bilevel optimization). We also adopt the convention $0/0 := 1$.

# 4 Algorithms and Main Results

## 4.1 Main Challenges and Algorithm Design

**Main Challenges.** While numerous adaptive gradient algorithms with adaptivity to stochastic gradient noise are developed in single-level optimization [48, 67, 42, 17, 56, 2], designing algorithms with such an adaptive guarantee in hierarchical optimization is nontrivial. The main challenges lies in the following two aspects. First, designing such an algorithm in hierarchical optimization requires a careful balance between the upper- and lower-level update [29, 34, 10], which is difficult to achieve without the knowledge of the noise magnitude of stochastic gradient. Second, applying AdaGrad-type algorithms (e.g., AdaGrad-Norm [67]) simultaneously to the upper- and lower-level variables will introduce complicated randomness dependency issues due to AdaGrad stepsizes [2], which are difficult to handle unless strong assumptions (e.g., bounded stochastic gradient, bounded function value) are imposed as in [47].

**Algorithm Design.** To address these main challenges, our proposed algorithms leverage the normalized stochastic gradient descent (SGD) with momentum for the upper-level variable [11] with a noise-aware adaptive momentum parameter and carefully crafted adaptive stepsize schemes for both levels. The momentum parameter automatically estimates the level of noise in the stochastic gradients on the fly, and this estimate is used to construct the stepsizes to maintain a balanced progress across both levels. These adaptive mechanisms, together with the momentum normalization technique, not only improve optimization stability but also make the theoretical convergence analysis more tractable. In particular, our proposed adaptive algorithms, namely Ada-Minimax and Ada-BiO, are designed for the minimax problem (1) and the bilevel problem (2) respectively. Both algorithms achieve sharp and adaptive convergence rates of $\widetilde{O}(1/\sqrt{T} + \sqrt{\bar{\sigma}}/T^{1/4})$ for the gradient norm, where $\bar{\sigma}$ denotes an upper bound on the stochastic gradient noise. We describe our methods in Algorithms 1 and 2 with novel parameter choices in Equations (3) and (4). Their respective convergence guarantees are stated in Theorems 4.1 and 4.2.

**Adaptive Parameter Choices.** For simplicity, let $\alpha_t = 1 - \beta_t$. In particular, for both Algorithms 1 and 2, we set $\alpha_t, \alpha_t', \eta_{x,t}, \eta_{y,t}$ as follows:

$$\alpha_t = \frac{\alpha}{\sqrt{\alpha^2 + \sum_{k=1}^{t} \|g_{x,k} - \tilde{g}_{x,k}\|^2}}, \quad \alpha_t' = \frac{\alpha}{\sqrt{\alpha^2 + \sum_{k=1}^{t} \|g_{x,k} - \tilde{g}_{x,k}\|^2 + \|g_{y,k}\|^2}}, \quad (3)$$

$$\eta_{x,t} = \frac{\eta \sqrt{\alpha_t'}}{\sqrt{t}}, \quad \text{and} \quad \eta_{y,t} = \frac{\eta}{\sqrt{\gamma^2 + \sum_{k=1}^{t} \|g_{y,k}\|^2}}. \quad (4)$$

| **Algorithm 1** Adaptive Algorithm for Minimax Optimization (Ada-Minimax) | **Algorithm 2** Adaptive Algorithm for Bilevel Optimization (Ada-BiO) |
|---|---|
| 1: **Input:** $x_1, y_1, m_1 = \nabla_x F(x_1, y_1; \xi_1)$ | 1: **Input:** $x_1, y_1, m_1 = \bar{\nabla} f(x_1, y_1; \bar{\xi}_1)$ |
| 2: **for** $t = 1, \ldots, T$ **do** | 2: **for** $t = 1, \ldots, T$ **do** |
| 3: $\quad m_t = \beta_t m_{t-1} + (1 - \beta_t) g_{x,t}$ | 3: $\quad m_t = \beta_t m_{t-1} + (1 - \beta_t) g_{x,t}$ |
| 4: $\quad x_{t+1} = x_t - \eta_{x,t} \frac{m_t}{\|m_t\|}$ | 4: $\quad x_{t+1} = x_t - \eta_{x,t} \frac{m_t}{\|m_t\|}$ |
| 5: $\quad y_{t+1} = y_t + \eta_{y,t} g_{y,t}$ | 5: $\quad y_{t+1} = y_t - \eta_{y,t} g_{y,t}$ |
| 6: **end for** | 6: **end for** |

In the above formulas, the terms $g_{x,t}, \tilde{g}_{x,t}$, and $g_{y,t}$ carry different meanings; see the subsequent sections (Sections 4.2 and 4.3) for their precise definitions. For simplicity, we set $\eta_x = \eta_y = \eta$ in analysis of Algorithms 1 and 2 (see Theorems 4.1 and 4.2). It is worth noting that this condition is not necessary for establishing convergence, as it only affects the universal constants in the rate.

## 4.2 Adaptive Algorithm for Minimax Optimization

Our proposed algorithm Ada-Minimax is presented in Algorithm 1. The algorithm updates the upper-level variable using normalized SGD with momentum [11] with adaptive and parameter-free choices for the momentum parameter and learning rates. The lower-level variable is updated by AdaGrad-Norm. In Equations (3) and (4), $g_{x,t} = \nabla_x F(x_t, y_t; \xi_t)$, $\tilde{g}_{x,t} = \nabla_x F(x_t, y_t; \xi'_t)$, and $g_{y,t} = \nabla_y F(x_t, y_t; \xi_t)$, with $\xi_t, \xi'_t$ being independent samples.

Intuitively, the term $\sum_{k=1}^{t} \|g_{x,k} - \tilde{g}_{x,k}\|^2$ in the denominator of $\alpha_t$ is designed to approximate the variance term $\sigma^2 T$ as in [11], and this choice is partly inspired by AdaGrad-Norm [14, 2]. Additionally, using $\alpha'_t$ instead of $\alpha_t$ in the design of $\eta_{x,t}$ effectively controls the ratio $\eta_{x,t}/\eta_{y,t}$ and facilitates establishing Lemma 5.7. It is worth noting that Assumption 3.2 plays a crucial role in deriving tight, high-probability upper and lower bounds for both $\sum_{k=1}^{t} \|g_{x,k} - \tilde{g}_{x,k}\|^2$ and $\alpha_t$, see Lemma 5.5 for details.

**Theorem 4.1.** *Under Assumptions 3.1 and 3.2 and the parameter choices in Equations* (3) *and* (4), *let $\bar{\sigma}_x = \sigma_y$, then for any $\delta \in (0, 1/7)$, it holds with probability at least $1 - 7\delta$ that*

$$\frac{1}{T} \sum_{t=1}^{T} \|\nabla \Phi(x_t)\| \leq \frac{C_m}{\eta \sqrt{\alpha}} \left( \frac{1}{\sqrt{T}} \left( \alpha^2 + \frac{L^2}{\mu^2 \eta^2} \left( 4D^2 L^2 + 2D\gamma \right) \right)^{1/4} \right.$$

$$\left. + \frac{1}{T^{3/8}} \left( \frac{2\sqrt{2} L^2 D \bar{\sigma}_x}{\mu^2 \eta^2} \right)^{1/4} + \frac{1}{T^{1/4}} \left( 5\bar{\sigma}_x^2 \right)^{1/4} \right),$$

*where $C_m = \widetilde{O}(\kappa_\sigma^4)$ and $D$ are defined in Equations* (24) *and* (38), *respectively.*

**Remark:** Theorem 4.1 demonstrates that Ada-Minimax achieves a rate of $\widetilde{O}(\sqrt{\bar{\sigma}_x}/T^{1/4})$ in the stochastic setting ($\sigma_x > 0$) and $\widetilde{O}(1/\sqrt{T})$ rate in the deterministic setting ($\bar{\sigma}_x = 0$). More importantly, our bound achieves the same bound of normalized SGD with momentum under known stochastic gradient variance [11]: it automatically interpolates between sharp rates in both high-noise and low-noise regimes *without the knowledge of noise level*. Specifically, the convergence rate improves from $\widetilde{O}(1/T^{1/4})$ to a faster $\widetilde{O}(1/\sqrt{T})$ when $\bar{\sigma}_x$ is sufficiently small, namely $\bar{\sigma}_x = O(1/\sqrt{T})$. Notably, this automatic rate interpolation does not require prior knowledge of any problem-dependent parameters, and our proposed Ada-Minimax algorithm is fully parameter-free. In contrast, TiAda [47] does not exhibit such a bound in the low-noise regime (e.g., $\bar{\sigma}_x = O(1/\sqrt{T})$), and its convergence rate is not optimal with respect to $\bar{\sigma}_x$ in the stochastic setting since their convergence rate (e.g., Theorem 3.2 in [47]) does not explicitly depend on $\bar{\sigma}_x$. See detailed proof of Theorem 4.1 in Appendix D. A comparison of adaptive methods for minimax optimization is also presented in Table 1.

## 4.3 Adaptive Algorithm for Bilevel Optimization

Our proposed algorithm Ada-Bio is presented in Algorithm 2. The overall framework closely resembles that of Algorithm 1. The upper-level variable is updated using normalized SGD with

Table 1: Comparison of Adaptive Methods for Minimax Optimization

| Method | Setting | Assumptions | High Probability | Complexity |
|---|---|---|---|---|
| TiAda [47] | Deterministic [47, Theorem 3.1] | Assumptions 3.1 to 3.3 in [47] | | $O(1/\sqrt{T})$ |
| TiAda [47] | Stochastic [47, Theorem 3.2] | Assumptions 3.1 to 3.6 in [47] | ✗ | $O(\text{poly}(G) \cdot (T^{\frac{\alpha-1}{2}} + T^{-\frac{\alpha}{2}} + T^{\frac{\beta-1}{2}} + T^{-\frac{\beta}{2}}))$ [1] |
| Ada-Minimax | Deterministic & Stochastic (Theorem 4.1 in this work) | Assumptions 3.1 and 3.2 in this work | ✓ | $\widetilde{O}(1/\sqrt{T} + \sigma_x^{1/4}/T^{3/8} + \sqrt{\bar{\sigma}_x}/T^{1/4})$ |

Table 2: Comparison of Adaptive Methods for Bilevel Optimization

| Method | Setting | Assumptions | High Probability | Complexity |
|---|---|---|---|---|
| S-TFBO [73] | Deterministic [73, Theorem 2] | Assumptions 1 to 3 in [73] | | $\widetilde{O}(1/\sqrt{T})$ |
| Ada-Bio | Deterministic & Stochastic (Theorem 4.2 in this work) | Assumptions 3.3 and 3.4 in this work | ✓ | $\widetilde{O}(1/\sqrt{T} + \sigma_{g,1}^{1/4}/T^{3/8} + (\sqrt{\bar{\sigma}_\phi} + \sqrt{\bar{\sigma}_{g,1}})/T^{1/4})$ |

momentum [11], employing adaptive choices for the momentum parameter and learning rate. This approach differs from those of [33, 25], where fixed, non-adaptive momentum parameters and learning rates are used. The lower-level variable is updated via AdaGrad-Norm. Here in Equations (3) and (4), $g_{x,t} = \bar{\nabla} f(x_t, y_t; \bar{\xi}_t)$, $\tilde{g}_{x,t} = \bar{\nabla} f(x_t, y_t; \bar{\xi}'_t)$, and $g_{y,t} = \nabla_y G(x_t, y_t; \zeta_t)$, where $\bar{\nabla} f(x, y; \bar{\xi})$ denotes the Neumann series approximation (see Appendix E.1 for further details), with $\bar{\xi}_t, \bar{\xi}'_t$ being independent samples.

**Theorem 4.2.** *Under Assumptions 3.3 and 3.4 and the parameter choices in Equations* (3) *and* (4), *for any $\delta \in (0, 1/7)$, choose $N \geq \frac{3 \log T}{2 \log(1/(1-\mu_g/l_{g,1}))}$, it holds with probability at least $1 - 7\delta$ that*

$$\frac{1}{T} \sum_{t=1}^{T} \|\nabla \Phi(x_t)\| \leq \frac{C_b}{\eta\sqrt{\alpha}} \left( \frac{1}{\sqrt{T}} \left( \alpha^2 + \frac{l_{g,1}^2}{\mu^2\eta^2} \left( 4D^2 l_{g,1}^2 + 2D\gamma \right) \right)^{1/4} \right.$$
$$\left. + \frac{1}{T^{3/8}} \left( \frac{2\sqrt{2} l_{g,1}^2 D \sigma_{g,1}}{\mu^2\eta^2} \right)^{1/4} + \frac{1}{T^{1/4}} \left( 4\bar{\sigma}_\phi^2 + \sigma_{g,1}^2 \right)^{1/4} \right),$$

*where $C_b = \widetilde{O}(\kappa_\sigma^4)$, $D$, and $\bar{\sigma}_\phi$ are defined in Equations* (24), (44) *and* (46), *respectively.*

**Remark:** Theorem 4.2 shows that Ada-Bio achieves a sharp rate of $\widetilde{O}((\bar{\sigma}_\phi^2 + \sigma_{g,1}^2)^{1/4}/T^{1/4})$ in the stochastic setting, where all noise terms introduced in Assumption 3.4 are positive. Moreover, it is obvious that Ada-Bio implicitly adapts to the noise level; in the noiseless case (where all noise parameters in Assumption 3.4 vanish), Ada-Bio automatically recovers the near-optimal $\widetilde{O}(1/\sqrt{T})$ rate. To the best of our knowledge, Theorem 4.2 provides the first sharp and adaptive convergence guarantee for stochastic bilevel optimization without any prior knowledge of the noise parameters specified in Assumption 3.4. In fact, we only require the knowledge of $\mu_g, l_{g,1}$ and $T$ due to the construction of Neumann series. See detailed proof of Theorem 4.2 in Appendix E. A comparison of adaptive methods for bilevel optimization is also presented in Table 2.

## 5 Theoretical Analysis

In this section, we provide the convergence analysis for Algorithms 1 and 2 with the adaptive parameter choices in Equations (3) and (4). We begin in Section 5.1 by analyzing an adaptive version of normalized SGD with momentum (Algorithm 3) in the nonconvex stochastic (single-level) optimization setting, where we establish a convergence rate of $\widetilde{O}(1/\sqrt{T} + \sqrt{\bar{\sigma}}/T^{1/4})$, where $\bar{\sigma}$ is an upper bound on the stochastic gradient noise. We then extend this novel framework for the upper-level analysis in both minimax and bilevel optimization, combining it with a generalized AdaGrad-Norm analysis in the (strongly) convex case [2] under time shift for the lower-level variables, presented in Section 5.2. Due to space limitations, we defer the full proofs to Appendices C to E.

### 5.1 Adaptive Normalized SGD with Momentum

With a slight abuse of notation, we consider minimizing an objective function $f(x) = \mathbb{E}[F(x; \xi)]$. We start with analyzing adaptive normalized SGD with momentum presented in Algorithm 3, where

---

[1] $G$ denotes the upper bound on the stochastic gradient norm, and $\alpha, \beta$ satisfy $0 < \beta < \alpha < 1$.

---

**Algorithm 3** Adaptive Normalized SGD with Momentum (Ada-NSGDM)

1: **Input:** $x_1, m_1 = \nabla F(x_1; \xi_1)$
2: **for** $t = 1, \dots, T$ **do**
3:     $m_t = \beta_t m_{t-1} + (1 - \beta_t) g_t$
4:     $x_{t+1} = x_t - \eta_t \frac{m_t}{\|m_t\|}$
5: **end for**

---

$g_t = \nabla F(x_t; \xi_t)$. This algorithm builds on the method introduced by [11], with the key difference being that we incorporate both an adaptive momentum parameter $\beta_t$ and an adaptive stepsize $\eta_t$, each of which varies across iterations. In particular, let $\alpha_t = 1 - \beta_t$ and we set $\alpha_t$ and $\eta_t$ as

$$\alpha_t = \frac{\alpha}{\sqrt{\alpha^2 + \sum_{k=1}^{t} \|g_k - \tilde{g}_k\|^2}} \quad \text{and} \quad \eta_t = \frac{\eta \sqrt{\alpha_t}}{\sqrt{t}}, \tag{5}$$

where $\tilde{g}_t = \nabla F(x_t; \xi_t')$ and $\xi_t, \xi_t'$ are independent samples. We will make the following assumptions.

**Assumption 5.1.** The objective function $f$ is $L$-smooth.

**Assumption 5.2.** The gradient oracle is unbiased, i.e., $\mathbb{E}[\nabla F(x; \xi) \mid x] = \nabla f(x)$, and with probability one, satisfies $\underline{\sigma} \leq \|\nabla F(x; \xi) - \nabla f(x)\| \leq \bar{\sigma}$.

Before proceeding, we introduce the definition of $\kappa_\sigma$ and $t_0$, which will be frequently used throughout the subsequent analysis. Specifically, we define (with the convention $0/0 := 1$)

$$\kappa_\sigma = \begin{cases} \bar{\sigma}/\underline{\sigma} & \underline{\sigma} > 0 \\ 1 & \bar{\sigma} = 0 \end{cases}, \quad c_0 = \frac{\sigma^2}{4\bar{\sigma}^2 - 2\underline{\sigma}^2}, \quad \text{and} \quad t_0 = \max\left\{2, \frac{A_T(\delta) + c_0\sqrt{B_T(\delta)}}{c_0^2}\right\}, \tag{6}$$

where $A_T(\cdot)$ and $B_T(\cdot)$ are logarithmic factors (double-log in $T$) defined in Lemma A.1.

We now present the main lemmas necessary to establish Theorem 5.6. All of these lemmas rely on Assumptions 5.1 and 5.2, unless explicitly stated otherwise. The full proof of these lemmas are deferred to Appendix B. The following lemma is a standard recursion for the momentum deviation.

**Lemma 5.3.** Define $\hat{\epsilon}_t = m_t - \nabla f(x_t)$ and $\epsilon_t = g_t - \nabla f(x_t)$. Further, let $S_t = \nabla f(x_{t-1}) - \nabla f(x_t)$. For all $t \geq 1$, it holds that

$$\hat{\epsilon}_t = \beta_{2:t}\hat{\epsilon}_1 + \sum_{k=2}^{t} \beta_{(k+1):t}\alpha_k\epsilon_k + \sum_{k=2}^{t} \beta_{k:t}S_k.$$

In order to obtain a high probability bound for $\|\hat{\epsilon}_t\|$, we need the following technical lemma, which leverages the concentration bound introduced in [55, Lemma 2.4] and tools from linear programming (see Lemma F.5 in Appendix F) to resolve the difficulties arising from statistical dependency among $\alpha_t, \beta_t$, and $\epsilon_t$.

**Lemma 5.4.** Let $0 \leq \underline{\alpha}_t \leq \alpha_t \leq \bar{\alpha}_t$ and $0 \leq \underline{\beta}_t \leq \beta_t \leq \bar{\beta}_t$, where $\underline{\alpha}_t, \bar{\alpha}_t, \underline{\beta}_t,$ and $\bar{\beta}_t$ are independent of $\mathcal{F}_t$. Then with probability at least $1 - 2\delta$, it holds for all $t \leq T$ that

$$\left\|\sum_{k=2}^{t} \beta_{(k+1):t}\alpha_k\epsilon_k\right\| \leq \bar{\sigma}\sqrt{\left(1 + 32\log\frac{2T}{\delta}\right)\sum_{k=2}^{t} \bar{\beta}_{(k+1):t}^2\bar{\alpha}_k^2}. \tag{7}$$

Next, we provide high-probability lower and upper bounds for $\alpha_t$ and $\beta_t$, which help us to derive tight upper bound for the right-hand side of Equation (7) (see Lemma G.2 in Appendix G). Our analysis relies on the martingale technique developed by [6], which uses an empirical Bernstein concentration bound introduced by [35]. Recall the definition of $t_0$ as in Equation (6). Lemma 5.5 indicates that $\alpha_t$ and $\beta_t$ reliably approximate the optimal momentum parameter settings after $t_0$ iterations: they are both upper- and lower-bounded by quantities of the same order, even without prior knowledge of the noise level $\sigma$.

**Lemma 5.5.** With probability at least $1 - \delta$, for all $t \leq T$,

$$\frac{\alpha}{\sqrt{\alpha^2 + 4\bar{\sigma}^2 t}} =: \underline{\alpha}_t \leq \alpha_t \leq \bar{\alpha}_t := \mathbb{I}(t < t_0) + \frac{\alpha}{\sqrt{\alpha^2 + \underline{\sigma}^2 t}}\mathbb{I}(t \geq t_0),$$

$$\left(1 - \frac{\alpha}{\sqrt{\alpha^2 + \underline{\sigma}^2 t}}\right)\mathbb{I}(t \geq t_0) =: \underline{\beta}_t \leq \beta_t \leq \bar{\beta}_t := 1 - \frac{\alpha}{\sqrt{\alpha^2 + 4\bar{\sigma}^2 t}}.$$

Now we are ready the present our main theorem regarding Algorithm 3.

**Theorem 5.6.** *Under Assumptions 5.1 and 5.2 and the parameter choices in Equation* (5)*, for any* $\delta \in (0, 1/3)$*, it holds with probability at least* $1 - 3\delta$ *that*

$$\frac{1}{T} \sum_{t=1}^{T} \|\nabla f(x_t)\| \leq \frac{C}{\eta} \left( \frac{1}{\sqrt{T}} + \frac{2\sqrt{\bar{\sigma}}}{\sqrt{\alpha}T^{1/4}} \right),$$

*where* $C = \widetilde{O}(\kappa_\sigma^4)$ *is defined in Equation* (19)*.*

**Remark:** To our knowledge, this is the first adaptive convergence guarantee for normalized SGD with momentum. Algorithm 3 achieves a rate of $\widetilde{O}(1/T^{1/4})$ in the stochastic setting and $\widetilde{O}(1/\sqrt{T})$ in the deterministic setting. This rate-interpolation occurs automatically without requiring any prior knowledge of problem-dependent parameters. We emphasize that Theorem 5.6 builds a general analytical framework for proving Theorems 4.1 and 4.2. See Appendix B for detailed proofs.

## 5.2   Proof Sketch of Theorems 4.1 and 4.2

In this section, we present a unified lower-level analysis applicable to both minimax and bilevel optimization. Recall from Sections 1 and 3 that the bilevel optimization problem (2) reduces to the minimax optimization problem (1) when $g = -f$. Therefore, we analyze line 5 of Algorithm 1 using the function $g$ and stochastic gradient descent (instead of the original $f$ and stochastic gradient ascent): $y_{t+1} = y_t - \eta_{y,t} g_{y,t}$, where $g_{y,t} = \nabla_y G(x_t, y_t; \xi_t) = -\nabla_y F(x_t, y_t; \xi_t)$. Note that the following lemma (Lemma 5.7) as well as Lemmas C.1 and C.2 in Appendix C are applicable to the proofs of both Theorems 4.1 and 4.2.

The following lemma provides high-probability guarantees for the lower-level estimation error, which are crucial for controlling and bounding the (hyper)gradient bias. The core of our result generalizes the AdaGrad-Norm analysis developed for convex settings by [2], accommodating iteration-dependent shifts induced by the upper-level variable and incorporating our novel adaptive parameter choices as detailed in Equations (3) and (4) and Sections 4.2 and 4.3.

**Lemma 5.7.** *With probability at least* $1 - 4\delta$*, for all* $t \leq T + 1$*,* $\bar{d}_t := \max_{k \leq t} \|y_k - y_k^*\| \leq D$*, and*

$$\sum_{k=1}^{t} \|y_k - y_k^*\|^2 \leq \frac{1}{\mu^2 \eta^2} \left( 4D^2 L^2 + 2D\gamma + 2\sqrt{2}D\sigma\sqrt{t} \right), \tag{8}$$

$$\sum_{k=1}^{t} \|y_k - y_k^*\| \leq \frac{1}{\mu\eta} \left( \left( \sqrt{2}DL + \sqrt{D\gamma} \right) \sqrt{t} + \sqrt{2D\sigma}t^{3/4} \right), \tag{9}$$

*where* $D$ *is defined in Equation* (24)*, and* $\sigma = \sigma_y$ *for Algorithm 1 and* $\sigma = \sigma_{g,1}$ *for Algorithm 2.*

Combining the upper-level analysis framework introduced in Section 5.1 with the lower-level estimation error bounds (i.e., bounds on the gradient/hypergradient estimation bias) established in Lemma 5.7, we can prove Theorems 4.1 and 4.2 similarly to how we derived Theorem 5.6. The complete proofs are deferred to Appendices C to E.

# 6   Experiments

In this section, we empirically evaluate our proposed algorithms on three tasks, including synthetic test functions (Section 6.1), deep AUC maximization (Section 6.2), and hyperparameter optimization (Appendix K). In addition, we further test the robustness of our algorithms by varying several key parameters (e.g., initial learning rates, initial momentum parameter), which is included in Section 6.3. The code is available at `https://github.com/MingruiLiu-ML-Lab/adaptive-hierarchical-optimization`.

## 6.1   Synthetic Experiments

We conduct synthetic experiments on a simple one-dimensional function $f(x, y) = \cos x + xy - \frac{1}{2}y^2$, which satisfies the nonconvex-strongly-concave minimax optimization setting. It is straightforward to

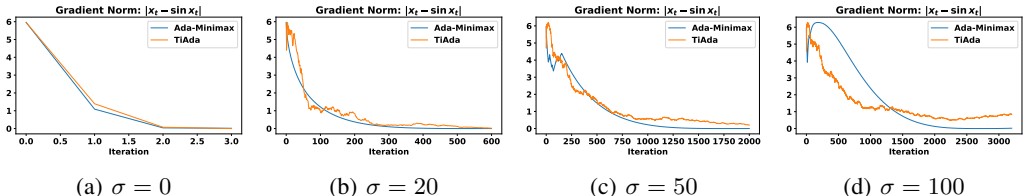

(a) $\sigma = 0$    (b) $\sigma = 20$    (c) $\sigma = 50$    (d) $\sigma = 100$

Figure 1: Synthetic experiments on a 1-dimensional function for minimax optimization.

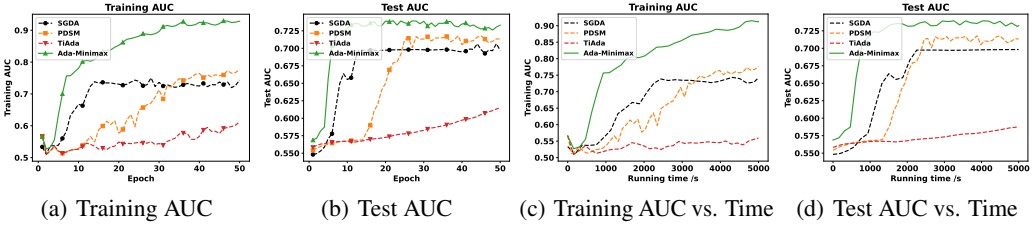

(a) Training AUC    (b) Test AUC    (c) Training AUC vs. Time    (d) Test AUC vs. Time

Figure 2: 2-layer Transformer for deep AUC maximization on imbalanced Sentiment140 dataset.

verify that $y^*(x) = x$, $\Phi(x) = f(x, y^*(x)) = \cos x + \frac{1}{2}x^2$, and $\nabla\Phi(x) = x - \sin x$. To simulate stochastic gradients, we add Gaussian noise sampled from $\mathcal{N}(0, \sigma^2)$ to the ground-truth gradients. As demonstrated in Figure 1, our proposed method, `Ada-Minimax`, consistently outperforms TiAda [47] across various noise magnitudes. These results clearly illustrate our algorithm's adaptivity to noise levels: specifically, as the noise magnitude decreases, our algorithm automatically achieves faster convergence. Notably, under high-noise regimes (e.g., $\sigma = 100$), TiAda fails to converge even after extensive parameter tuning, whereas our algorithm successfully converges. The hyperparameter settings are included in Appendix I.

## 6.2 Deep AUC Maximization

The Area Under the ROC Curve (AUC) is a performance measure of classifiers [31, 32], which is widely used in the imbalanced data classification setting. Deep AUC Maximization (DAM) [79, 76] is a new paradigm for learning a deep neural network by maximizing the AUC score of the model on a dataset. Recent studies [75, 76, 53] have shown great success of deep AUC maximization in various domains (e.g., medical image classification and drug discovery). Following [75, 54, 53], AUC maximization can be formulated as a minimax problem,

$$\min_{\boldsymbol{w} \in \mathbb{R}^d, (a,b) \in \mathbb{R}^2} \max_{\alpha \in \mathbb{R}} f(\boldsymbol{w}, a, b, \alpha) = \mathbb{E}_{\xi \sim \mathcal{D}}[F(\boldsymbol{w}, a, b, \alpha; \xi)], \tag{10}$$

where $F(\boldsymbol{w}, a, b, \alpha; \boldsymbol{z}) = (1 - p)(h(\boldsymbol{w}; \boldsymbol{x}) - a)^2 \mathbb{I}_{[y=1]} + p(h(\boldsymbol{w}; \boldsymbol{x}) - b)^2 \mathbb{I}_{[y=-1]} + 2(1 + \alpha)(ph(\boldsymbol{w}; \boldsymbol{x})\mathbb{I}_{[y=-1]} - (1 - p)h(\boldsymbol{w}; \boldsymbol{x})\mathbb{I}_{[y=1]}) - p(1 - p)\alpha^2$, $\boldsymbol{w}$ is the parameter of a deep neural network (e.g., a two-layer transformer as the predictive model), $h(\boldsymbol{w}; \boldsymbol{x})$ is the score function parameterized by $\boldsymbol{w}$ with the input data $\boldsymbol{x}$, $\xi = (\boldsymbol{x}, y)$ is a random sample from training set $\mathcal{D}$ with input $\boldsymbol{x}$ and a binary label $y \in \{-1, 1\}$. The imbalanced ratio $p$ is the proportion of the positive samples in the training set. Therefore, $(\boldsymbol{w}, a, b)$ and $\alpha$ are primal and dual variables respectively.

To verify the effectiveness of our proposed Algorithm 1, we run a practical variant (refer to Appendix J) of our algorithm in deep AUC maximization experiments on imbalanced text classification, and compare with other minimax baselines, including SGDA [50], PDSM [30]), and an adaptive minimax algorithm TiAda [47]. We first construct the imbalanced binary classification dataset Sentiment140 [23] (under Creative Commons Attribution 4.0 License). The practical variant of our algorithm replaces the term $\sum_{k=1}^{t} \|g_{x,k} - \tilde{g}_{x,k}\|^2$ in Equation (3) with $\sum_{k=1}^{t} \|g_{x,k} - g_{x,k-1}\|^2$, where $g_{x,k-1}$ denotes the gradient of $x$ computed at the previous iteration (i.e., $(k-1)$-th iteration). Additionally, we modify the step size from $\eta_{x,t} = \eta_x \sqrt{\alpha'_t}/\sqrt{t}$ to $\eta_{x,t} = \eta_x \sqrt{\alpha'_t}/\sqrt{T}$ (note that this change does not affect the convergence of Algorithm 1). In this subsection, with a slight abuse of notation, we use $\eta_x$ to denote $\eta_x/\sqrt{T}$. Following the data setting in [76], we randomly remove

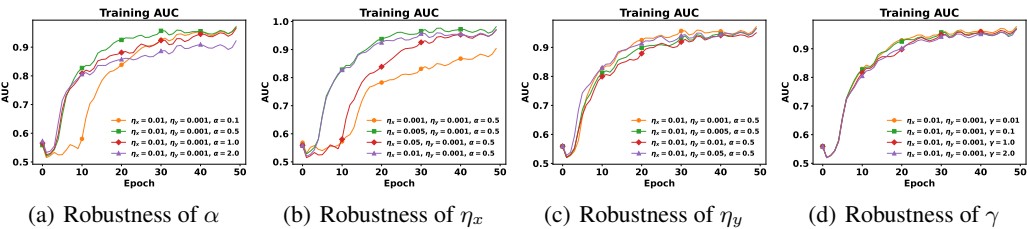

| (a) Robustness of $\alpha$ | (b) Robustness of $\eta_x$ | (c) Robustness of $\eta_y$ | (d) Robustness of $\gamma$ |

Figure 3: Robustness of hyperparameters.

positive samples (labeled as 1) from the training set until the proportion of positive samples is exactly 0.9 (i.e., $p = 0.9$). In the experiment, we adopt a two-layer transformer as the classifier with the hidden size of 4096 and the output dimension of 2. The hyperparameter settings of each baseline and experimental details are described in Appendix J. The comparison results of training and test curve over 50 epochs are shown in Figure 2. From subfigures (a) and (b), our algorithm Ada-Minimax shows 20% higher training AUC and 2% higher test AUC than the best compared algorithm PDSM. From running time curve (c) and (d), our algorithm demonstrates the fastest convergence rate than other baselines.

## 6.3 Hyperparameter Robustness Analysis

We investigate the robustness of our method to the hyperparameters $\alpha$, $\eta_y$, $\eta_x$, and $\gamma$ by varying each parameter independently while keeping others fixed, as shown in Figure 3. Specifically, Figure 3(a) indicates that changing $\alpha$ within the range $[0.1, 2.0]$ has minimal impact on convergence speed and final AUC performance. In Figure 3(b), varying $\eta_y$ between 0.001 and 0.05 yield nearly overlapping curves after the initial training stage. Similarly, Figure 3(c) shows that varying $\eta_x$ from 0.001 to 0.01 affects only early-stage training dynamics without compromising the final performance; however, increasing $\eta_x$ to 0.05 results in a noticeable decline in the final training AUC. Lastly, Figure 3(d) illustrates that the algorithm maintains consistent performance across a wide range of $\gamma$ values $[0.01, 2.0]$. Therefore, these results demonstrate that our algorithm exhibits strong robustness across broad ranges of these hyperparameters, significantly reducing the time required for hyperparameter tuning in practice.

## 7 Conclusion

We introduced two novel adaptive algorithms for nonconvex-strongly-concave minimax optimization and nonconvex-strongly-convex bilevel optimization. Both algorithms achieve sharp and adaptive convergence rates: they automatically adapt to unknown variance in stochastic gradient estimates. Our approach leverages the momentum normalization framework along with novel adaptive schemes for jointly setting the momentum parameter and the learning rate. Experimental results validate and support our theoretical analyses. One limitation of our work is the assumption that the stochastic gradient noise is lower-bounded. In future work, we aim to remove this assumption while maintaining the sharp convergence guarantees.

## Acknowledgements

We would like to thank the anonymous reviewers for their helpful comments. We would like to thank Francesco Orabona and El Mehdi Saad for helpful discussions about the concentration inequalities. This work has been supported by the Presidential Scholarship, the ORIEI seed funding, and the IDIA P3 fellowship from George Mason University, and NSF award #2436217, #2425687. The Computations were run on Hopper, a research computing cluster provided by the Office of Research Computing at George Mason University (URL: https://orc.gmu.edu).

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

# Contents

# A   Martingale Concentration Bounds and Basic Inequalities

**Lemma A.1** ([6, Corollary 1]). *Let $X_t$ be adapted to $\mathcal{F}_t$ such that $|X_t| \leq 1$ with probability 1 for all t. Then, for every $\delta \in (0,1)$ and any $\hat{X}_t \in \mathcal{F}_{t-1}$ such that $|\hat{X}_t| \leq 1$ with probability 1,*

$$\Pr\left(\exists t < \infty : \left|\sum_{s \leq t}(X_s - \mathbb{E}[X_s \mid \mathcal{F}_{s-1}])\right| \geq \sqrt{A_t(\delta)\sum_{s \leq t}(X_s - \hat{X}_s)^2 + B_t(\delta)}\right) \leq \delta,$$

*where $A_t(\delta) = 16\log\left(\frac{60\log(6t)}{\delta}\right)$ and $B_t(\delta) = 16\log^2\left(\frac{60\log(6t)}{\delta}\right)$.*

**Lemma A.2** ([55, Lemma 2.4]). *Suppose $X_1, \ldots, X_T$ is a martingale difference sequence adapted to a filtration $\mathcal{F}_1, \ldots, \mathcal{F}_T$ in a Hilbert space such that $\|X_t\| \leq R_t$ almost surely for some $R_t \in \mathcal{F}_{t-1}$. Then for any $\delta \in (0,1)$, with probability at least $1-\delta$, for any fixed t we have*

$$\left\|\sum_{s=1}^{t} X_s\right\| \leq 4\sqrt{\log\frac{2}{\delta}\sum_{s=1}^{T} R_s^2}.$$

*Proof of Lemma A.2.*  The proof concludes by setting $R_t \in \mathcal{F}_{t-1}$ in [55, Lemma 2.4].  □

**Lemma A.3** ([2, Lemma 4]). *Let $g_1, \ldots, g_T \in \mathbb{R}^d$ be an arbitrary sequence of vectors, and let $G_0 > 0$. For all $t \geq 1$, define*

$$G_t = \sqrt{G_0^2 + \sum_{s=1}^{t}\|g_s\|^2}.$$

*Then*

$$\sum_{t=1}^{T}\frac{\|g_t\|^2}{G_t} \leq 2\sqrt{\sum_{t=1}^{T}\|g_t\|^2}, \qquad and \qquad \sum_{t=1}^{T}\frac{\|g_t\|^2}{G_t^2} \leq 2\log\frac{G_T}{G_0}.$$

**Lemma A.4** ([2, Lemma 6]). *For Ada-NSGDM (Algorithm 3) we have*

$$\sum_{t=1}^{T}\frac{\|g_t\|^2}{G_t^2} \leq C_1 := \log\left(1 + \frac{2\bar{\sigma}^2 T + 4\eta^2 L^2 T^3 + 8L\Delta_1 T}{\gamma^2}\right), \tag{11}$$

*where $\Delta_1 = f(x_1) - f^*$.*

# B   Proofs of Section 5.1

## B.1   Technical Lemmas

**Lemma 5.3.** *Define $\hat{\epsilon}_t = m_t - \nabla f(x_t)$ and $\epsilon_t = g_t - \nabla f(x_t)$. Further, let $S_t = \nabla f(x_{t-1}) - \nabla f(x_t)$. For all $t \geq 1$, it holds that*

$$\hat{\epsilon}_t = \beta_{2:t}\hat{\epsilon}_1 + \sum_{k=2}^{t}\beta_{(k+1):t}\alpha_k\epsilon_k + \sum_{k=2}^{t}\beta_{k:t}S_k.$$

*Proof of Lemma 5.3.* The proof follows from a straightforward calculation:

$$\begin{aligned}
\hat{\epsilon}_t &= m_t - \nabla f(x_t) \\
&= \beta_t m_{t-1} + (1 - \beta_t)g_t - \nabla f(x_t) \\
&= \beta_t(\hat{\epsilon}_{t-1} + \nabla f(x_{t-1})) + (1 - \beta_t)(\epsilon_t + \nabla f(x_t)) - \nabla f(x_t) \\
&= \beta_t\hat{\epsilon}_{t-1} + (1 - \beta_t)\epsilon_t + \beta_t S_t.
\end{aligned}$$

Unrolling the recursion and using $\alpha_t = 1 - \beta_t$ yields the result. $\qquad\square$

**Lemma 5.4.** *Let $0 \leq \underline{\alpha}_t \leq \alpha_t \leq \bar{\alpha}_t$ and $0 \leq \underline{\beta}_t \leq \beta_t \leq \bar{\beta}_t$, where $\underline{\alpha}_t, \bar{\alpha}_t, \underline{\beta}_t$, and $\bar{\beta}_t$ are independent of $\mathcal{F}_t$. Then with probability at least $1 - 2\delta$, it holds for all $t \leq T$ that*

$$\left\|\sum_{k=2}^t \beta_{(k+1):t}\alpha_k\epsilon_k\right\| \leq \bar{\sigma}\sqrt{\left(1 + 32\log\frac{2T}{\delta}\right)\sum_{k=2}^t \bar{\beta}_{(k+1):t}^2\bar{\alpha}_k^2}. \tag{7}$$

*Proof of Lemma 5.4.* Define $\gamma_{k,t} := \beta_{(k+1):t}\alpha_k$ and $\mathcal{I}_t = \{(i,j) \mid 3 \leq i \leq t,\, 2 \leq j < i\}$. Then for all $k \leq t$,

$$\underline{\beta}_{(k+1):t}\underline{\alpha}_k =: \underline{\gamma}_{k,t} \leq \gamma_{k,t} \leq \bar{\gamma}_{k,t} := \bar{\beta}_{(k+1):t}\bar{\alpha}_k. \tag{12}$$

By Lemma F.5, there exists a set $\{b_{ij,t}^*\}_{(i,j)\in\mathcal{I}_t}$ with each $b_{ij,t}^*$ satisfying either $b_{ij,t}^* = \underline{\gamma}_{i,t}\underline{\gamma}_{j,t}$ or $b_{ij,t}^* = \bar{\gamma}_{i,t}\bar{\gamma}_{j,t}$ for every pair $(i,j)$, such that

$$\sum_{i=3}^t \sum_{j=2}^{i-1} \gamma_{i,t}\gamma_{j,t}\langle\epsilon_i, \epsilon_j\rangle \leq \sum_{i=3}^t \sum_{j=2}^{i-1} b_{ij,t}^*\langle\epsilon_i, \epsilon_j\rangle.$$

Applying Lemma A.2 with $X_i = \left\langle\epsilon_i, \sum_{j=2}^{i-1} b_{ij,t}^*\epsilon_j\right\rangle$ and $R_i = \bar{\sigma}\left\|\sum_{j=2}^{i-1} b_{ij,t}^*\epsilon_j\right\| \in \mathcal{F}_{i-1}$, and using a union bound over $t$, with probability at least $1 - \delta$, for all $t \leq T$,

$$\sum_{i=3}^t \sum_{j=2}^{i-1} b_{ij,t}^*\langle\epsilon_i, \epsilon_j\rangle = \sum_{i=3}^t \left\langle\epsilon_i, \sum_{j=2}^{i-1} b_{ij,t}^*\epsilon_j\right\rangle \leq 4\sqrt{\log\frac{2T}{\delta}\sum_{i=3}^t \bar{\sigma}^2\left\|\sum_{j=2}^{i-1} b_{ij,t}^*\epsilon_j\right\|^2}. \tag{13}$$

Applying Lemma A.2 again with $X_j = b_{ij,t}^*\epsilon_j$ and $R_j = b_{ij,t}^*\bar{\sigma} \in \mathbb{R}$, and using a union bound over $i$, with probability at least $1 - \delta$, for all $i \leq T$,

$$\left\|\sum_{j=2}^{i-1} b_{ij,t}^*\epsilon_j\right\|^2 \leq 16\log\frac{2T}{\delta}\sum_{j=2}^{i-1}(b_{ij,t}^*\bar{\sigma})^2. \tag{14}$$

Combing Equations (13) and (14), with probability at least $1 - 2\delta$ (via a union bound), for all $t \leq T$,

$$\begin{aligned}
\sum_{i=3}^t \sum_{j=2}^{i-1} b_{ij,t}^*\langle\epsilon_i, \epsilon_j\rangle &\leq 16\bar{\sigma}^2\log\frac{2T}{\delta}\sqrt{\sum_{i=3}^t \sum_{j=2}^{i-1}(b_{ij,t}^*)^2} \\
&\leq 16\bar{\sigma}^2\log\frac{2T}{\delta}\sqrt{\sum_{i=3}^t \sum_{j=2}^{i-1}(\bar{\gamma}_{i,t}\bar{\gamma}_{j,t})^2} \leq 16\log\frac{2T}{\delta}\sum_{i=2}^t \bar{\gamma}_{i,t}^2\bar{\sigma}^2,
\end{aligned}$$

where the second inequality uses $b_{ij,t}^* \leq \bar{\gamma}_{i,t}\bar{\gamma}_{j,t}$. Hence, with probability at least $1 - 2\delta$,

$$\begin{aligned}
\left\|\sum_{k=2}^t \beta_{(k+1):t}\alpha_k\epsilon_k\right\|^2 &= \sum_{k=2}^t \gamma_{k,t}^2\|\epsilon_k\|^2 + 2\sum_{i=3}^t \sum_{j=2}^{i-1} \gamma_{i,t}\gamma_{j,t}\langle\epsilon_i, \epsilon_j\rangle \\
&\leq \sum_{k=2}^t \bar{\gamma}_{k,t}^2\bar{\sigma}^2 + 32\log\frac{2T}{\delta}\sum_{i=2}^t \bar{\gamma}_{i,t}^2\bar{\sigma}^2 \\
&\leq \left(1 + 32\log\frac{2T}{\delta}\right)\sum_{k=2}^t \bar{\gamma}_{k,t}^2\bar{\sigma}^2.
\end{aligned}$$

Plugging in the definition of $\bar{\gamma}_{k,t}$ as in Equation (12) yields the result. $\qquad\square$

**Lemma 5.5.** *With probability at least $1 - \delta$, for all $t \leq T$,*

$$\frac{\alpha}{\sqrt{\alpha^2 + 4\bar{\sigma}^2 t}} =: \underline{\alpha}_t \leq \alpha_t \leq \bar{\alpha}_t := \mathbb{I}(t < t_0) + \frac{\alpha}{\sqrt{\alpha^2 + \underline{\sigma}^2 t}} \mathbb{I}(t \geq t_0),$$

$$\left(1 - \frac{\alpha}{\sqrt{\alpha^2 + \underline{\sigma}^2 t}}\right) \mathbb{I}(t \geq t_0) =: \underline{\beta}_t \leq \beta_t \leq \bar{\beta}_t := 1 - \frac{\alpha}{\sqrt{\alpha^2 + 4\bar{\sigma}^2 t}}.$$

*Proof of Lemma 5.5.* Consider the case $0 < \underline{\sigma} \leq \bar{\sigma}$. By Assumption 5.2 and Young's inequality,

$$\sum_{k=1}^{t} \|g_k - \tilde{g}_k\|^2 \leq 2 \sum_{k=1}^{t} \|g_k - \nabla f(x_k)\|^2 + \|\tilde{g}_k - \nabla f(x_k)\|^2 \leq 4\bar{\sigma}^2 t. \tag{15}$$

We proceed to derive high probability lower bound for $\sum_{k=1}^{t} \|g_k - \tilde{g}_k\|^2$. Denote $\sigma_t^2 = \mathbb{E}_{t-1}[\|g_t - \nabla f(x_t)\|^2]$. Let $Z_t = \|g_t - \tilde{g}_t\|^2 - 2\sigma_t^2$, then $\{Z_t\}_{t \geq 1}$ is a martingale difference sequence since

$$\mathbb{E}_{t-1}[Z_t] = \mathbb{E}_{t-1}[\|g_t - \tilde{g}_t\|^2 - 2\sigma_t^2]$$
$$= \mathbb{E}_{t-1}[\|g_t - \nabla f(x_t)\|^2 + \|\tilde{g}_t - \nabla f(x_t)\|^2 - 2\langle g_t - \nabla f(x_t), \tilde{g}_t - \nabla f(x_t)\rangle] - 2\sigma_t^2$$
$$= 0.$$

Using Assumption 5.2 and Young's inequality again, we have

$$Z_t \geq -2\sigma_t^2 \quad \text{and} \quad Z_t \leq 2\|g_t - \nabla f(x_t)\|^2 + 2\|\tilde{g}_t - \nabla f(x_t)\|^2 - 2\sigma_t^2 \leq 4\bar{\sigma}^2 - 2\sigma_t^2.$$

This implies that

$$|Z_t| \leq \max\left\{2\sigma_t^2, 4\bar{\sigma}^2 - 2\sigma_t^2\right\} = 4\bar{\sigma}^2 - 2\sigma_t^2,$$

where the last equality is due to $\sigma_t \leq \bar{\sigma}$ almost surely. Define $X_t = Z_t/(4\bar{\sigma}^2 - 2\sigma_t^2)$, then $|X_t| \leq 1$ with probability 1. Applying Lemma A.1 with the $X_s$ we defined and $\hat{X}_s = 0$, for any $\delta \in (0, 1)$, with probability at least $1 - \delta$, for all $t \leq T$,

$$\left|\sum_{k=1}^{t} X_k\right| \leq \sqrt{A_t(\delta) \sum_{k=1}^{t} X_k^2 + B_t(\delta)} \leq \sqrt{A_t(\delta) \cdot t + B_t(\delta)}, \tag{16}$$

where the last inequality uses $\sum_{k=1}^{t} X_k^2 \leq t$ since $|X_k| \leq 1$. Recall the definition of $t_0$ and $c_0$ as in Equation (6) ($t_0$ is the solution to the equation $A_T(\delta) \cdot t + B_T(\delta) = c_0^2 t^2$), for all $t \geq t_0$,

$$\sqrt{A_t(\delta) \cdot t + B_t(\delta)} \leq \sqrt{A_T(\delta) \cdot t + B_T(\delta)} \leq c_0 t = \frac{\underline{\sigma}^2 t}{4\bar{\sigma}^2 - 2\underline{\sigma}^2}.$$

Then, expanding Equation (16) and using the above condition yields that, with probability at least $1 - \delta$, for all $t_0 \leq t \leq T$,

$$\sum_{k=1}^{t} \frac{\|g_k - \tilde{g}_k\|^2 - 2\sigma_k^2}{4\bar{\sigma}^2 - 2\sigma_k^2} \geq -\frac{\underline{\sigma}^2 t}{4\bar{\sigma}^2 - 2\underline{\sigma}^2} \quad \Longrightarrow \quad \sum_{k=1}^{t} \|g_k - \tilde{g}_k\|^2 \geq \underline{\sigma}^2 t. \tag{17}$$

We conclude the proof by combining Equations (15) and (17) and noting that the results also hold for the case $\underline{\sigma} = \bar{\sigma} = 0$. $\qquad\square$

**Lemma B.1** (Descent Lemma). *Under Assumptions 5.1 and 5.2, define $\hat{\epsilon}_t := m_t - \nabla f(x_t)$, then*

$$f(x_{t+1}) \leq f(x_t) - \eta_t \|\nabla f(x_t)\| + 2\eta_t \|\hat{\epsilon}_t\| + \frac{L\eta_t^2}{2}.$$

*Further, define $\Delta_1 := f(x_1) - f^*$, taking summation and rearranging we have*

$$\sum_{t=1}^{T} \eta_t \|\nabla f(x_t)\| \leq \Delta_1 + 2 \sum_{t=1}^{T} \eta_t \|\hat{\epsilon}_t\| + \frac{L}{2} \sum_{t=1}^{T} \eta_t^2.$$

## B.2  Proof of Theorem 5.6

Before proving Theorem 5.6, let us define (recall the definition of $\kappa_\sigma$ and $t_0$ in Equation (6), here $\kappa_\sigma = \bar{\sigma}/\sigma$ in single-level optimization)

$$\Delta_1 = f(x_1) - f^*, \quad t_0 = \max\left\{2, 8\left(32\kappa_\sigma^4 - 30\kappa_\sigma^2 + 7\right)\log\left(\frac{60\log(6T)}{\delta}\right)\right\}, \tag{18}$$

$$\begin{aligned}
C = {} & \Delta_1 + 4\eta\bar{\sigma}\left(\sqrt{t_0 - 1} - 1 + e\sqrt{\kappa_\sigma}\left(1 + 2\sqrt{\bar{\sigma}/\alpha}\right)\right) + \frac{L\eta^2}{2}(1 + \log T) \\
& + 2\eta\sqrt{1 + 32\log(2T/\delta)}\left(\left(t_0 - 1 + 2e\sqrt{t_0 - 2}\sqrt{\kappa_\sigma}\left(1 + 2\sqrt{\bar{\sigma}/\alpha}\right)\right)\bar{\sigma} + 3\sqrt{e}\kappa_\sigma^2\alpha\log T\right) \\
& + 4L\eta^2\left((t_0 - 1)\left(1 + e\sqrt{\kappa_\sigma}\left(1 + 2\sqrt{\bar{\sigma}/\alpha}\right)\right) + e(\sqrt{\kappa_\sigma} + 2\kappa_\sigma)\log T\right).
\end{aligned} \tag{19}$$

**Theorem 5.6.** *Under Assumptions 5.1 and 5.2 and the parameter choices in Equation* (5)*, for any $\delta \in (0, 1/3)$, it holds with probability at least $1 - 3\delta$ that*

$$\frac{1}{T}\sum_{t=1}^{T}\|\nabla f(x_t)\| \leq \frac{C}{\eta}\left(\frac{1}{\sqrt{T}} + \frac{2\sqrt{\bar{\sigma}}}{\sqrt{\alpha}T^{1/4}}\right),$$

*where $C = \widetilde{O}(\kappa_\sigma^4)$ is defined in Equation* (19)*.*

*Proof of Theorem 5.6.* Without loss of generality, we assume $t_0$ is an integer (see definition in Equation (6)). By Lemmas 5.3 to 5.5, B.1 and G.2, with probability at least $1 - 3\delta$,

$$\begin{aligned}
\sum_{t=1}^{T}\eta_t\|\nabla f(x_t)\| &\leq \Delta_1 + 2\sum_{t=1}^{T}\eta_t\|\hat{\epsilon}_t\| + \frac{L}{2}\sum_{t=1}^{T}\eta_t^2 \\
&\leq \Delta_1 + 2\sum_{t=1}^{T}\eta_t\left(\beta_{2:t}\|\hat{\epsilon}_1\| + \left\|\sum_{k=2}^{t}\beta_{(k+1):t}\alpha_k\epsilon_k\right\| + \sum_{k=2}^{t}\beta_{k:t}\|S_k\|\right) + \frac{L}{2}\sum_{t=1}^{T}\eta_t^2 \\
&\leq \Delta_1 + 2\sum_{t=1}^{T}\eta_t\left(\beta_{2:t}\bar{\sigma} + \bar{\sigma}\sqrt{\left(1 + 32\log\frac{2T}{\delta}\right)\sum_{k=2}^{t}\bar{\beta}_{(k+1):t}^2\bar{\alpha}_k^2} + L\sum_{k=2}^{t}\beta_{k:t}\eta_{k-1}\right) + \frac{L}{2}\sum_{t=1}^{T}\eta_t^2 \\
&\leq \Delta_1 + 4\eta\bar{\sigma}\left(\sqrt{t_0 - 1} - 1 + e\sqrt{\kappa_\sigma}\left(1 + 2\sqrt{\bar{\sigma}/\alpha}\right)\right) + \frac{L\eta^2}{2}(1 + \log T) \\
&\quad + 2\eta\sqrt{1 + 32\log(2T/\delta)}\left(\left(t_0 - 1 + 2e\sqrt{t_0 - 2}\sqrt{\kappa_\sigma}\left(1 + 2\sqrt{\bar{\sigma}/\alpha}\right)\right)\bar{\sigma} + 3\sqrt{e}\kappa_\sigma^2\alpha\log T\right) \\
&\quad + 4L\eta^2\left((t_0 - 1)\left(1 + e\sqrt{\kappa_\sigma}\left(1 + 2\sqrt{\bar{\sigma}/\alpha}\right)\right) + e(\sqrt{\kappa_\sigma} + 2\kappa_\sigma)\log T\right) \\
&= C,
\end{aligned}$$

where the third inequality uses $\|\hat{\epsilon}_1\| = \|\epsilon_1\| \leq \bar{\sigma}$ and $\|S_k\| = \|\nabla f(x_{k-1}) - \nabla f(x_k)\| \leq L\eta_{k-1}$, and the last inequality is due to the definition of $C$. Then, using $\eta_t \geq \eta_T$ for $t \leq T$,

$$\sum_{t=1}^{T}\eta_T\|\nabla f(x_t)\| \leq \sum_{t=1}^{T}\eta_t\|\nabla f(x_t)\| \leq C.$$

Therefore, with probability at least $1 - 3\delta$,

$$\frac{1}{T}\sum_{t=1}^{T}\|\nabla f(x_t)\| \leq \frac{C}{T\eta_T} \leq \frac{C(\alpha^2 + 4\bar{\sigma}^2T)^{1/4}\sqrt{T}}{\eta\sqrt{\alpha}T} \leq \frac{C}{\eta}\left(\frac{1}{\sqrt{T}} + \frac{2\sqrt{\bar{\sigma}}}{\sqrt{\alpha}T^{1/4}}\right).$$

$\square$

## C  Proof of Section 5.2

The core of our result in this section generalizes the AdaGrad-Norm analysis developed for convex settings by [2], accommodating iteration-dependent shifts induced by the upper-level variable $x_t$ and

incorporating our novel adaptive parameter choices as detailed in Equations (3) and (4) and Sections 4.2 and 4.3. In particular, Lemmas C.1 and C.2 are direct applications of [2, Lemmas 15 and 16], whereas Lemma 5.7 extends [2, Lemma 13] to account for time shifts.

Recall from Sections 1 and 3 that the bilevel optimization problem (2) reduces to the minimax optimization problem (1) when $g = -f$. Therefore, we analyze line 5 of Algorithm 1 using the function $g$ and stochastic gradient descent (instead of the original $f$ and stochastic gradient ascent): $y_{t+1} = y_t - \eta_{y,t} g_{y,t}$, where $g_{y,t} = \nabla_y G(x_t, y_t; \xi_t) = -\nabla_y F(x_t, y_t; \xi_t)$. Note that the following lemmas (Lemmas 5.7, C.1 and C.2) are applicable to the proofs of both Theorems 4.1 and 4.2.

**Additional Notations.** Let $y_t^* = y^*(x_t)$ and $d_t = \|y_t - y_t^*\|$. Define $\bar{d}_t := \max_{k \leq t} d_t$. In the proof below, we use a "decorrelated step size" given by

$$\hat{\eta}_{y,t} := \frac{\eta}{\sqrt{G_{y,t-1}^2 + \|\nabla_y g(x_t, y_t)\|^2}}, \quad \text{where} \quad G_{y,t} = \sqrt{\gamma^2 + \sum_{k=1}^t \|g_{y,k}\|^2}.$$

**Lemma C.1.** *Let $\bar{d}_t' = \max\{\bar{d}_t, \eta\}$. Then with probability at least $1 - 2\delta$, it holds that for all $t \leq T$,*

$$\sum_{k=1}^t \hat{\eta}_{y,k} \langle \nabla_y g(x_k, y_k) - g_{y,k}, y_k - y_k^* \rangle$$

$$\leq 2\bar{d}_t' \sqrt{A_t(\delta/\log(4T)) \sum_{k \leq t} \eta_{y,k-1}^2 \|\nabla_y g(x_k, y_k) - g_{y,k}\|^2 + \eta_{y,0}^2 \sigma^2 B_t(\delta/\log(4T))}$$

*and*

$$\sum_{k=1}^t \hat{\eta}_{y,k}^2 \langle \nabla_y g(x_k, y_k) - g_{y,k}, y_k - y_k^* \rangle$$

$$\leq 2\bar{d}_t' \eta_{y,0} \sqrt{A_t(\delta/\log(4T)) \sum_{k \leq t} \eta_{y,k-1}^2 \|\nabla_y g(x_k, y_k) - g_{y,k}\|^2 + \eta_{y,0}^2 \sigma^2 B_t(\delta/\log(4T))},$$

*where $A_t(\cdot)$ and $B_t(\cdot)$ are defined in Lemma A.1, and $\sigma = \sigma_y$ for Algorithm 1 and $\sigma = \sigma_{g,1}$ for Algorithm 2.*

*Proof of Lemma C.1.* In order to invoke Lemma A.1 we will replace $y_k - y_k^*$ with a version which is scaled and projected to the unit ball. We denote $a_s = 2^{s-1} \bar{d}_1'$ and $s_t = \lceil \log(\bar{d}_t'/\bar{d}_1') \rceil + 1$. Thus, $\bar{d}_t \leq \bar{d}_t' \leq a_{s_t} \leq 2\bar{d}_t'$. Since $\|y_{k+1} - y_{k+1}^*\| \leq \|y_k - y_k^*\| + \eta$ for all $s$, $\bar{d}_t \leq d_1 + \eta(t-1)$ and $1 \leq s_t \leq \lceil \log(t) \rceil + 1 \leq \log(4T)$. Defining the projection to the unit ball, $\Pi_1(x) = x/\max\{1, \|x\|\}$,

$$\frac{y_k - y_k^*}{a_{s_t}} = \Pi_1 \left( \frac{y_k - y_k^*}{a_{s_t}} \right)$$

Note that

$$\|\hat{\eta}_{y,k}(\nabla_y g(x_k, y_k) - g_{y,k})\| \leq \eta_{y,0} \sigma. \tag{20}$$

Thus,

$$\sum_{k=1}^t \frac{\hat{\eta}_{y,k} \langle \nabla_y g(x_k, y_k) - g_{y,k}, y_k - y_k^* \rangle}{\eta_{y,0} \sigma a_{s_t}} = \sum_{k=1}^t \left\langle \frac{\hat{\eta}_{y,k}(\nabla_y g(x_k, y_k) - g_{y,k})}{\eta_{y,0} \sigma}, \Pi_1 \left( \frac{y_k - y_k^*}{a_{s_t}} \right) \right\rangle$$

$$\leq \left| \sum_{k=1}^t \left\langle \frac{\hat{\eta}_{y,k}(\nabla_y g(x_k, y_k) - g_{y,k})}{\eta_{y,0} \sigma}, \Pi_1 \left( \frac{y_k - y_k^*}{a_{s_t}} \right) \right\rangle \right| \tag{21}$$

$$\leq \max_{1 \leq s \leq \lfloor \log(4T) \rfloor} \left| \sum_{k=1}^t \left\langle \frac{\hat{\eta}_{y,k}(\nabla_y g(x_k, y_k) - g_{y,k})}{\eta_{y,0} \sigma}, \Pi_1 \left( \frac{y_k - y_k^*}{a_s} \right) \right\rangle \right|.$$

Let $X_k^{(s)}$ be defined as

$$X_k^{(s)} = \left\langle \frac{\hat{\eta}_{y,k}(\nabla_y g(x_k, y_k) - g_{y,k})}{\eta_{y,0} \sigma}, \Pi_1 \left( \frac{y_k - y_k^*}{a_s} \right) \right\rangle$$

for some $s$. Then $X_k^{(s)}$ is a martingale difference sequence since

$$\mathbb{E}_{k-1}[g_{y,k}] = \nabla_y g(x_k, y_k) \quad \implies \quad \mathbb{E}_k[X_k^{(s)}] = 0.$$

Also note that $X_k^{(s)} \le 1$ with probability 1. Using Lemma A.1 with the $X_k^{(s)}$ we defined and $\hat{X}_k = 0$, for any $k$ and $\delta' \in (0,1)$, with probability at least $1 - \delta'$, for all $t \le T$,

$$\left| \sum_{k \le t} X_k^{(s)} \right| \le \sqrt{A_t(\delta') \sum_{k \le t} (X_k^{(s)})^2 + B_t(\delta')}.$$

We can upper bound $(X_k^{(s)})^2$,

$$\begin{aligned}
(X_k^{(s)})^2 &\le \frac{\hat{\eta}_{y,k}^2 \|\nabla_y g(x_k, y_k) - g_{y,k}\|^2}{(\eta_{y,0}\sigma)^2} \left\| \Pi_1 \left( \frac{y_k - y_k^*}{a_k} \right) \right\|^2 \\
&\le \frac{\hat{\eta}_{y,k}^2 \|\nabla_y g(x_k, y_k) - g_{y,k}\|^2}{(\eta_{y,0}\sigma)^2} \\
&\le \frac{\eta_{y,k-1}^2 \|\nabla_y g(x_k, y_k) - g_{y,k}\|^2}{(\eta_{y,0}\sigma)^2},
\end{aligned}$$

where the first inequality uses Cauchy-Schwarz inequality, the second inequality is due to $\|\Pi_1(x)\| \le 1$, and the last inequality uses $\hat{\eta}_{y,k} \le \eta_{y,k-1}$. Thus, returning to Equation (21) multiplied by $\eta_{y,0}\sigma a_{s_t}$, with probability at least $1 - \delta' \log(4T)$ (union bound for all $1 \le s \le \lfloor \log(4T) \rfloor$),

$$\sum_{k=1}^t \hat{\eta}_{y,k} \langle \nabla_y g(x_k, y_k) - g_{y,k}, y_k - y_k^* \rangle \le \eta_{y,0}\sigma a_{s_t} \sqrt{A_t(\delta') \sum_{k \le t} \frac{\eta_{y,k-1}^2}{(\eta_{y,0}\sigma)^2} \|\nabla_y g(x_k, y_k) - g_{y,k}\|^2 + B_t(\delta')}.$$

As $a_{s_t} \le 2\bar{d}_t'$, picking $\delta' = \delta/\log(4T)$, with probability at least $1 - \delta$,

$$\sum_{k=1}^t \hat{\eta}_{y,k} \langle \nabla_y g(x_k, y_k) - g_{y,k}, y_k - y_k^* \rangle \tag{22}$$

$$\le 2\bar{d}_t' \sqrt{A_t(\delta/\log(4T)) \sum_{k \le t} \eta_{y,k-1}^2 \|\nabla_y g(x_k, y_k) - g_{y,k}\|^2 + \eta_{y,0}^2 \sigma^2 B_t(\delta/\log(4T))}.$$

Similarly, replacing $\hat{\eta}_{y,k}$ by $\hat{\eta}_{y,k}^2$ and $\eta_{y,0}\sigma$ by $\eta_{y,0}^2\sigma$ in Equation (20), following the analysis above, and using $\eta_{y,k-1} \le \eta_{y,0}$ yields that, with probability at least $1 - \delta$,

$$\sum_{k=1}^t \hat{\eta}_{y,k}^2 \langle \nabla_y g(x_k, y_k) - g_{y,k}, y_k - y_k^* \rangle \tag{23}$$

$$\le 2\bar{d}_t' \sqrt{A_t(\delta/\log(4T)) \sum_{k \le t} \eta_{y,k-1}^4 \|\nabla_y g(x_k, y_k) - g_{y,k}\|^2 + \eta_{y,0}^4 \sigma^2 B_t(\delta/\log(4T))}$$

$$\le 2\bar{d}_t' \eta_{y,0} \sqrt{A_t(\delta/\log(4T)) \sum_{k \le t} \eta_{y,k-1}^2 \|\nabla_y g(x_k, y_k) - g_{y,k}\|^2 + \eta_{y,0}^2 \sigma^2 B_t(\delta/\log(4T))}.$$

We conclude by applying a union bound over the two events (Equations (22) and (23)). $\qquad \square$

**Lemma C.2.** *With probability at least $1 - 2\delta$, for all $1 \le t \le T$,*

$$\sum_{k=1}^t \eta_{y,k-1}^2 \|\nabla_y g(x_k, y_k) - g_{y,k}\|^2 \le C_2 := 2\eta^2 \log\left(1 + \frac{\sigma^2 T}{2\gamma^2}\right) + \frac{7\eta^2\sigma^2}{\gamma^2} \log\frac{T}{\delta},$$

*where $\sigma = \sigma_y$ for Algorithm 1 and $\sigma = \sigma_{g,1}$ for Algorithm 2.*

*Proof of Lemma C.2.* The proof concludes by applying [2, Lemma 16] with $\eta_{s-1} = \eta_{y,k-1}$, $\nabla f(x_s) = \nabla_y g(x_k, y_k)$, $g_s = g_{y,k}$, $\sigma = \sigma$ or $\sigma = \sigma_{g,1}$, and $\eta_0 = \eta_{y,0} = \eta/\gamma$. $\qquad \square$

## C.1 Proof of Lemma 5.7

Before proving Lemma 5.7, let us define

$$D^2 := d_1^2 + \left(1 + \frac{\mu\eta}{\gamma}\right) C_1 + \frac{\eta L^2}{\mu^3}(\mu\eta + \alpha + \gamma)(1 + \log T) + \frac{\eta^2}{4} \tag{24}$$

$$+ \frac{4(1 + 2\mu\eta/\gamma)^2 C_2^2}{\eta^2} + 16 \left(1 + \frac{\mu\eta}{\gamma}\right)^2 \left(A_T(\delta)C_2 + \frac{\eta^2\sigma^2}{\gamma^2}B_T(\delta)\right),$$

where $A_t(\delta), B_t(\delta), C_1, C_2$ are defined in Lemmas A.1, A.4 and C.2, respectively

**Lemma 5.7.** *With probability at least* $1 - 4\delta$*, for all* $t \le T + 1$*,* $\bar{d}_t := \max_{k \le t} \|y_k - y_k^*\| \le D$*, and*

$$\sum_{k=1}^{t} \|y_k - y_k^*\|^2 \le \frac{1}{\mu^2\eta^2} \left(4D^2L^2 + 2D\gamma + 2\sqrt{2}D\sigma\sqrt{t}\right), \tag{8}$$

$$\sum_{k=1}^{t} \|y_k - y_k^*\| \le \frac{1}{\mu\eta} \left(\left(\sqrt{2}DL + \sqrt{D\gamma}\right)\sqrt{t} + \sqrt{2D\sigma}t^{3/4}\right), \tag{9}$$

*where* $D$ *is defined in Equation* (24)*, and* $\sigma = \sigma_y$ *for Algorithm 1 and* $\sigma = \sigma_{g,1}$ *for Algorithm 2.*

*Proof of Lemma 5.7.* Rolling a single step of SGD,

$$\|y_{k+1} - y_k^*\|^2 = \|y_k - y_k^*\|^2 - 2\eta_{y,k}\langle g_{y,k}, y_k - y_k^*\rangle + \eta_{y,k}^2\|g_{y,k}\|^2.$$

Since $f(x_k, \cdot)$ is $\mu$-strongly convex, then

$$-2\eta_{y,k}\langle g_{y,k}, y_k - y_k^*\rangle = -2\eta_{y,k}\langle\nabla_y g(x_k, y_k), y_k - y_k^*\rangle + 2\eta_{y,k}\langle\nabla_y g(x_k, y_k) - g_{y,k}, y_k - y_k^*\rangle$$

$$\le -2\mu\eta_{y,k}\|y_k - y_k^*\|^2 + 2\eta_{y,k}\langle\nabla_y g(x_k, y_k) - g_{y,k}, y_k - y_k^*\rangle.$$

Hence,

$$\|y_{k+1} - y_k^*\|^2 \le (1 - 2\mu\eta_{y,k})\|y_k - y_k^*\|^2 + 2\eta_{y,k}\langle\nabla_y g(x_k, y_k) - g_{y,k}, y_k - y_k^*\rangle + \eta_{y,k}^2\|g_{y,k}\|^2.$$

By Young's inequality and Lemma D.1,

$$\|y_{k+1} - y_{k+1}^*\|^2 \le (1 + \mu\eta_{y,k})\|y_k - y_k^*\|^2 + \left(1 + \frac{1}{\mu\eta_{y,k}}\right)\|y_{k+1}^* - y_k^*\|^2$$

$$\le (1 - \mu\eta_{y,k})\|y_k - y_k^*\|^2 + 2(1 + \mu\eta_{y,k})\eta_{y,k}\langle\nabla_y g(x_k, y_k) - g_{y,k}, y_k - y_k^*\rangle$$

$$+ (1 + \mu\eta_{y,k})\eta_{y,k}^2\|g_{y,k}\|^2 + \left(1 + \frac{1}{\mu\eta_{y,k}}\right)\frac{L^2\eta_{x,k}^2}{\mu^2}.$$

Summing from $k = 1$ to $t$, applying Lemma A.4, and using $\eta_{x,k} \le \eta/\sqrt{k}$,

$$\|y_{t+1} - y_{t+1}^*\|^2 \le \|y_1 - y_1^*\|^2 - \sum_{k=1}^{t}\mu\eta_{y,k}\|y_k - y_k^*\|^2 + \sum_{k=1}^{t}(1 + \mu\eta_{y,k})\eta_{y,k}^2\|g_{y,k}\|^2 + \frac{L^2\eta_{x,k}^2}{\mu^2}$$

$$+ 2\sum_{k=1}^{t}(\eta_{y,k} + \mu\eta_{y,k}^2)\langle\nabla_y g(x_k, y_k) - g_{y,k}, y_k - y_k^*\rangle + \frac{L^2}{\mu^2}\sum_{k=1}^{t}\frac{\eta_{x,k}^2}{\mu\eta_{y,k}}$$

$$\le \|y_1 - y_1^*\|^2 - \sum_{k=1}^{t}\mu\eta_{y,k}\|y_k - y_k^*\|^2 + (1 + \mu\eta_{y,0})C_1 + \frac{L^2\eta^2}{\mu^2}(1 + \log T)$$

$$+ \underbrace{2\sum_{k=1}^{t}(\eta_{y,k} + \mu\eta_{y,k}^2)\langle\nabla_y g(x_k, y_k) - g_{y,k}, y_k - y_k^*\rangle}_{(A)} + \underbrace{\frac{L^2}{\mu^2}\sum_{k=1}^{t}\frac{\eta_{x,k}^2}{\mu\eta_{y,k}}}_{(B)}. \tag{25}$$

**Bounding** $(A)$**.** In order to create a martingale we replace $\eta_{y,k} = \eta/\sqrt{G_{y,k-1}^2 + \|g_{y,k}\|^2}$ with $\hat{\eta}_{y,k} = \eta/\sqrt{G_{y,k-1}^2 + \|\nabla_y g(x_k, y_k)\|^2}$, then

$$(A_1) = \sum_{k=1}^{t}\eta_{y,k}\langle\nabla_y g(x_k, y_k) - g_{y,k}, y_k - y_k^*\rangle = \sum_{k=1}^{t}\hat{\eta}_{y,k}\langle\nabla_y g(x_k, y_k) - g_{y,k}, y_k - y_k^*\rangle \tag{26}$$

$$+ \sum_{k=1}^{t}(\eta_{y,k} - \hat{\eta}_{y,k})\langle \nabla_y g(x_k, y_k) - g_{y,k}, y_k - y_k^*\rangle.$$

Observe that

$$|\eta_{y,k} - \hat{\eta}_{y,k}| = \eta \frac{\left|\sqrt{G_{y,k-1}^2 + \|\nabla_y g(x_k, y_k)\|^2} - \sqrt{G_{y,k-1}^2 + \|g_{y,k}\|^2}\right|}{\sqrt{G_{y,k-1}^2 + \|g_{y,k}\|^2}\sqrt{G_{y,k-1}^2 + \|\nabla_y g(x_k, y_k)\|^2}}$$

$$\leq \eta \frac{\left|\|\nabla_y g(x_k, y_k)\|^2 - \|g_{y,k}\|^2\right|}{\sqrt{G_{y,k-1}^2 + \|g_{y,k}\|^2}\sqrt{G_{y,k-1}^2 + \|\nabla_y g(x_k, y_k)\|^2}\left(\sqrt{G_{y,k-1}^2 + \|g_{y,k}\|^2} + \sqrt{G_{y,k-1}^2 + \|\nabla_y g(x_k, y_k)\|^2}\right)}$$

$$\leq \eta \frac{\|\nabla_y g(x_k, y_k) - g_{y,k}\|(\|\nabla_y g(x_k, y_k)\| + \|g_{y,k}\|)}{\sqrt{G_{y,k-1}^2 + \|g_{y,k}\|^2}\sqrt{G_{y,k-1}^2 + \|\nabla_y g(x_k, y_k)\|^2}\left(\sqrt{G_{y,k-1}^2 + \|g_{y,k}\|^2} + \sqrt{G_{y,k-1}^2 + \|\nabla_y g(x_k, y_k)\|^2}\right)}$$

$$\leq \eta \frac{\|\nabla_y g(x_k, y_k) - g_{y,k}\|}{\sqrt{G_{y,k-1}^2 + \|g_{y,k}\|^2}\sqrt{G_{y,k-1}^2 + \|\nabla_y g(x_k, y_k)\|^2}}. \tag{27}$$

Thus,

$$\sum_{k=1}^{t}(\eta_{y,k} - \hat{\eta}_{y,k})\langle \nabla_y g(x_k, y_k) - g_{y,k}, y_k - y_k^*\rangle \leq \sum_{k=1}^{t}|\eta_{y,k} - \hat{\eta}_{y,k}|\|\nabla_y g(x_k, y_k) - g_{y,k}\|d_k$$

$$\leq \bar{d}_t \sum_{k=1}^{t}|\eta_{y,k} - \hat{\eta}_{y,k}|\|\nabla_y g(x_k, y_k) - g_{y,k}\|$$

$$\leq \eta \bar{d}_t \sum_{k=1}^{t} \frac{\|\nabla_y g(x_k, y_k) - g_{y,k}\|^2}{\sqrt{G_{y,k-1}^2 + \|g_{y,k}\|^2}\sqrt{G_{y,k-1}^2 + \|\nabla_y g(x_k, y_k)\|^2}}$$

$$\leq \frac{\bar{d}_t}{\eta} \sum_{k=1}^{t} \eta_{y,k-1}^2 \|\nabla_y g(x_k, y_k) - g_{y,k}\|^2. \tag{28}$$

Combing Equations (26) and (28) with Lemma C.1, with probability at least $1 - 2\delta$,

$$(A_1) \leq \frac{\bar{d}_t}{\eta} \sum_{k=1}^{t} \eta_{y,k-1}^2 \|\nabla_y g(x_k, y_k) - g_{y,k}\|^2 \tag{29}$$

$$+ 2\bar{d}_t' \sqrt{A_t(\delta/\log(4T)) \sum_{k\leq t} \eta_{y,k-1}^2 \|\nabla_y g(x_k, y_k) - g_{y,k}\|^2 + \eta_{y,0}^2 \sigma^2 B_t(\delta/\log(4T))}.$$

Similarly,

$$(A_2) = \sum_{k=1}^{t} \eta_{y,k}^2 \langle \nabla_y g(x_k, y_k) - g_{y,k}, y_k - y_k^*\rangle = \sum_{k=1}^{t} \hat{\eta}_{y,k}^2 \langle \nabla_y g(x_k, y_k) - g_{y,k}, y_k - y_k^*\rangle \tag{30}$$

$$+ \sum_{k=1}^{t}(\eta_{y,k}^2 - \hat{\eta}_{y,k}^2)\langle \nabla_y g(x_k, y_k) - g_{y,k}, y_k - y_k^*\rangle.$$

Using Equation (27),

$$\sum_{k=1}^{t}(\eta_{y,k}^2 - \hat{\eta}_{y,k}^2)\langle \nabla_y g(x_k, y_k) - g_{y,k}, y_k - y_k^*\rangle \leq \sum_{k=1}^{t}|\eta_{y,k} + \hat{\eta}_{y,k}||\eta_{y,k} - \hat{\eta}_{y,k}|\|\nabla_y g(x_k, y_k) - g_{y,k}\|d_k$$

$$\leq 2\eta_{y,0}\bar{d}_t \sum_{k=1}^{t}|\eta_{y,k} - \hat{\eta}_{y,k}|\|\nabla_y g(x_k, y_k) - g_{y,k}\|$$

$$\leq 2\eta_{y,0}\eta \bar{d}_t \sum_{k=1}^{t} \frac{\|\nabla_y g(x_k, y_k) - g_{y,k}\|^2}{\sqrt{G_{y,k-1}^2 + \|g_{y,k}\|^2}\sqrt{G_{y,k-1}^2 + \|\nabla_y g(x_k, y_k)\|^2}}$$

$$\leq \frac{2\eta_{y,0}\bar{d}_t}{\eta}\sum_{k=1}^{t}\eta_{y,k-1}^2\|\nabla_y g(x_k,y_k)-g_{y,k}\|^2. \tag{31}$$

Combing Equations (30) and (31) with Lemma C.1, with probability at least $1-2\delta$,

$$(A_2) \leq \frac{2\mu\eta_{y,0}\bar{d}_t}{\eta}\sum_{k=1}^{t}\eta_{y,k-1}^2\|\nabla_y g(x_k,y_k)-g_{y,k}\|^2 \tag{32}$$
$$+2\bar{d}_t'\eta_{y,0}\sqrt{A_t(\delta/\log(4T))\sum_{k\leq t}\eta_{y,k-1}^2\|\nabla_y g(x_k,y_k)-g_{y,k}\|^2 + \eta_{y,0}^2\sigma^2 B_t(\delta/\log(4T))}.$$

Hence, due to $(A) \leq 2(A_1)+2(A_2)$ and Equations (29) and (32), with probability at least $1-2\delta$,

$$(A) \leq \frac{2(1+2\mu\eta_{y,0})\bar{d}_t}{\eta}\sum_{k=1}^{t}\eta_{y,k-1}^2\|\nabla_y g(x_k,y_k)-g_{y,k}\|^2$$
$$+4(1+\mu\eta_{y,0})\bar{d}_t'\sqrt{A_t(\delta/\log(4T))\sum_{k\leq t}\eta_{y,k-1}^2\|\nabla_y g(x_k,y_k)-g_{y,k}\|^2 + \eta_{y,0}^2\sigma^2 B_t(\delta/\log(4T))}.$$

Using $ab \leq a^2/2 + b^2/2$,

$$(A) \leq \frac{\bar{d}_t^2}{4} + \frac{4(1+2\mu\eta_{y,0})^2}{\eta^2}\left(\sum_{k=1}^{t}\eta_{y,k-1}^2\|\nabla_y g(x_k,y_k)-g_{y,k}\|^2\right)^2 + \frac{\bar{d}_t'^2}{4}$$
$$+16(1+\mu\eta_{y,0})^2\left(A_t(\delta/\log(4T))\sum_{k\leq t}\eta_{y,k-1}^2\|\nabla_y g(x_k,y_k)-g_{y,k}\|^2 + \eta_{y,0}^2\sigma^2 B_t(\delta/\log(4T))\right)$$
$$\leq \frac{\bar{d}_t^2}{2} + \frac{4(1+2\mu\eta_{y,0})^2}{\eta^2}\left(\sum_{k=1}^{t}\eta_{y,k-1}^2\|\nabla_y g(x_k,y_k)-g_{y,k}\|^2\right)^2 + \frac{\eta^2}{4}$$
$$+16(1+\mu\eta_{y,0})^2\left(A_t(\delta/\log(4T))\sum_{k\leq t}\eta_{y,k-1}^2\|\nabla_y g(x_k,y_k)-g_{y,k}\|^2 + \eta_{y,0}^2\sigma^2 B_t(\delta/\log(4T))\right).$$

Under a union bound with Lemma C.2, with probability at least $1-4\delta$,

$$(A) \leq \frac{\bar{d}_t^2}{2} + \frac{4(1+2\mu\eta_{y,0})^2 C_2^2}{\eta^2} + \frac{\eta^2}{4} + 16(1+\mu\eta_{y,0})^2\left(A_t(\delta/\log(4T))C_2 + \eta_{y,0}^2\sigma^2 B_t(\delta/\log(4T))\right)$$
$$\leq \frac{\bar{d}_t^2}{2} + \frac{4(1+2\mu\eta_{y,0})^2 C_2^2}{\eta^2} + \frac{\eta^2}{4} + 16(1+\mu\eta_{y,0})^2\left(A_T(\delta)C_2 + \eta_{y,0}^2\sigma^2 B_T(\delta)\right).$$

**Bounding** $(B)$**.** By the definitions of $\eta_{x,t}$ and $\eta_{y,t}$,

$$\sum_{k=1}^{t}\frac{\eta_{x,k}^2}{\mu\eta_{y,k}} = \frac{\eta\alpha}{\mu}\sum_{k=1}^{t}\frac{\sqrt{\gamma^2+\sum_{s=1}^{k}\|g_{y,s}\|^2}}{k\sqrt{\alpha^2+\sum_{s=1}^{k}\|g_{x,s}-\tilde{g}_{x,s}\|^2+\|g_{y,s}\|^2}} \leq \frac{\eta\alpha}{\mu}\left(1+\frac{\gamma}{\alpha}\right)\sum_{k=1}^{t}\frac{1}{k} \tag{33}$$
$$\leq \frac{\eta(\alpha+\gamma)}{\mu}(1+\log T). \tag{34}$$

Then

$$(B) = \frac{L^2}{\mu^2}\sum_{k=1}^{t}\frac{\eta_{x,k}^2}{\mu\eta_{y,k}} \leq \frac{L^2\eta(\alpha+\gamma)}{\mu^3}(1+\log T).$$

Thus, returning to Equation (25) and using the definition of $D$, with probability at least $1-4\delta$,

$$d_{t+1}^2 \leq \frac{\bar{d}_t^2}{2} + d_1^2 + (1+\mu\eta_{y,0})C_1 + \frac{L^2\eta^2}{\mu^2}(1+\log T) + \frac{L^2\eta(\alpha+\gamma)}{\mu^3}(1+\log T)$$

$$+ \frac{4(1 + 2\mu\eta_{y,0})^2 C_2^2}{\eta^2} + \frac{\eta^2}{4} + 16(1 + \mu\eta_{y,0})^2 \left( A_T(\delta)C_2 + \eta_{y,0}^2 \sigma^2 B_T(\delta) \right)$$

$$\leq \frac{\bar{d}_t^2}{2} + \frac{D^2}{2}. \tag{35}$$

We use induction to show that with probability at least $1 - 4\delta$, $\bar{d}_t^2 \leq D^2$ for all $1 \leq t \leq T + 1$. Note that for $t = 1$, $\bar{d}_1^2 = d_1^2 \leq D^2$. Assume $\bar{d}_k \leq D^2$ for all $k \leq t \leq T$; then for $k = t + 1$, $d_{t+1}^2 \leq \bar{d}_t^2/2 + D^2/2 \leq D^2$ due to Equation (35). Thus, $\bar{d}_{t+1}^2 = \max\{d_{t+1}^2, \bar{d}_t^2\} \leq D^2$.

We proceed to prove Equations (8) and (9). Rearranging Equation (25), using Equation (35) and $\bar{d}_t^2 \leq D^2$, with probability at least $1 - 4\delta$,

$$\sum_{k=1}^{t} \mu\eta_{y,k} \|y_k - y_k^*\|^2 \leq \frac{\bar{d}_t^2}{2} + \frac{D^2}{2} \leq D^2. \tag{36}$$

Using $\eta_{y,k} \leq \eta_{y,t}$ for $k \leq t$,

$$\sum_{k=1}^{t} \|y_k - y_k^*\|^2 \leq \frac{D^2}{\mu\eta_{y,t}} = \frac{D\sqrt{\gamma^2 + \sum_{k=1}^{t} \|g_{y,k}\|^2}}{\mu\eta} \leq \frac{D\sqrt{\gamma^2 + 2\sigma^2 t + 2L^2 \sum_{k=1}^{t} \|y_k - y_k^*\|^2}}{\mu\eta}$$

$$\leq \frac{D\sqrt{\gamma^2 + 2\sigma^2 t}}{\mu\eta} + \frac{\sqrt{2}DL}{\mu\eta}\sqrt{\sum_{k=1}^{t} \|y_k - y_k^*\|^2},$$

where the second inequality uses Young's inequality and Assumption 3.1, and the last inequality is due to $\sqrt{a + b} \leq \sqrt{a} + \sqrt{b}$ for $a, b \geq 0$. Solving the inequality gives

$$\sqrt{\sum_{k=1}^{t} \|y_k - y_k^*\|^2} \leq \frac{1}{\mu\eta}\left( \sqrt{2}DL + \sqrt{D\left(\gamma + \sqrt{2}\sigma\sqrt{t}\right)} \right),$$

which implies that

$$\sum_{k=1}^{t} \|y_k - y_k^*\|^2 \leq \frac{1}{\mu^2\eta^2}\left( 4D^2L^2 + 2D\gamma + 2\sqrt{2}D\sigma\sqrt{t} \right),$$

and

$$\sum_{k=1}^{t} \|y_k - y_k^*\| \leq \sqrt{t \sum_{k=1}^{t} \|y_k - y_k^*\|^2} \leq \frac{\sqrt{t}}{\mu\eta}\left( \sqrt{2}DL + \sqrt{D\left(\gamma + \sqrt{2}\sigma\sqrt{t}\right)} \right)$$

$$\leq \frac{1}{\mu\eta}\left( \left(\sqrt{2}DL + \sqrt{D\gamma}\right)\sqrt{t} + \sqrt{2D\sigma}t^{3/4} \right).$$

$\square$

## D  Analysis of Algorithm 1

### D.1  Technical Lemmas

**Lemma D.1** ([50, Lemma 4.3]). *Under Assumption 3.1, $y^*(x)$ is $L/\mu$-Lipschitz and $\Phi(x)$ is $(\mu + L)L/\mu$-smooth.*

**Lemma D.2.** *Define $\hat{\epsilon}_t = m_t - \nabla\Phi(x_t)$, $\epsilon_t^B = \nabla_x f(x_t, y_t) - \nabla\Phi(x_t)$, and $\epsilon_t = g_{x,t} - \nabla_x f(x_t, y_t)$. Further, let $S_t = \nabla\Phi(x_{t-1}) - \nabla\Phi(x_t)$. For all $t \geq 1$, it holds that*

$$\hat{\epsilon}_t = \beta_{2:t}\hat{\epsilon}_1 + \sum_{k=2}^{t} \beta_{(k+1):t}\alpha_k\epsilon_k + \sum_{k=2}^{t} \beta_{(k+1):t}\alpha_k\epsilon_k^B + \sum_{k=2}^{t} \beta_{k:t}S_k.$$

*Proof of Lemma D.2.* The proof follows from a straightforward calculation:

$$
\begin{aligned}
\hat{\epsilon}_t &= m_t - \nabla\Phi(x_t) \\
&= \beta_t m_{t-1} + (1 - \beta_t)g_{x,t} - \nabla\Phi(x_t) \\
&= \beta_t(\hat{\epsilon}_{t-1} + \nabla\Phi(x_{t-1})) + (1 - \beta_t)(\epsilon_t + \epsilon_t^{\mathsf{B}} + \nabla\Phi(x_t)) - \nabla\Phi(x_t) \\
&= \beta_t\hat{\epsilon}_{t-1} + (1 - \beta_t)\epsilon_t + (1 - \beta_t)\epsilon_t^{\mathsf{B}} + \beta_t S_t.
\end{aligned}
$$

Unrolling the recursion and using $\alpha_t = 1 - \beta_t$ yields the result. $\qquad\square$

**Lemma D.3** (Descent Lemma). *Under Assumptions 3.1 and 3.2, define $\hat{\epsilon}_t := m_t - \nabla\Phi(x_t)$, then*

$$
\Phi(x_{t+1}) \le \Phi(x_t) - \eta_{x,t}\|\nabla\Phi(x_t)\| + 2\eta_{x,t}\|\hat{\epsilon}_t\| + \frac{(\mu + L)L\eta_{x,t}^2}{2\mu}.
$$

*Further, define $\Delta_1 := \Phi(x_1) - \Phi^*$, taking summation and rearranging we have*

$$
\sum_{t=1}^{T}\eta_{x,t}\|\nabla\Phi(x_t)\| \le \Delta_1 + 2\sum_{t=1}^{T}\eta_{x,t}\|\hat{\epsilon}_t\| + \frac{(\mu + L)L}{2\mu}\sum_{t=1}^{T}\eta_{x,t}^2.
$$

### D.2 Proof of Theorem 4.1

Before proving Theorem 4.1, let us define (recall the definition of $\kappa_\sigma$ and $t_0$ in Equation (6), here $\kappa_\sigma = \bar{\sigma}_x/\underline{\sigma}_x$ in minimax optimization)

$$
\Delta_1 = \Phi(x_1) - \Phi^*, \quad t_0 = \max\left\{2, 8\left(32\kappa_\sigma^4 - 30\kappa_\sigma^2 + 7\right)\log\left(\frac{60\log(6T)}{\delta}\right)\right\}, \tag{37}
$$

$$
\begin{aligned}
C_m &= \Delta_1 + 4\eta\bar{\sigma}_x\left(\sqrt{t_0 - 1} - 1 + e\sqrt{\kappa_\sigma}\left(1 + 2\sqrt{\bar{\sigma}_x/\alpha}\right)\right) + \frac{(\mu + L)L\eta^2}{2\mu}(1 + \log T) \\
&\quad + 2\eta\sqrt{1 + 32\log(2T/\delta)}\left(\left(t_0 - 1 + 2e\sqrt{t_0 - 2}\sqrt{\kappa_\sigma}\left(1 + 2\sqrt{\bar{\sigma}_x/\alpha}\right)\right)\bar{\sigma}_x + 3\sqrt{e}\kappa_\sigma^2\alpha\log T\right) \\
&\quad + \frac{4(\mu + L)L\eta^2}{\mu}\left(t_0 - 1 + e\sqrt{\kappa_\sigma}\left(1 + 2\sqrt{\bar{\sigma}_x/\alpha}\right) + e(\sqrt{\kappa_\sigma} + 2\kappa_\sigma)\log T\right) \\
&\quad + \left(LD\sqrt{\frac{\eta(\alpha + \gamma)}{\mu}(1 + \log T)}\right)\mathbb{I}(\bar{\sigma}_x = 0) \tag{38} \\
&\quad + \left(\frac{2\eta LD}{3}((t_0 - 1)^{3/2} - 1) + 2(t_0 - 2)LD\eta e\sqrt{\kappa_\sigma}\left(1 + 2\sqrt{\bar{\sigma}_x/\alpha}\right)\right. \\
&\quad + \left.\frac{2L\alpha}{\mu\underline{\sigma}_x}\left(1 + 2e\sqrt{\kappa_\sigma}\left(1 + 2\sqrt{\bar{\sigma}_x/\alpha}\right)\right)\left(2\left(\sqrt{2}DL + \sqrt{D\gamma}\right) + \sqrt{2D\sigma_y}(1 + \log T)\right)\right)\mathbb{I}(\underline{\sigma}_x > 0).
\end{aligned}
$$

**Theorem 4.1.** *Under Assumptions 3.1 and 3.2 and the parameter choices in Equations (3) and (4), let $\bar{\sigma}_x = \sigma_y$, then for any $\delta \in (0, 1/7)$, it holds with probability at least $1 - 7\delta$ that*

$$
\begin{aligned}
\frac{1}{T}\sum_{t=1}^{T}\|\nabla\Phi(x_t)\| \le \frac{C_m}{\eta\sqrt{\alpha}}&\left(\frac{1}{\sqrt{T}}\left(\alpha^2 + \frac{L^2}{\mu^2\eta^2}\left(4D^2L^2 + 2D\gamma\right)\right)^{1/4}\right. \\
&\quad + \left.\frac{1}{T^{3/8}}\left(\frac{2\sqrt{2}L^2D\bar{\sigma}_x}{\mu^2\eta^2}\right)^{1/4} + \frac{1}{T^{1/4}}\left(5\bar{\sigma}_x^2\right)^{1/4}\right),
\end{aligned}
$$

*where $C_m = \widetilde{O}(\kappa_\sigma^4)$ and $D$ are defined in Equations (24) and (38), respectively.*

*Proof of Theorem 4.1.* Without loss of generality, we assume $t_0$ is an integer (see definition in Equation (6)). By Lemmas D.2, D.3, G.2 and G.3, with probability at least $1 - 7\delta$,

$$
\sum_{t=1}^{T}\eta_{x,t}\|\nabla\Phi(x_t)\| \le \Delta_1 + 2\sum_{t=1}^{T}\eta_{x,t}\|\hat{\epsilon}_t\| + \frac{(\mu + L)L}{2\mu}\sum_{t=1}^{T}\eta_{x,t}^2
$$

$$\leq \Delta_1 + 2\sum_{t=1}^{T} \eta_{x,t} \left( \beta_{2:t}\|\hat{\epsilon}_1\| + \left\|\sum_{k=2}^{t} \beta_{(k+1):t}\alpha_k\epsilon_k\right\| + \left\|\sum_{k=2}^{t} \beta_{(k+1):t}\alpha_k\epsilon_k^{\text{B}}\right\| + \sum_{k=2}^{t} \beta_{k:t}\|S_k\| \right) + \frac{(\mu+L)L}{2\mu}\sum_{t=1}^{T}\eta_{x,t}^2$$

$$\leq \Delta_1 + 2\sum_{t=1}^{T} \eta_t \left( \beta_{2:t}\|\hat{\epsilon}_1\| + \left\|\sum_{k=2}^{t} \beta_{(k+1):t}\alpha_k\epsilon_k\right\| + \sum_{k=2}^{t} \beta_{k:t}\|S_k\| \right) + 2\sum_{t=1}^{T} \eta_{x,t}\left\|\sum_{k=2}^{t} \beta_{(k+1):t}\alpha_k\epsilon_k^{\text{B}}\right\| + \frac{(\mu+L)L}{2\mu}\sum_{t=1}^{T}\eta_{x,t}^2$$

$$\leq \Delta_1 + 4\eta\bar{\sigma}_x \left( \sqrt{t_0-1} - 1 + e\sqrt{\kappa_\sigma}\left(1 + 2\sqrt{\bar{\sigma}_x/\alpha}\right) \right) + \frac{(\mu+L)L\eta^2}{2\mu}(1 + \log T)$$

$$+ 2\eta\sqrt{1 + 32\log(2T/\delta)}\left( \left(t_0 - 1 + 2e\sqrt{t_0-2}\sqrt{\kappa_\sigma}\left(1 + 2\sqrt{\bar{\sigma}_x/\alpha}\right)\right)\bar{\sigma}_x + 3\sqrt{e}\kappa_\sigma^2\alpha\log T \right)$$

$$+ \frac{4(\mu+L)L\eta^2}{\mu}\left( t_0 - 1 + e\sqrt{\kappa_\sigma}\left(1 + 2\sqrt{\bar{\sigma}_x/\alpha}\right) + e(\sqrt{\kappa_\sigma} + 2\kappa_\sigma)\log T \right)$$

$$+ \left( LD\sqrt{\frac{\eta(\alpha+\gamma)}{\mu}(1 + \log T)} \right)\mathbb{I}(\bar{\sigma}_x = 0)$$

$$+ \left( \frac{2\eta LD}{3}((t_0-1)^{3/2} - 1) + 2(t_0-2)LD\eta e\sqrt{\kappa_\sigma}\left(1 + 2\sqrt{\bar{\sigma}_x/\alpha}\right) \right.$$

$$\left. + \frac{2L\alpha}{\mu\underline{\sigma}_x}\left(1 + 2e\sqrt{\kappa_\sigma}\left(1 + 2\sqrt{\bar{\sigma}_x/\alpha}\right)\right)\left(2\left(\sqrt{2}DL + \sqrt{D\gamma}\right) + \sqrt{2D\sigma_y}(1 + \log T)\right) \right)\mathbb{I}(\underline{\sigma}_x > 0)$$

$$= C_m.$$

Then, using $\eta_{x,t} \geq \eta_{x,T}$ for $t \leq T$,

$$\sum_{t=1}^{T} \eta_{x,T}\|\nabla\Phi(x_t)\| \leq \sum_{t=1}^{T} \eta_{x,t}\|\nabla\Phi(x_t)\| \leq C_m.$$

Therefore, by Lemma 5.7, with probability at least $1 - 7\delta$,

$$\frac{1}{T}\sum_{t=1}^{T}\|\nabla\Phi(x_t)\| \leq \frac{C_m}{T\eta_{x,T}} = \frac{C_m\sqrt{T}}{\eta\sqrt{\alpha}T}\left( \alpha^2 + \sum_{t=1}^{t}\|g_{x,t} - \tilde{g}_{x,t}\|^2 + \|g_{y,t}\|^2 \right)^{1/4}$$

$$\leq \frac{C_m}{\eta\sqrt{\alpha}\sqrt{T}}\left( \alpha^2 + 4\bar{\sigma}_x^2 T + \sigma_y^2 T + L^2\sum_{t=1}^{T}\|y_t - y_t^*\|^2 \right)^{1/4}$$

$$\leq \frac{C_m}{\eta\sqrt{\alpha}\sqrt{T}}\left( \alpha^2 + 4\bar{\sigma}_x^2 T + \sigma_y^2 T + \frac{L^2}{\mu^2\eta^2}\left(4D^2L^2 + 2D\gamma + 2\sqrt{2}D\sigma_y\sqrt{T}\right) \right)^{1/4}$$

$$\leq \frac{C_m}{\eta\sqrt{\alpha}}\left( \frac{1}{\sqrt{T}}\left(\alpha^2 + \frac{L^2}{\mu^2\eta^2}\left(4D^2L^2 + 2D\gamma\right)\right)^{1/4} + \frac{1}{T^{3/8}}\left(\frac{2\sqrt{2}L^2 D\sigma_y}{\mu^2\eta^2}\right)^{1/4} + \frac{1}{T^{1/4}}\left(4\bar{\sigma}_x^2 + \sigma_y^2\right)^{1/4} \right).$$

Setting $\bar{\sigma}_x = \sigma_y$ completes the proof. $\qquad\square$

## E   Analysis of Algorithm 2

### E.1   Neumann Series

For bilevel optimization problems with lower-level strong convexity, we estimate the hypergradient

$$\nabla\Phi(x) = \nabla_x f(x, y^*(x)) - \nabla_{xy}^2 g(x, y^*(x))[\nabla_{yy}^2 g(x, y^*(x))]^{-1}\nabla_y f(x, y^*(x))$$

via the Neumann series approach [22, 40, 34, 43]:

$$\bar{\nabla}f(x, y; \bar{\xi}) = \nabla_x F(x, y; \xi) - \nabla_{xy}^2 G(x, y; \zeta^{(0)})H_{yy}\nabla_y F(x, y; \xi), \tag{39}$$

where the matrix $H_{yy}$ is defined by

$$H_{yy} = \frac{1}{l_{g,1}}\sum_{n=0}^{N-1}\prod_{j=1}^{q}\left( I - \frac{\nabla_{yy}^2 G(x, y; \zeta^{(n,j)})}{l_{g,1}} \right), \tag{40}$$

and the set of random variables $\bar{\xi}$ is defined as

$$\bar{\xi} := \{\xi, \zeta^{(0)}, \bar{\zeta}^{(0)}, \dots, \bar{\zeta}^{(N-1)}\}, \quad \text{with} \quad \bar{\zeta}^{(n)} := \{\zeta^{(n,1)}, \dots, \zeta^{(n,n)}\} \quad \text{for} \ n \geq 0.$$

In addition, define the gradient approximation of $\Phi$ as

$$\bar{\nabla} f(x, y) = \nabla_x f(x, y) - \nabla_{xy}^2 g(x, y)[\nabla_{yy}^2 g(x, y)]^{-1} \nabla_y f(x, y). \tag{41}$$

## E.2 Technical Lemmas

**Lemma E.1** ([22, Lemma 2.2]). *Under Assumptions 3.3 and 3.4, we have*

$$\|\bar{\nabla} f(x, y) - \nabla\Phi(x)\| \leq L_f \|y - y^*(x)\|, \quad \|y^*(x_1) - y^*(x_2)\| \leq L_y \|x_1 - x_2\|,$$

$$\|\nabla\Phi(x_1) - \nabla\Phi(x_2)\| \leq L_F \|x_1 - x_2\|,$$

*where the constants $L_f, L_y, L_F$ are defined as*

$$L_f := l_{f,1} + \frac{l_{g,1} l_{f,1}}{\mu_g} + \frac{l_{f,0}}{\mu_g}\left(l_{g,2} + \frac{l_{g,1} l_{g,2}}{\mu_g}\right), \quad L_y = \frac{l_{g,1}}{\mu_g}, \tag{42}$$

$$L_F := l_{f,1} + \frac{l_{g,1}(l_{f,1} + L_f)}{\mu_g} + \frac{l_{f,0}}{\mu_g}\left(l_{g,2} + \frac{l_{g,1} l_{g,2}}{\mu_g}\right). \tag{43}$$

**Lemma E.2** ([22, Lemma 3.2], [34, Lemma 1]). *Under Assumptions 3.3 and 3.4, we have*

$$\|\mathbb{E}[H_{yy}]\| \leq \|H_{yy}\| \leq \frac{1}{\mu_g}, \quad \|[\nabla_{yy}^2 g(x, y)]^{-1} - \mathbb{E}[H_{yy}]\| \leq \frac{1}{\mu_g}\left(1 - \frac{\mu_g}{l_{g,1}}\right)^N,$$

$$\|\bar{\nabla} f(x, y) - \mathbb{E}[\bar{\nabla} f(x, y; \bar{\xi})]\| \leq \frac{l_{g,1} l_{f,0}}{\mu_g}\left(1 - \frac{\mu_g}{l_{g,1}}\right)^N.$$

**Lemma E.3.** *Under Assumptions 3.3 and 3.4, we have*

$$\|\bar{\nabla} f(x, y; \bar{\xi}) - \mathbb{E}[\bar{\nabla} f(x, y; \bar{\xi})]\| \leq \bar{\sigma}_\phi := \sigma_f + \frac{l_{f,0} \sigma_{g,2}}{\mu_g} + \frac{2 l_{g,1} l_{f,0}}{\mu_g}\mathbb{I}(\sigma_{g,2} \neq 0) + \frac{l_{g,1} \sigma_f}{\mu_g}. \tag{44}$$

*Proof of Lemma E.3.* By triangle inequality,

$\|\bar{\nabla} f(x, y; \bar{\xi}) - \mathbb{E}[\bar{\nabla} f(x, y; \bar{\xi})]\|$

$= \|(\nabla_x F(x, y; \xi) - \nabla_{xy}^2 G(x, y; \zeta^{(0)}) H_{yy} \nabla_y F(x, y; \xi)) - (\nabla_x f(x, y) - \nabla_{xy}^2 g(x, y) \mathbb{E}[H_{yy}] \nabla_y f(x, y))\|$

$\leq \|\nabla_x F(x, y; \xi) - \nabla_x f(x, y)\| + \|(\nabla_{xy}^2 G(x, y; \zeta^{(0)}) - \nabla_{xy}^2 g(x, y)) H_{yy} \nabla_y F(x, y; \xi)\|$

$\quad + \|\nabla_{xy}^2 g(x, y)(H_{yy} - \mathbb{E}[H_{yy}])\nabla_y F(x, y; \xi)\| + \|\nabla_{xy}^2 g(x, y)\mathbb{E}[H_{yy}](\nabla_y F(x, y; \xi) - \nabla_y f(x, y))\|.$

By Assumptions 3.3 and 3.4 and Lemma E.2, we have

$$\|\nabla_x F(x, y; \xi) - \nabla_x f(x, y)\| \leq \sigma_f,$$

and

$$\|(\nabla_{xy}^2 G(x, y; \zeta^{(0)}) - \nabla_{xy}^2 g(x, y)) H_{yy} \nabla_y F(x, y; \xi)\|$$
$$\leq \|\nabla_{xy}^2 G(x, y; \zeta^{(0)}) - \nabla_{xy}^2 g(x, y)\|\|H_{yy}\|\|\nabla_y F(x, y; \xi)\| \leq \frac{l_{f,0} \sigma_{g,2}}{\mu_g},$$

and

$$\|\nabla_{xy}^2 g(x, y)(H_{yy} - \mathbb{E}[H_{yy}])\nabla_y F(x, y; \xi)\|$$
$$\leq \|\nabla_{xy}^2 g(x, y)\|\|(H_{yy} - \mathbb{E}[H_{yy}])\|\|\nabla_y F(x, y; \xi)\| \leq \frac{2 l_{g,1} l_{f,0}}{\mu_g}\mathbb{I}(\sigma_{g,2} \neq 0),$$

and

$$\|\nabla_{xy}^2 g(x, y)\mathbb{E}[H_{yy}](\nabla_y F(x, y; \xi) - \nabla_y f(x, y))\| \leq \frac{l_{g,1} \sigma_f}{\mu_g}.$$

Hence, using the definition of $\bar{\sigma}_\phi$ as in Equation (44) we obtain

$$\|\bar{\nabla} f(x, y; \bar{\xi}) - \mathbb{E}[\bar{\nabla} f(x, y; \bar{\xi})]\| \leq \sigma_f + \frac{l_{f,0} \sigma_{g,2}}{\mu_g} + \frac{2 l_{g,1} l_{f,0}}{\mu_g}\mathbb{I}(\sigma_{g,2} \neq 0) + \frac{l_{g,1} \sigma_f}{\mu_g} = \bar{\sigma}_\phi.$$

$\square$

**Lemma E.4.** *Define* $\hat{\epsilon}_t = m_t - \nabla\Phi(x_t)$, $\epsilon_t^{\text{B}} = \bar{\nabla}f(x_t, y_t) - \nabla\Phi(x_t)$, $\epsilon_t^{\text{N}} = \mathbb{E}_{t-1}[g_{x,t}] - \bar{\nabla}f(x_t, y_t)$, *and* $\epsilon_t = g_{x,t} - \mathbb{E}_{t-1}[g_{x,t}]$. *Further, let* $S_t = \nabla\Phi(x_{t-1}) - \nabla\Phi(x_t)$. *For all* $t \geq 1$, *it holds that*

$$\hat{\epsilon}_t = \beta_{2:t}\hat{\epsilon}_1 + \sum_{k=2}^{t} \beta_{(k+1):t}\alpha_k\epsilon_k + \sum_{k=2}^{t} \beta_{(k+1):t}\alpha_k\epsilon_k^{\text{B}} + \sum_{k=2}^{t} \beta_{(k+1):t}\alpha_k\epsilon_k^{\text{N}} + \sum_{k=2}^{t} \beta_{k:t}S_k.$$

*Proof of Lemma E.4.* The proof follows from a straightforward calculation:

$$\begin{aligned}
\hat{\epsilon}_t &= m_t - \nabla\Phi(x_t) \\
&= \beta_t m_{t-1} + (1 - \beta_t)g_{x,t} - \nabla\Phi(x_t) \\
&= \beta_t(\hat{\epsilon}_{t-1} + \nabla\Phi(x_{t-1})) + (1 - \beta_t)(\epsilon_t + \epsilon_t^{\text{B}} + \epsilon_t^{\text{N}} + \nabla\Phi(x_t)) - \nabla\Phi(x_t) \\
&= \beta_t\hat{\epsilon}_{t-1} + (1 - \beta_t)\epsilon_t + (1 - \beta_t)\epsilon_t^{\text{B}} + (1 - \beta_t)\epsilon_t^{\text{N}} + \beta_t S_t.
\end{aligned}$$

Unrolling the recursion and using $\alpha_t = 1 - \beta_t$ yields the result. $\qquad\square$

**Lemma E.5** (Descent Lemma)**.** *Under Assumptions 3.3 and 3.4, define* $\hat{\epsilon}_t := m_t - \nabla\Phi(x_t)$, *then*

$$\Phi(x_{t+1}) \leq \Phi(x_t) - \eta_{x,t}\|\nabla\Phi(x_t)\| + 2\eta_{x,t}\|\hat{\epsilon}_t\| + \frac{L_F\eta_{x,t}^2}{2}.$$

*Further, define* $\Delta_1 := \Phi(x_1) - \Phi^*$, *taking summation and rearranging we have*

$$\sum_{t=1}^{T} \eta_{x,t}\|\nabla\Phi(x_t)\| \leq \Delta_1 + 2\sum_{t=1}^{T} \eta_{x,t}\|\hat{\epsilon}_t\| + \frac{L_F}{2}\sum_{t=1}^{T} \eta_{x,t}^2.$$

### E.3   Proof of Theorem 4.2

Before proving Theorem 4.2, let us define (recall the definition of $\kappa_\sigma$, $t_0$, and $\bar{\sigma}_\phi$ in Equations (6) and (44), here $\kappa_\sigma = \bar{\sigma}_\phi/\underline{\sigma}_\phi$ in bilevel optimization)

$$\Delta_1 = \Phi(x_1) - \Phi^*, \quad t_0 = \max\left\{2, 8\left(32\kappa_\sigma^4 - 30\kappa_\sigma^2 + 7\right)\log\left(\frac{60\log(6T)}{\delta}\right)\right\}, \tag{45}$$

$$\begin{aligned}
C_b &= \Delta_1 + 4\eta\bar{\sigma}_\phi\left(\sqrt{t_0 - 1} - 1 + e\sqrt{\kappa_\sigma}\left(1 + 2\sqrt{\bar{\sigma}_\phi/\alpha}\right)\right) + \frac{L_F\eta^2}{2}(1 + \log T) \\
&\quad + 2\eta\sqrt{1 + 32\log(2T/\delta)}\left(\left(t_0 - 1 + 2e\sqrt{t_0 - 2}\sqrt{\kappa_\sigma}\left(1 + 2\sqrt{\bar{\sigma}_\phi/\alpha}\right)\right)\bar{\sigma}_\phi + 3\sqrt{e}\kappa_\sigma^2\alpha\log T\right) \\
&\quad + 4L_F\eta^2\left(t_0 - 1 + e\sqrt{\kappa_\sigma}\left(1 + 2\sqrt{\bar{\sigma}_\phi/\alpha}\right) + e(\sqrt{\kappa_\sigma} + 2\kappa_\sigma)\log T\right) \\
&\quad + \frac{2\eta l_{g,1}l_{f,0}}{3\mu_g} + \left(L_f D\sqrt{\frac{\eta(\alpha + \gamma)}{\mu}}(1 + \log T)\right)\mathbb{I}(\bar{\sigma}_\phi = 0) \tag{46} \\
&\quad + \left(\frac{2\eta L_f D}{3}((t_0 - 1)^{3/2} - 1) + 2(t_0 - 2)L_f D\eta e\sqrt{\kappa_\sigma}\left(1 + 2\sqrt{\bar{\sigma}_\phi/\alpha}\right)\right. \\
&\quad \left. + \frac{2L_f\alpha}{\mu\underline{\sigma}_\phi}\left(1 + 2e\sqrt{\kappa_\sigma}\left(1 + 2\sqrt{\bar{\sigma}_\phi/\alpha}\right)\right)\left(2\left(\sqrt{2}Dl_{g,1} + \sqrt{D\gamma}\right) + \sqrt{2D\sigma_{g,1}}(1 + \log T)\right)\right)\mathbb{I}(\underline{\sigma}_\phi > 0).
\end{aligned}$$

**Theorem 4.2.** *Under Assumptions 3.3 and 3.4 and the parameter choices in Equations (3) and (4), for any* $\delta \in (0, 1/7)$, *choose* $N \geq \frac{3\log T}{2\log(1/(1 - \mu_g/l_{g,1}))}$, *it holds with probability at least* $1 - 7\delta$ *that*

$$\begin{aligned}
\frac{1}{T}\sum_{t=1}^{T}\|\nabla\Phi(x_t)\| &\leq \frac{C_b}{\eta\sqrt{\alpha}}\left(\frac{1}{\sqrt{T}}\left(\alpha^2 + \frac{l_{g,1}^2}{\mu^2\eta^2}\left(4D^2l_{g,1}^2 + 2D\gamma\right)\right)^{1/4}\right. \\
&\quad \left. + \frac{1}{T^{3/8}}\left(\frac{2\sqrt{2}l_{g,1}^2 D\sigma_{g,1}}{\mu^2\eta^2}\right)^{1/4} + \frac{1}{T^{1/4}}\left(4\bar{\sigma}_\phi^2 + \sigma_{g,1}^2\right)^{1/4}\right),
\end{aligned}$$

*where* $C_b = \widetilde{O}(\kappa_\sigma^4)$, $D$, *and* $\bar{\sigma}_\phi$ *are defined in Equations (24), (44) and (46), respectively.*

*Proof of Theorem 4.2.* Without loss of generality, we assume $t_0$ is an integer (see definition in Equation (6)). By Lemmas E.4, E.5, G.2 and G.3, with probability at least $1 - 7\delta$,

$$\sum_{t=1}^{T} \eta_{x,t} \|\nabla \Phi(x_t)\| \leq \Delta_1 + 2 \sum_{t=1}^{T} \eta_{x,t} \|\hat{\epsilon}_t\| + \frac{L_F}{2} \sum_{t=1}^{T} \eta_{x,t}^2$$

$$\leq \Delta_1 + 2 \sum_{t=1}^{T} \eta_{x,t} \left( \beta_{2:t} \|\hat{\epsilon}_1\| + \left\| \sum_{k=2}^{t} \beta_{(k+1):t} \alpha_k \epsilon_k \right\| + \left\| \sum_{k=2}^{t} \beta_{(k+1):t} \alpha_k \epsilon_k^{\mathtt{B}} \right\| \right.$$

$$\left. + \left\| \sum_{k=2}^{t} \beta_{(k+1):t} \alpha_k \epsilon_k^{\mathtt{N}} \right\| + \sum_{k=2}^{t} \beta_{k:t} \|S_k\| \right) + \frac{L_F}{2} \sum_{t=1}^{T} \eta_{x,t}^2$$

$$\leq \Delta_1 + 4\eta \bar{\sigma}_\phi \left( \sqrt{t_0 - 1} - 1 + e\sqrt{\kappa_\sigma} \left( 1 + 2\sqrt{\bar{\sigma}_\phi/\alpha} \right) \right) + \frac{L_F \eta^2}{2} (1 + \log T)$$

$$+ 2\eta \sqrt{1 + 32 \log(2T/\delta)} \left( \left( t_0 - 1 + 2e\sqrt{t_0 - 2}\sqrt{\kappa_\sigma} \left( 1 + 2\sqrt{\bar{\sigma}_\phi/\alpha} \right) \right) \bar{\sigma}_\phi + 3\sqrt{e}\kappa_\sigma^2 \alpha \log T \right)$$

$$+ 4L_F \eta^2 \left( t_0 - 1 + e\sqrt{\kappa_\sigma} \left( 1 + 2\sqrt{\bar{\sigma}_\phi/\alpha} \right) + e(\sqrt{\kappa_\sigma} + 2\kappa_\sigma) \log T \right)$$

$$+ \frac{2\eta T^{3/2} l_{g,1} l_{f,0}}{3\mu_g} \left( 1 - \frac{\mu_g}{l_{g,1}} \right)^N + \left( L_f D \sqrt{\frac{\eta(\alpha + \gamma)}{\mu}} (1 + \log T) \right) \mathbb{I}(\bar{\sigma}_\phi = 0)$$

$$+ \left( \frac{2\eta L_f D}{3} ((t_0 - 1)^{3/2} - 1) + 2(t_0 - 2) L_f D \eta e \sqrt{\kappa_\sigma} \left( 1 + 2\sqrt{\bar{\sigma}_\phi/\alpha} \right) \right.$$

$$\left. + \frac{2L_f \alpha}{\mu \sigma_\phi} \left( 1 + 2e\sqrt{\kappa_\sigma} \left( 1 + 2\sqrt{\bar{\sigma}_\phi/\alpha} \right) \right) \left( 2 \left( \sqrt{2}D l_{g,1} + \sqrt{D\gamma} \right) + \sqrt{2D\sigma_{g,1}}(1 + \log T) \right) \right) \mathbb{I}(\sigma_\phi > 0)$$

$$\leq C_b.$$

Then, using $\eta_{x,t} \geq \eta_{x,T}$ for $t \leq T$,

$$\sum_{t=1}^{T} \eta_{x,T} \|\nabla \Phi(x_t)\| \leq \sum_{t=1}^{T} \eta_{x,t} \|\nabla \Phi(x_t)\| \leq C_b.$$

Therefore, by Lemma 5.7, with probability at least $1 - 7\delta$,

$$\frac{1}{T} \sum_{t=1}^{T} \|\nabla \Phi(x_t)\| \leq \frac{C_b}{T \eta_{x,T}} = \frac{C_b \sqrt{T}}{\eta \sqrt{\alpha} T} \left( \alpha^2 + \sum_{t=1}^{t} \|g_{x,t} - \tilde{g}_{x,t}\|^2 + \|g_{y,t}\|^2 \right)^{1/4}$$

$$\leq \frac{C_b}{\eta \sqrt{\alpha} \sqrt{T}} \left( \alpha^2 + 4\bar{\sigma}_\phi^2 T + \sigma_{g,1}^2 T + l_{g,1}^2 \sum_{t=1}^{T} \|y_t - y_t^*\|^2 \right)^{1/4}$$

$$\leq \frac{C_b}{\eta \sqrt{\alpha} \sqrt{T}} \left( \alpha^2 + 4\bar{\sigma}_\phi^2 T + \sigma_{g,1}^2 T + \frac{l_{g,1}^2}{\mu^2 \eta^2} \left( 4D^2 l_{g,1}^2 + 2D\gamma + 2\sqrt{2}D\sigma_{g,1} \sqrt{T} \right) \right)^{1/4}$$

$$\leq \frac{C_b}{\eta \sqrt{\alpha}} \left( \frac{1}{\sqrt{T}} \left( \alpha^2 + \frac{l_{g,1}^2}{\mu^2 \eta^2} \left( 4D^2 l_{g,1}^2 + 2D\gamma \right) \right)^{1/4} + \frac{1}{T^{3/8}} \left( \frac{2\sqrt{2} l_{g,1}^2 D\sigma_{g,1}}{\mu^2 \eta^2} \right)^{1/4} + \frac{1}{T^{1/4}} \left( 4\bar{\sigma}_\phi^2 + \sigma_{g,1}^2 \right)^{1/4} \right).$$

$$\square$$

# F   Linear Programming Basics

**Definition F.1** (General Form of Linear Programming [3, Section 1.1])**.** The linear programming problem can be written as

$$\min_{x \in \mathbb{R}^n} c^\top x \tag{47}$$
$$\text{s.t.,} \quad Ax \geq b.$$

**Definition F.2** ([3, Definition 2.1]). A **polyhedron** is a set that can be described in the form $\{x \in \mathbb{R}^n \mid Ax \geq b\}$, where $A \in \mathbb{R}^{m \times n}$ is a matrix and $b \in \mathbb{R}^n$ is a vector.

**Definition F.3** ([3, Definition 2.6]). Let $P$ be a polyhedron. A vector $x \in P$ is an **extreme point** of $P$ if we cannot find two vectors $y, z \in P$, both different from $x$, a scalar $\lambda \in [0, 1]$, such that $x = \lambda y + (1 - \lambda)z$.

**Theorem F.4** ([3, Theorem 2.8]). *Consider the linear programming problem of minimizing $c^\top x$ over a polyhedron $P$. Suppose that $P$ has at least one extreme point. Then, either the optimal cost is equal to $-\infty$, or there exists an extreme point which is optimal.*

**Lemma F.5.** *Assume $0 \leq \underline{\alpha}_t \leq \alpha_t \leq \bar{\alpha}_t$ and $0 \leq \underline{\beta}_t \leq \beta_t \leq \bar{\beta}_t$. Further, let $\epsilon_i \in \mathbb{R}^d$ and denote $\gamma_{k,t} := \beta_{(k+1):t}\alpha_k$, $\underline{\gamma}_{k,t} := \underline{\beta}_{(k+1):t}\underline{\alpha}_k$, and $\bar{\gamma}_{k,t} := \bar{\beta}_{(k+1):t}\bar{\alpha}_k$. There exists a set $\{b_{ij,t}^*\}$ with each $b_{ij,t}^*$ satisfying either $b_{ij,t}^* = \underline{\gamma}_{i,t}\underline{\gamma}_{j,t}$ or $b_{ij,t}^* = \bar{\gamma}_{i,t}\bar{\gamma}_{j,t}$ for every pair $(i, j)$, such that*

$$\sum_{i=3}^{t}\sum_{j=2}^{i-1}\gamma_{i,t}\gamma_{j,t}\langle\epsilon_i, \epsilon_j\rangle \leq \sum_{i=3}^{t}\sum_{j=2}^{i-1}b_{ij,t}^*\langle\epsilon_i, \epsilon_j\rangle.$$

*Proof of Lemma F.5.* Consider the following constrained optimization problem:

$$\max_{\gamma_t} \sum_{i=3}^{t}\sum_{j=2}^{i-1}\gamma_{i,t}\gamma_{j,t}\langle\epsilon_i, \epsilon_j\rangle \tag{48}$$
$$\text{s.t.,} \quad \underline{\alpha}_i \leq \alpha_i \leq \bar{\alpha}_i, \quad \underline{\beta}_i \leq \beta_i \leq \bar{\beta}_i, \quad \forall i \leq t.$$

A relaxed version of problem (48) is:

$$\max_{\gamma_t} \sum_{i=3}^{t}\sum_{j=2}^{i-1}\gamma_{i,t}\gamma_{j,t}\langle\epsilon_i, \epsilon_j\rangle \tag{49}$$
$$\text{s.t.,} \quad \underline{\gamma}_{i,t} \leq \gamma_{i,t} \leq \bar{\gamma}_{i,t}, \quad \forall i \leq t.$$

Moreover, problem (49) is equivalent to:

$$\min_{\gamma_t} \sum_{i=3}^{t}\sum_{j=2}^{i-1}\gamma_{i,t}\gamma_{j,t}(-\langle\epsilon_i, \epsilon_j\rangle) \tag{50}$$
$$\text{s.t.,} \quad \underline{\gamma}_{i,t} \leq \gamma_{i,t} \leq \bar{\gamma}_{i,t}, \quad \forall i \leq t.$$

Now we proceed to verify that

(a) A relaxed version of Equation (50), namely Equation (53), is a linear programming problem of minimizing $c_t^\top x_t$ over a polyhedron $P_t$ for some $c_t, x_t, P_t$;

(b) $P_t$ has at least one extreme point;

(c) The optimal cost of Equation (50) is not equal to $-\infty$.

Fact (a). We first define a few notations. Define $c_{ij}, x_{ij,t}, \underline{b}_{ij,t}, \bar{b}_{ij,t}$, and the index set $\mathcal{I}_t$ as

$$c_{ij} = -\langle\epsilon_i, \epsilon_j\rangle, \quad x_{ij,t} = \gamma_{i,t}\gamma_{j,t}, \quad \underline{b}_{ij,t} = \underline{\gamma}_{i,t}\underline{\gamma}_{j,t}, \quad \bar{b}_{ij,t} = \bar{\gamma}_{i,t}\bar{\gamma}_{j,t}, \quad \mathcal{I}_t = \{(i, j) \mid 3 \leq i \leq t, 2 \leq j < i\}.$$

Let $c_t, x, P_t$ be defined as

$$c_t = (c_{ij})_{(i,j)\in\mathcal{I}_t} = \begin{bmatrix} c_{32} \\ \vdots \\ c_{t(t-1)} \end{bmatrix}, \quad x_t = (x_{ij,t})_{(i,j)\in\mathcal{I}_t} = \begin{bmatrix} x_{32,t} \\ \vdots \\ x_{t(t-1),t} \end{bmatrix}, \quad P_t = \{x_t \mid A_t x_t \geq b_t\},$$
$$\tag{51}$$

where $A_t, b_t$ are defined as

$$\underline{b}_t = (\underline{b}_{ij,t})_{(i,j)\in\mathcal{I}_t}, \quad \bar{b}_t = (\bar{b}_{ij,t})_{(i,j)\in\mathcal{I}_t}, \quad A_t = \begin{bmatrix} 1 & \cdots & 0 \\ \vdots & \ddots & \vdots \\ 0 & \cdots & 1 \\ -1 & \cdots & 0 \\ \vdots & \ddots & \vdots \\ 0 & \cdots & -1 \end{bmatrix}, \quad b_t = \begin{bmatrix} \underline{b}_t \\ -\bar{b}_t \end{bmatrix} = \begin{bmatrix} \underline{b}_{32,t} \\ \vdots \\ \underline{b}_{t(t-1),t} \\ -\bar{b}_{32,t} \\ \vdots \\ -\bar{b}_{t(t-1),t} \end{bmatrix}.$$
(52)

According to Definition F.1, the optimization problem in Equation (50) can be relaxed into the following linear programming formulation (with a potentially higher objective value):

$$\min_{x_t} c_t^\top x_t$$
$$\text{s.t.,} \quad A_t x_t \geq b_t.$$
(53)

Fact (b). We will show that the set of extreme points of $P_t$ is

$$\mathcal{S}_t = \left\{ [b^*_{32,t} \ \cdots \ b^*_{t(t-1),t}]^\top \mid b^*_{ij,t} = \underline{b}_{ij,t} \text{ or } b^*_{ij,t} = \bar{b}_{ij,t} \text{ for } 2 \leq j < i \leq t \right\}.$$

( $\implies$ ) Let $x_t = [b^*_{32,t} \ \cdots \ b^*_{t(t-1),t}]^\top \in \mathcal{S}$. Check that $A_t x_t \geq b_t$, thus $x_t \in P_t$. Assume there exists $y, z \in P_t$ (both different from $x_t$) and a scalar $\lambda \in (0,1)$, such that $x_t = \lambda y + (1-\lambda)z$. Note that at least one element of $y$ differs from the corresponding element in $x_t$, denote this element by $y_{ij}$, where $(i,j) \in \mathcal{I}_t$. We consider the following two cases:

- If $y_{ij} > x_{ij,t} = \underline{b}_{ij,t}$, then

$$z_{ij} = \frac{x_{ij,t} - \lambda y_{ij}}{1 - \lambda} < x_{ij,t} = \underline{b}_{ij,t}.$$

  This implies that $z \notin P_t$ since $A_t z \not\geq b_t$.

- If $y_{ij} < x_{ij,t} = \bar{b}_{ij,t}$, then

$$z_{ij} = \frac{x_{ij,t} - \lambda y_{ij}}{1 - \lambda} > x_{ij,t} = \bar{b}_{ij,t}.$$

  This implies that $z \notin P_t$ since $A_t z \not\geq b_t$.

Therefore, $z \notin P_t$ in both cases. By Definition F.3, $x_t$ is an extreme point of $P_t$.

( $\impliedby$ ) Assume there exists some $x_t \in P_t$ such that $x_t \notin \mathcal{S}_t$. Then there must be at least one element of $x_t$, denoted by $x_{ij,t}$, satisfying $x_{ij,t} \neq \underline{b}_{ij,t}$ and $x_{ij,t} \neq \bar{b}_{ij,t}$. Let $y, z$, and $x_t$ differ only in the $ij$-th element, and define $y_{ij}, z_{ij}$ as

$$y_{ij} = x_{ij,t} - \min\left\{ x_{ij,t} - \underline{b}_{ij,t}, \bar{b}_{ij,t} - x_{ij,t} \right\} \quad \text{and} \quad y_{ij} = x_{ij,t} + \min\left\{ x_{ij,t} - \underline{b}_{ij,t}, \bar{b}_{ij,t} - x_{ij,t} \right\}.$$

Then $y, z \in P_t$ since $A_t y \geq b_t$ and $A_t z \geq b_t$. Note that $x_t = (y+z)/2$, hence by Definition F.3, $x_t$ is not an extreme point of $P_t$.

Fact (c). If $t$ is finite, then

$$\left| -\sum_{i=3}^{t} \sum_{j=2}^{i-1} \gamma_{i,t} \gamma_{j,t} \langle \epsilon_i, \epsilon_j \rangle \right| \leq \sum_{i=3}^{t} \sum_{j=2}^{i-1} \gamma_{i,t} \gamma_{j,t} |\langle \epsilon_i, \epsilon_j \rangle| \leq \sum_{i=3}^{t} \sum_{j=2}^{i-1} \bar{\gamma}_{i,t} \bar{\gamma}_{j,t} |\langle \epsilon_i, \epsilon_j \rangle| < \infty.$$

Hence,

$$-\sum_{i=3}^{t} \sum_{j=2}^{i-1} \gamma_{i,t} \gamma_{j,t} \langle \epsilon_i, \epsilon_j \rangle > -\infty.$$

Combining Fact (a), Fact (b), Fact (c), and using Theorem F.4, we know that there exists an extreme point $x_t^* \in \mathcal{S}$ such that

$$\sum_{i=3}^{t} \sum_{j=2}^{i-1} b^*_{ij,t} (-\langle \epsilon_i, \epsilon_j \rangle) = c_t^\top x_t^* \leq c_t^\top x_t = \sum_{i=3}^{t} \sum_{j=2}^{i-1} \gamma_{i,t} \gamma_{j,t} (-\langle \epsilon_i, \epsilon_j \rangle).$$

Therefore, for problem (53),

$$\sum_{i=3}^{t}\sum_{j=2}^{i-1}\gamma_{i,t}\gamma_{j,t}\langle\epsilon_i,\epsilon_j\rangle \le \sum_{i=3}^{t}\sum_{j=2}^{i-1}b_{ij,t}^*\langle\epsilon_i,\epsilon_j\rangle.$$

We conclude the proof by noting that problem (53) is a relaxed version of problem (48). □

## G   Useful Algebraic Facts

**Lemma G.1.** *Let $p \in (0,1]$ and $q \in (0,1)$. Further, let $a,b \in \mathbb{N}_{\ge 2}$ with $a \le b$, and $c,c_1,c_2 > 0$.*

*(a) We have*

$$\prod_{t=a}^{b}(1-(1+ct)^{-q}) \le \exp\left(\frac{(1+ac)^{1-q}-(1+bc)^{1-q}}{c(1-q)}\right).$$

*(b) If $p \ge q$ and $c_1 \le c_2$, then*

$$\sum_{t=a}^{b}(1+c_1t)^{-q/2}t^{-p}\prod_{k=a}^{t}(1-(1+c_2k)^{-q})$$

$$\le \left(\frac{c_2}{c_1}\right)^{q/2}(a-1)^{-p}(1+(a-1)c_2)^{q/2}\exp\left(\frac{(1+ac_2)^{1-q}-(1+(a-1)c_2)^{1-q}}{c_2(1-q)}\right).$$

*(c) If $c_1 \le c_2$ and $(p-q)(1+(a-1)c_2)^{q-1} \le 1/2$, then*

$$\sum_{t=a}^{b}(1+c_1t)^{-p}\prod_{k=t+1}^{b}(1-(1+c_2k)^{-q}) \le 2\left(\frac{c_2}{c_1}\right)^{p}(1+(b+1)c_2)^{q-p}\exp\left(\frac{(1+c_2)^{1-q}-1}{c_2(1-q)}\right).$$

*Proof of Lemma G.1.* We prove the results individually.

**Lemma G.1(a).** Using $1-x \le \exp(-x)$ and the monotonicity of $(1+ct)^{-q}$,

$$\prod_{t=a}^{b}(1-(1+ct)^{-q}) \le \exp\left(-\sum_{t=a}^{b}(1+ct)^{-q}\right) \le \exp\left(-\int_{a}^{b+1}(1+ct)^{-q}dt\right)$$

$$= \exp\left(\frac{1}{c(1-q)}\left((1+ac)^{1-q}-(1+(b+1)c)^{1-q}\right)\right)$$

$$\le \exp\left(\frac{1}{c(1-q)}\left((1+ac)^{1-q}-(1+bc)^{1-q}\right)\right).$$

**Lemma G.1(b).** By Lemma G.1(a),

$$\sum_{t=a}^{b}(1+c_1t)^{-q/2}t^{-p}\prod_{k=a}^{t}(1-(1+c_2k)^{-q})$$

$$\le \exp\left(\frac{(1+ac_2)^{1-q}}{c_2(1-q)}\right)\sum_{t=a}^{b}(1+c_1t)^{-q/2}t^{-p}\exp\left(-\frac{(1+c_2t)^{1-q}}{c_2(1-q)}\right).$$

Using the monotonicity of $(1+c_1t)^{-q/2}t^{-p}\exp\left(-\frac{(1+c_2t)^{1-q}}{c_2(1-q)}\right)$ and $c_1 \le c_2$,

$$\sum_{t=a}^{b}(1+c_1t)^{-q/2}t^{-p}\exp\left(-\frac{(1+c_2t)^{1-q}}{c_2(1-q)}\right) \le \int_{a-1}^{b}(1+c_1t)^{-q/2}t^{-p}\exp\left(-\frac{(1+c_2t)^{1-q}}{c_2(1-q)}\right)dt$$

$$\leq \int_{a-1}^{b} \frac{(1+c_2t)^{q/2}}{(1+c_1t)^{q/2}}(1+c_2t)^{-q/2}t^{-p}\exp\left(-\frac{(1+c_2t)^{1-q}}{c_2(1-q)}\right)dt$$

$$\leq \frac{(1+bc_2)^{q/2}}{(1+bc_1)^{q/2}}\int_{a-1}^{b}(1+c_2t)^{-q/2}t^{-p}\exp\left(-\frac{(1+c_2t)^{1-q}}{c_2(1-q)}\right)dt$$

$$\leq \left(\frac{c_2}{c_1}\right)^{q/2}\underbrace{\int_{a-1}^{b}(1+c_2t)^{-q/2}t^{-p}\exp\left(-\frac{(1+c_2t)^{1-q}}{c_2(1-q)}\right)dt}_{(I)}.$$

We continue to bound term $(I)$. By partial integration and $p \geq q$,

$$I = (a-1)^{-p}(1+(a-1)c_2)^{q/2}\exp\left(-\frac{(1+(a-1)c_2)^{1-q}}{c_2(1-q)}\right) - b^{-p}(1+bc_2)^{q/2}\exp\left(-\frac{(1+bc_2)^{1-q}}{c_2(1-q)}\right)$$

$$+ \int_{a-1}^{b}\left(\left(-\frac{p}{t}-pc_2+\frac{qc_2}{2}\right)(1+c_2t)^{q-1}\right)(1+c_2t)^{-q/2}t^{-p}\exp\left(-\frac{(1+c_2t)^{1-q}}{c_2(1-q)}\right)dt$$

$$\leq (a-1)^{-p}(1+(a-1)c_2)^{q/2}\exp\left(-\frac{(1+(a-1)c_2)^{1-q}}{c_2(1-q)}\right) - b^{-p}(1+bc_2)^{q/2}\exp\left(-\frac{(1+bc_2)^{1-q}}{c_2(1-q)}\right)$$

$$- \left(\frac{p}{b}+pc_2-\frac{qc_2}{2}\right)(1+bc_2)^{q-1}\int_{a-1}^{b}(1+c_2t)^{-q/2}t^{-p}\exp\left(-\frac{(1+c_2t)^{1-q}}{c_2(1-q)}\right)dt.$$

Rearranging it yields

$$I \leq \frac{(a-1)^{-p}(1+(a-1)c_2)^{q/2}\exp\left(-\frac{(1+(a-1)c_2)^{1-q}}{c_2(1-q)}\right) - b^{-p}(1+bc_2)^{q/2}\exp\left(-\frac{(1+bc_2)^{1-q}}{c_2(1-q)}\right)}{1+\left(\frac{p}{b}+pc_2-\frac{qc_2}{2}\right)(1+bc_2)^{q-1}}$$

$$\leq (a-1)^{-p}(1+(a-1)c)^{q}\exp\left(-\frac{(1+(a-1)c)^{1-q}}{c(1-q)}\right).$$

Thus, we obtain

$$\sum_{t=a}^{b}(1+c_1t)^{-q/2}t^{-p}\prod_{k=a}^{t}(1-(1+c_2k)^{-q}) \leq \exp\left(\frac{(1+ac_2)^{1-q}}{c_2(1-q)}\right)\left(\frac{c_2}{c_1}\right)^{q/2}I$$

$$\leq \left(\frac{c_2}{c_1}\right)^{q/2}(a-1)^{-p}(1+(a-1)c_2)^{q/2}\exp\left(\frac{(1+ac_2)^{1-q}-(1+(a-1)c_2)^{1-q}}{c_2(1-q)}\right).$$

**Lemma G.1(c).** Using $1-x \leq \exp(-x)$,

$$\sum_{t=a}^{b}(1+c_1t)^{-p}\prod_{k=t+1}^{b}(1-(1+c_2k)^{-q}) \leq \exp\left(-\sum_{k=1}^{b}(1+c_2k)^{-q}\right)\sum_{t=a}^{b}(1+c_1t)^{-p}\exp\left(\sum_{k=1}^{t}(1+c_2k)^{-q}\right).$$

Using the monotonicity of $(1+c_2k)^{-q}$, we have

$$\exp\left(-\sum_{k=1}^{b}(1+c_2k)^{-q}\right) \leq \exp\left(-\int_{1}^{b+1}(1+c_2k)^{-q}dk\right) = \exp\left(\frac{(1+c_2)^{1-q}-(1+(b+1)c_2)^{1-q}}{c_2(1-q)}\right)$$

and

$$\exp\left(\sum_{k=1}^{t}(1+c_2k)^{-q}\right) \leq \exp\left(\int_{0}^{t}(1+c_2k)^{-q}dk\right) \leq \exp\left(\frac{(1+c_2t)^{1-q}-1}{c_2(1-q)}\right).$$

Due to $c_1 \leq c_2$ and the monotonicity of $\left(\frac{1+c_2t}{1+c_1t}\right)^{p}$, we continue to bound

$$\sum_{t=a}^{b}(1+c_1t)^{-p}\exp\left(\sum_{k=1}^{t}(1+c_2k)^{-q}\right) \leq \sum_{t=a}^{b}(1+c_1t)^{-p}\exp\left(\frac{(1+c_2t)^{1-q}-1}{c_2(1-q)}\right)$$

$$= \sum_{t=a}^{b} \frac{(1+c_2 t)^p}{(1+c_1 t)^p} (1+c_2 t)^{-p} \exp\left(\frac{(1+c_2 t)^{1-q}-1}{c_2(1-q)}\right)$$

$$\leq \frac{(1+bc_2)^p}{(1+bc_1)^p} \sum_{t=a}^{b} (1+c_2 t)^{-p} \exp\left(\frac{(1+c_2 t)^{1-q}-1}{c_2(1-q)}\right)$$

$$\leq \left(\frac{c_2}{c_1}\right)^p \sum_{t=a}^{b} (1+c_2 t)^{-p} \exp\left(\frac{(1+c_2 t)^{1-q}-1}{c_2(1-q)}\right).$$

Denote $h(t)$ as

$$h(t) := (1+c_2 t)^{-p} \exp\left(\frac{(1+c_2 t)^{1-q}-1}{c_2(1-q)}\right).$$

By simple calculation,

$$h'(t) = \left(-pc_2 + (1+c_2 t)^{1-q}\right)(1+c_2 t)^{-p-1} \exp\left(\frac{(1+c_2 t)^{1-q}-1}{c_2(1-q)}\right).$$

Define $t_1$ as

$$t_1 := \frac{(pc_2)^{\frac{1}{1-q}}-1}{c_2}.$$

Note that $h(t)$ is monotonically decreasing for $t \leq t_1$ and monotonically increasing for $t \geq t_1$.

If $t_1 \leq a$, then

$$\sum_{t=a}^{b} (1+c_2 t)^{-p} \exp\left(\frac{(1+c_2 t)^{1-q}-1}{c_2(1-q)}\right) \leq \int_a^{b+1} (1+c_2 t)^{-p} \exp\left(\frac{(1+c_2 t)^{1-q}-1}{c_2(1-q)}\right) dt.$$

If $a \leq t_1 \leq b$, then

$$\sum_{t=a}^{b} (1+c_2 t)^{-p} \exp\left(\frac{(1+c_2 t)^{1-q}-1}{c_2(1-q)}\right) \leq \left(\sum_{t=a}^{\lfloor t_1 \rfloor} + \sum_{t=\lceil t_1 \rceil}^{b}\right)(1+c_2 t)^{-p} \exp\left(\frac{(1+c_2 t)^{1-q}-1}{c_2(1-q)}\right)$$

$$\leq \left(\int_{a-1}^{\lfloor t_1 \rfloor} + \int_{\lceil t_1 \rceil}^{b+1}\right)(1+c_2 t)^{-p} \exp\left(\frac{(1+c_2 t)^{1-q}-1}{c_2(1-q)}\right) dt$$

$$\leq \int_{a-1}^{b+1} (1+c_2 t)^{-p} \exp\left(\frac{(1+c_2 t)^{1-q}-1}{c_2(1-q)}\right) dt.$$

If $t_1 \geq b$, then

$$\sum_{t=a}^{b} (1+c_2 t)^{-p} \exp\left(\frac{(1+c_2 t)^{1-q}-1}{c_2(1-q)}\right) \leq \int_{a-1}^{b} (1+c_2 t)^{-p} \exp\left(\frac{(1+c_2 t)^{1-q}-1}{c_2(1-q)}\right) dt.$$

Therefore, based on the three cases above,

$$\sum_{t=a}^{b} (1+c_2 t)^{-p} \exp\left(\frac{(1+c_2 t)^{1-q}-1}{c_2(1-q)}\right) \leq \int_{a-1}^{b+1} (1+c_2 t)^{-p} \exp\left(\frac{(1+c_2 t)^{1-q}-1}{c_2(1-q)}\right) dt =: I'.$$

We proceed to upper bound the integral $I'$. By partial integration and $(p-q)(1+(a-1)c_2)^{q-1} \leq 1/2$,

$$I' = \int_{a-1}^{b+1} (1+c_2 t)^{q-p}(1+c_2 t)^{-q} \exp\left(\frac{(1+c_2 t)^{1-q}-1}{c_2(1-q)}\right) dt$$

$$= (1+(b+1)c_2)^{q-p} \exp\left(\frac{(1+(b+1)c_2)^{1-q}-1}{c_2(1-q)}\right) - (1+(a-1)c_2)^{q-p} \exp\left(\frac{(1+(a-1)c_2)^{1-q}-1}{c_2(1-q)}\right)$$

$$+ (p-q)\int_{a-1}^{b+1} (1+c_2 t)^{q-p-1} \exp\left(\frac{(1+c_2 t)^{1-q}-1}{c_2(1-q)}\right) dt$$

$$\leq (1+(b+1)c_2)^{q-p} \exp\left(\frac{(1+(b+1)c_2)^{1-q}-1}{c_2(1-q)}\right) - (1+(a-1)c_2)^{q-p} \exp\left(\frac{(1+(a-1)c_2)^{1-q}-1}{c_2(1-q)}\right)$$

$$+ (p-q)(1+(a-1)c_2)^{q-1} \int_{a-1}^{b+1} (1+c_2 t)^{-p} \exp\left(\frac{(1+c_2 t)^{1-q}-1}{c_2(1-q)}\right) dt$$

$$\leq (1+(b+1)c_2)^{q-p} \exp\left(\frac{(1+(b+1)c_2)^{1-q}-1}{c_2(1-q)}\right) - (1+(a-1)c_2)^{q-p} \exp\left(\frac{(1+(a-1)c_2)^{1-q}-1}{c_2(1-q)}\right)$$

$$+ \frac{1}{2} \int_{a-1}^{b+1} (1+c_2 t)^{-p} \exp\left(\frac{(1+c_2 t)^{1-q}-1}{c_2(1-q)}\right) dt.$$

Rearranging it yields

$$I' \leq 2(1+(b+1)c_2)^{q-p} \exp\left(\frac{(1+(b+1)c_2)^{1-q}-1}{c_2(1-q)}\right).$$

Therefore,

$$\sum_{t=a}^{b} (1+c_1 t)^{-p} \prod_{k=t+1}^{b} \left(1-(1+c_2 k)^{-q}\right)$$

$$\leq \exp\left(\frac{(1+c_2)^{1-q}-(1+(b+1)c_2)^{1-q}}{c_2(1-q)}\right) \left(\frac{c_2}{c_1}\right)^p \sum_{t=a}^{b} (1+c_2 t)^{-p} \exp\left(\frac{(1+c_2 t)^{1-q}-1}{c_2(1-q)}\right)$$

$$\leq \exp\left(\frac{(1+c_2)^{1-q}-(1+(b+1)c_2)^{1-q}}{c_2(1-q)}\right) \left(\frac{c_2}{c_1}\right)^p I'$$

$$\leq 2\left(\frac{c_2}{c_1}\right)^p (1+(b+1)c_2)^{q-p} \exp\left(\frac{(1+c_2)^{1-q}-1}{c_2(1-q)}\right).$$

$\square$

**Lemma G.2.** *For all $t \geq 1$, let $\alpha_t$, $\beta_t$, $\eta_t$, and $\kappa_\sigma$ be defined as in Equations (5) and (6):*

$$\alpha_t = \frac{\alpha}{\sqrt{\alpha^2 + \sum_{k=1}^{t} \|g_k - \tilde{g}_k\|^2}}, \quad \beta_t = 1 - \alpha_t, \quad \eta_t = \frac{\eta\sqrt{\alpha_t}}{\sqrt{t}}, \quad \text{and} \quad \kappa_\sigma = \begin{cases} \bar{\sigma}/\underline{\sigma} & \underline{\sigma} > 0 \\ 1 & \bar{\sigma} = 0 \end{cases}.$$

*Then with probability at least $1-\delta$, we have*

*(a) For $a \geq t_0$, $\sum_{t=a}^{T} \eta_t \beta_{a:t} \leq 2\eta e \sqrt{\kappa_\sigma}\left((a-1)^{-1/2} + 2\sqrt{\bar{\sigma}/\alpha}(a-1)^{-1/4}\right)$.*

*(b) $\sum_{t=1}^{T} \eta_t \beta_{2:t} \leq 2\eta\left(\sqrt{t_0 - 1} - 1 + e\sqrt{\kappa_\sigma}\left(1 + 2\sqrt{\bar{\sigma}/\alpha}\right)\right)$.*

*(c) $\sum_{t=1}^{T} \eta_t \sqrt{\sum_{k=2}^{t} \bar{\beta}_{(k+1):t}^2 \bar{\alpha}_k^2}$*
  *$\leq \eta\left(t_0 - 1 + 2e\sqrt{t_0 - 2}\sqrt{\kappa_\sigma}\left(1 + 2\sqrt{\bar{\sigma}/\alpha}\right) + 3\sqrt{e}\kappa_\sigma \alpha \underline{\sigma}^{-1} \log T\right) \mathbb{I}(\underline{\sigma} > 0).$*

*(d) $\sum_{t=1}^{T} \eta_t \sum_{k=2}^{t} \eta_{k-1} \beta_{k:t} \leq 2\eta^2\left((t_0 - 1)\left(1 + e\sqrt{\kappa_\sigma}\left(1 + 2\sqrt{\bar{\sigma}/\alpha}\right)\right) + e(\sqrt{\kappa_\sigma} + 2\kappa_\sigma)\log T\right).$*

*(e) $\sum_{t=1}^{T} \eta_t^2 \leq \eta^2(1 + \log T)$.*

*Proof of Lemma G.2.* We prove the results individually. Without loss of generality, we assume $t_0$ is an integer. By Lemma 5.5, with probability at least $1-\delta$, we have $\underline{\alpha}_t \leq \alpha_t \leq \bar{\alpha}_t$ and $\underline{\beta}_t \leq \beta_t \leq \bar{\beta}_t$.

**Lemma G.2(a).** Consider the case where $0 < \underline{\sigma} \leq \bar{\sigma}$. Apply Lemma G.1(b) with $a = a \geq t_0$, $b = T$, $p = q = 1/2$, $c_1 = \underline{\sigma}^2/\alpha^2$, and $c_2 = 4\bar{\sigma}^2/\alpha^2$,

$$\sum_{t=a}^{T} \eta_t \beta_{a:t} \leq 2\eta\sqrt{\kappa_\sigma}(a-1)^{-1/2}\left(1 + \frac{4\bar{\sigma}^2}{\alpha^2}(a-1)\right)^{1/4} \exp\left(\frac{\sqrt{1+4a\bar{\sigma}^2/\alpha^2} - \sqrt{1+4(a-1)\bar{\sigma}^2/\alpha^2}}{2\bar{\sigma}^2/\alpha^2}\right)$$

$$\leq 2\eta\sqrt{\kappa_\sigma}(a-1)^{-1/2}\left(1+2\sqrt{\bar\sigma/\alpha}(a-1)^{1/4}\right)\exp(1)$$

$$= 2\eta e\sqrt{\kappa_\sigma}\left((a-1)^{-1/2}+2\sqrt{\bar\sigma/\alpha}(a-1)^{-1/4}\right)\leq 2\eta e\sqrt{\kappa_\sigma}\left(1+2\sqrt{\bar\sigma/\alpha}\right),$$

where the second inequality uses $(1+x)^{1/4}\leq 1+x^{1/4}$, and the last inequality is due to $a\geq t_0\geq 2$. The bound also holds for the case where $\underline\sigma=\bar\sigma=0$.

**Lemma G.2(b).** Apply Lemma G.2(a) with $a=t_0$,

$$\sum_{t=t_0}^{T}\eta_t\beta_{t_0:t}\leq 2\eta e\sqrt{\kappa_\sigma}\left(1+2\sqrt{\bar\sigma/\alpha}\right)(t_0-1)^{-1/4}\leq 2\eta e\sqrt{\kappa_\sigma}\left(1+2\sqrt{\bar\sigma/\alpha}\right).$$

Hence, using $\eta_t\leq\eta/\sqrt{t}$ and $\beta_t\leq 1$,

$$\sum_{t=1}^{T}\eta_t\beta_{2:t}=\sum_{t=1}^{t_0-1}\eta_t\beta_{2:t}+\sum_{t=t_0}^{T}\eta_t\beta_{2:t}\leq 2\eta(\sqrt{t_0-1}-1)+\sum_{t=t_0}^{T}\eta_t\beta_{t_0:t}$$

$$\leq 2\eta\left(\sqrt{t_0-1}-1+e\sqrt{\kappa_\sigma}\left(1+2\sqrt{\bar\sigma/\alpha}\right)\right).$$

**Lemma G.2(c).** Consider the case where $0<\underline\sigma\leq\bar\sigma$. Apply Lemma G.1(c) with $a=t_0$, $b=t$, $p=1$, $q=1/2$, $c_1=\underline\sigma^2/\alpha^2$, and $c_2=4\bar\sigma^2/\alpha^2$,

$$\sqrt{\sum_{k=t_0}^{t}\bar\alpha_k^2\bar\beta_{(k+1):t}^2}\leq\sqrt{\frac{12\bar\sigma^2}{\underline\sigma^2}\left(1+\frac{4\bar\sigma^2}{\alpha^2}(t+1)\right)^{-1/2}\exp\left(\frac{\sqrt{1+4\bar\sigma^2/\alpha^2}-1}{2\bar\sigma^2/\alpha^2}\right)}$$

$$\leq\sqrt{12e\kappa_\sigma^2\left(1+\frac{4\bar\sigma^2}{\alpha^2}(t+1)\right)^{-1/2}}=\sqrt{12e}\kappa_\sigma\left(1+\frac{4\bar\sigma^2}{\alpha^2}(t+1)\right)^{-1/4}.$$

Using the definition of $\eta_t$,

$$\sum_{t=t_0}^{T}\eta_t\sqrt{\sum_{k=t_0}^{t}\bar\beta_{(k+1):t}^2\bar\alpha_k^2}\leq\eta\sqrt{12e}\kappa_\sigma\sum_{t=t_0}^{T}\left(1+\frac{\underline\sigma^2}{\alpha^2}t\right)^{-1/4}t^{-1/2}\left(1+\frac{4\bar\sigma^2}{\alpha^2}(t+1)\right)^{-1/4}$$

$$\leq 3\eta\sqrt{e}\kappa_\sigma\sum_{t=t_0}^{T}\frac{\alpha}{\sqrt{\underline\sigma\bar\sigma}}t^{-1}\leq 3\eta\sqrt{e}\kappa_\sigma\alpha\underline\sigma^{-1}\log T.$$

Thus,

$$\sum_{t=1}^{T}\eta_t\sqrt{\sum_{k=2}^{t}\bar\beta_{(k+1):t}^2\bar\alpha_k^2}=\sum_{t=1}^{t_0-1}\eta_t\sqrt{\sum_{k=2}^{t}\bar\beta_{(k+1):t}^2\bar\alpha_k^2}+\sum_{t=t_0}^{T}\eta_t\sqrt{\sum_{k=2}^{t}\bar\beta_{(k+1):t}^2\bar\alpha_k^2}$$

$$\leq\sum_{t=1}^{t_0-1}\frac{\eta\sqrt{t-1}}{\sqrt{t}}+\sum_{t=t_0}^{T}\eta_t\sqrt{\sum_{k=2}^{t_0-1}\bar\beta_{(k+1):t}^2\bar\alpha_k^2+\sum_{k=t_0}^{t}\bar\beta_{(k+1):t}^2\bar\alpha_k^2}$$

$$\leq\eta(t_0-1)+\sqrt{t_0-2}\sum_{t=t_0}^{T}\eta_t\beta_{t_0:t}+\sum_{t=t_0}^{T}\eta_t\sqrt{\sum_{k=t_0}^{t}\bar\beta_{(k+1):t}^2\bar\alpha_k^2}$$

$$\leq\eta(t_0-1)+2\eta e\sqrt{t_0-2}\sqrt{\kappa_\sigma}\left(1+2\sqrt{\bar\sigma/\alpha}\right)+3\eta\sqrt{e}\kappa_\sigma\alpha\underline\sigma^{-1}\log T$$

$$=\eta\left(t_0-1+2e\sqrt{t_0-2}\sqrt{\kappa_\sigma}\left(1+2\sqrt{\bar\sigma/\alpha}\right)+3\sqrt{e}\kappa_\sigma\alpha\underline\sigma^{-1}\log T\right),$$

where the first inequality uses $\eta_t\leq\eta/\sqrt{t}$, the second inequality is due to $\sqrt{a+b}\leq\sqrt{a}+\sqrt{b}$ for $a,b\geq 0$, and the third inequality uses Lemma G.2(a).

For the case $\underline\sigma=\bar\sigma=0$, we have $\alpha_t=1$ and $\beta_t=0$, hence $\sum_{t=1}^{T}\eta_t\sqrt{\sum_{k=2}^{t}\bar\beta_{(k+1):t}^2\bar\alpha_k^2}=0$.

**Lemma G.2(d).** We have

$$\sum_{t=1}^{T} \eta_t \sum_{k=2}^{t} \eta_{k-1}\beta_{k:t} = \sum_{t=1}^{t_0-1} \eta_t \sum_{k=2}^{t} \eta_{k-1}\beta_{k:t} + \sum_{t=t_0}^{T} \eta_t \sum_{k=2}^{t} \eta_{k-1}\beta_{k:t}$$

$$\leq \sum_{t=1}^{t_0-1} \frac{\eta}{\sqrt{t}} \sum_{k=1}^{t-1} \frac{\eta}{\sqrt{k}} + \sum_{t=t_0}^{T} \eta_t \sum_{k=2}^{t_0} \eta_{k-1}\beta_{k:t} + \sum_{t=t_0}^{T} \eta_t \sum_{k=t_0+1}^{t} \eta_{k-1}\beta_{k:t}$$

$$\leq 2\eta^2(t_0-1) + \eta(t_0-1)\sum_{t=t_0}^{T} \eta_t \beta_{t_0:t} + \sum_{t=t_0}^{T} \eta_t \sum_{k=t_0+1}^{t} \eta_{k-1}\beta_{k:t}$$

$$\leq 2\eta^2(t_0-1) + 2\eta^2 e(t_0-1)\sqrt{\kappa_\sigma}\left(1+2\sqrt{\bar\sigma/\alpha}\right) + \sum_{t=t_0}^{T} \eta_t \sum_{k=t_0+1}^{t} \eta_{k-1}\beta_{k:t},$$

where the first inequality uses $\eta_t \leq \eta/\sqrt{t}$, the second inequality is due to $\eta_t \leq \eta$, and the last inequality uses Lemma G.2(a). We continue to bound the last term above:

$$\sum_{t=t_0}^{T} \eta_t \sum_{k=t_0+1}^{t} \eta_{k-1}\beta_{k:t} = \sum_{t=t_0}^{T} \sum_{k=t_0+1}^{t} \eta_t \eta_{k-1}\beta_{k:t} = \sum_{k=t_0+1}^{T} \eta_{k-1} \sum_{t=k}^{T} \eta_t \beta_{k:t}$$

$$\leq 2\eta e\sqrt{\kappa_\sigma} \sum_{k=t_0+1}^{T} \eta_{k-1}\left((k-1)^{-1/2} + 2\sqrt{\bar\sigma/\alpha}(k-1)^{-1/4}\right)$$

$$\leq 2\eta e\sqrt{\kappa_\sigma} \sum_{k=t_0}^{T-1} \eta k^{-1} + 2\eta\sqrt{\kappa_\sigma}k^{-1}$$

$$\leq 2\eta^2 e\sqrt{\kappa_\sigma}(1+2\sqrt{\kappa_\sigma})\log T,$$

where the first inequality uses Lemma G.2(a), and the second inequality is due to $\eta_t = \eta\sqrt{\alpha_t}/\sqrt{t} \leq \eta/\sqrt{t}$. Therefore,

$$\sum_{t=1}^{T} \eta_t \sum_{k=2}^{t} \eta_{k-1}\beta_{k:t} \leq 2\eta^2\left(t_0-1+e(t_0-1)\sqrt{\kappa_\sigma}\left(1+2\sqrt{\bar\sigma/\alpha}\right)+e(\sqrt{\kappa_\sigma}+2\kappa_\sigma)\log T\right).$$

**Lemma G.2(e).** By the definition of $\eta_t$ and the fact that $\alpha_t \leq 1$,

$$\sum_{t=1}^{T} \eta_t^2 = \sum_{t=1}^{T} \frac{\eta^2\alpha_t}{t} \leq \sum_{t=1}^{T} \frac{\eta^2}{t} \leq \eta^2(1+\log T).$$

$\square$

**Lemma G.3.** *For all $t \geq 1$, let $\alpha_t$, $\beta_t$, and $\eta_{x,t}$ be defined as in Equations (3) and (4) (see Sections 4.2 and 4.3), and let $\epsilon_t^B$ be defined as in Lemma D.2 for minimax optimization and in Lemma E.4 for bilevel optimization:*

$$\alpha_t = \frac{\alpha}{\sqrt{\alpha^2 + \sum_{k=1}^{t}\|g_{x,k}-\tilde g_{x,k}\|^2}}, \quad \beta_t = 1-\alpha_t, \quad \epsilon_t^B = \begin{cases} \nabla_x f(x_t,y_t) - \nabla\Phi(x_t) & \text{(minimax)} \\ \bar\nabla f(x_t,y_t) - \nabla\Phi(x_t) & \text{(bilevel)} \end{cases}$$

$$\alpha_t' = \frac{\alpha}{\sqrt{\alpha^2 + \sum_{k=1}^{t}\|g_{x,k}-\tilde g_{x,k}\|^2 + \|g_{y,k}\|^2}}, \quad \text{and} \quad \eta_{x,t} = \frac{\eta\sqrt{\alpha_t'}}{\sqrt{t}}.$$

*Then with probability at least $1-4\delta$, we have*

$$\sum_{t=1}^{T} \eta_{x,t} \left\|\sum_{k=2}^{t} \beta_{(k+1):t}\alpha_k\epsilon_k^B\right\| \leq \left(LD\sqrt{\frac{\eta(\alpha+\gamma)}{\mu}(1+\log T)}\right)\mathbb{I}(\bar\sigma_u = 0)$$

$$+ \left( \frac{2\eta LD}{3}((t_0 - 1)^{3/2} - 1) + 2(t_0 - 2)LD\eta e\sqrt{\kappa_\sigma}\left(1 + 2\sqrt{\bar{\sigma}_u/\alpha}\right)\right.$$

$$\left. + \frac{2L\alpha}{\mu\underline{\sigma}_u}\left(1 + 2e\sqrt{\kappa_\sigma}\left(1 + 2\sqrt{\bar{\sigma}_u/\alpha}\right)\right)\left(2\left(\sqrt{2}DL + \sqrt{D\gamma}\right) + \sqrt{2D\sigma_l}(1 + \log T)\right)\right)\mathbb{I}(\underline{\sigma}_u > 0),$$

*where (with a slight abuse of notation for L) $L = L, \bar{\sigma}_u = \bar{\sigma}_x, \underline{\sigma}_u = \underline{\sigma}_x, \sigma_l = \sigma_y$ for Algorithm 1, and $L = l_{g,1}, \bar{\sigma}_u = \bar{\sigma}_\phi, \underline{\sigma}_u = \underline{\sigma}_\phi, \sigma_l = \sigma_{g,1}$ for Algorithm 2.*

*Proof of Lemma G.3.* We consider the cases $\underline{\sigma}_u = \bar{\sigma}_u = 0$ and $0 < \underline{\sigma}_u \leq \bar{\sigma}_u$ separately.

**Case $\underline{\sigma}_u = \bar{\sigma}_u = 0$.** In this case,

$$\alpha_t = 1, \quad \beta_t = 0, \quad \alpha_t' = \frac{\alpha}{\sqrt{\alpha^2 + \sum_{k=1}^t \|g_{y,k}\|^2}} \quad \text{and} \quad \eta_t = \frac{\eta\sqrt{\alpha_t'}}{\sqrt{t}}.$$

By Assumption 3.1, $\|\epsilon_t^{\text{B}}\| = \|\nabla_x f(x_t, y_t) - \nabla\Phi(x_t)\| \leq L\|y_t - y_t^*\|$. Thus,

$$\sum_{t=1}^T \eta_{x,t}\left\|\sum_{k=2}^t \beta_{(k+1):t}\alpha_k\epsilon_k^{\text{B}}\right\| = \sum_{t=1}^T \eta_{x,t}\|\alpha_t\epsilon_t^{\text{B}}\| \leq L\sum_{t=1}^T \eta_{x,t}\|y_t - y_t^*\|.$$

Using Cauchy–Schwarz inequality and Equations (33) and (36), with probability at least $1 - 4\delta$,

$$\sum_{t=1}^T \eta_{x,t}\|y_t - y_t^*\| \leq \sqrt{\sum_{t=1}^T \frac{\eta_{x,t}^2}{\mu\eta_{y,t}}}\sqrt{\sum_{t=1}^T \mu\eta_{y,t}\|y_t - y_t^*\|^2} \leq D\sqrt{\frac{\eta(\alpha + \gamma)}{\mu}}(1 + \log T).$$

Hence,

$$\sum_{t=1}^T \eta_{x,t}\left\|\sum_{k=2}^t \beta_{(k+1):t}\alpha_k\epsilon_k^{\text{B}}\right\| \leq LD\sqrt{\frac{\eta(\alpha + \gamma)}{\mu}}(1 + \log T). \tag{54}$$

**Case $0 < \underline{\sigma}_u \leq \bar{\sigma}_u$.** By triangle inequality,

$$\sum_{t=1}^T \eta_{x,t}\left\|\sum_{k=2}^t \beta_{(k+1):t}\alpha_k\epsilon_k^{\text{B}}\right\| \leq \sum_{t=1}^T \eta_{x,t}\sum_{k=2}^t \beta_{(k+1):t}\alpha_k\|\epsilon_k^{\text{B}}\| \leq L\sum_{t=1}^T \eta_{x,t}\sum_{k=2}^t \beta_{(k+1):t}\alpha_k\|y_k - y_k^*\|.$$

Then with probability at least $1 - 4\delta$,

$$\sum_{t=1}^T \eta_{x,t}\sum_{k=2}^t \beta_{(k+1):t}\alpha_k\|y_k - y_k^*\| = \sum_{t=1}^{t_0-1} \eta_{x,t}\sum_{k=2}^t \beta_{(k+1):t}\alpha_k\|y_k - y_k^*\| + \sum_{t=t_0}^T \eta_{x,t}\sum_{k=2}^t \beta_{(k+1):t}\alpha_k\|y_k - y_k^*\|$$

$$\leq D\sum_{t=1}^{t_0-1} t\eta_{x,t} + \sum_{t=t_0}^T \eta_{x,t}\sum_{k=2}^{t_0-1} \beta_{(k+1):t}\alpha_k\|y_k - y_k^*\| + \sum_{t=t_0}^T \eta_{x,t}\sum_{k=t_0}^t \beta_{(k+1):t}\alpha_k\|y_k - y_k^*\|$$

$$\leq \frac{2\eta D}{3}((t_0 - 1)^{3/2} - 1) + (t_0 - 2)D\sum_{t=t_0}^T \eta_{x,t}\beta_{t_0:t} + \sum_{t=t_0}^T \eta_{x,t}\sum_{k=t_0}^t \beta_{(k+1):t}\alpha_k\|y_k - y_k^*\|$$

$$\leq \frac{2\eta D}{3}((t_0 - 1)^{3/2} - 1) + 2(t_0 - 2)D\eta e\sqrt{\kappa_\sigma}\left(1 + 2\sqrt{\bar{\sigma}_u/\alpha}\right) + \sum_{t=t_0}^T \eta_{x,t}\sum_{k=t_0}^t \beta_{(k+1):t}\alpha_k\|y_k - y_k^*\|.$$

Swapping the order of summation for the last term, and applying Lemma G.2(a),

$$\sum_{t=t_0}^T \eta_{x,t}\sum_{k=t_0}^t \beta_{(k+1):t}\alpha_k\|y_k - y_k^*\| = \sum_{k=t_0}^T \alpha_k\|y_k - y_k^*\|\sum_{t=k}^T \eta_{x,t}\beta_{(k+1):t}$$

$$= \sum_{k=t_0}^T \alpha_k\|y_k - y_k^*\|\left(\eta_{x,k} + \sum_{t=k+1}^T \eta_{x,t}\beta_{(k+1):t}\right)$$

$$\leq \sum_{k=t_0}^{T} \alpha_k \|y_k - y_k^*\| \left( \frac{\eta\sqrt{\alpha_k}}{\sqrt{k}} + 2\eta e\sqrt{\kappa_\sigma}\left(1 + 2\sqrt{\bar{\sigma}_u/\alpha}\right) k^{-1/4} \right)$$

$$\leq \frac{\eta\alpha}{\underline{\sigma}_u}\left(1 + 2e\sqrt{\kappa_\sigma}\left(1 + 2\sqrt{\bar{\sigma}_u/\alpha}\right)\right) \sum_{k=t_0}^{T} k^{-3/4}\|y_k - y_k^*\|.$$

Using summation by parts $\sum_{t=1}^{T} a_t(b_t - b_{t-1}) = a_T b_T - a_1 b_0 - \sum_{t=1}^{T-1}(a_{t+1} - a_t)b_t$ with $a_t = t^{-3/4}$ and $b_t = \sum_{k=1}^{t} \|y_k - y_k^*\|$,

$$\sum_{k=t_0}^{T} k^{-3/4}\|y_k - y_k^*\| \leq T^{-3/4} \sum_{k=1}^{T} \|y_k - y_k^*\| + \sum_{t=1}^{T-1}(t^{-3/4} - (t+1)^{-3/4}) \sum_{k=1}^{t} \|y_k - y_k^*\|$$

$$\leq T^{-3/4} \sum_{k=1}^{T} \|y_k - y_k^*\| + \frac{3}{4} \sum_{t=1}^{T-1} t^{-7/4} \sum_{k=1}^{t} \|y_k - y_k^*\|$$

$$\leq \frac{1}{\mu\eta} T^{-3/4}\left(\left(\sqrt{2}DL + \sqrt{D\gamma}\right)\sqrt{T} + \sqrt{2D\sigma_l}T^{3/4}\right) + \frac{3}{4\mu\eta} \sum_{t=1}^{T-1} t^{-7/4}\left(\left(\sqrt{2}DL + \sqrt{D\gamma}\right)\sqrt{t} + \sqrt{2D\sigma_l}t^{3/4}\right)$$

$$\leq \frac{1}{\mu\eta}\left(\left(\sqrt{2}DL + \sqrt{D\gamma}\right)T^{-1/4} + \sqrt{2D\sigma_l}\right) + \frac{3}{4\mu\eta}\left(4\left(\sqrt{2}DL + \sqrt{D\gamma}\right) + \sqrt{2D\sigma_l}(1 + \log T)\right)$$

$$\leq \frac{2}{\mu\eta}\left(2\left(\sqrt{2}DL + \sqrt{D\gamma}\right) + \sqrt{2D\sigma_l}(1 + \log T)\right),$$

where the second inequality uses $t^{-3/4} - (t+1)^{-3/4} \leq 3t^{-7/4}/4$, and the third inequality is due to Lemma 5.7. Therefore,

$$\sum_{t=1}^{T} \eta_{x,t} \left\| \sum_{k=2}^{t} \beta_{(k+1):t}\alpha_k \epsilon_k^{\mathsf{B}} \right\| \leq \frac{2\eta LD}{3}((t_0 - 1)^{3/2} - 1) + 2(t_0 - 2)LD\eta e\sqrt{\kappa_\sigma}\left(1 + 2\sqrt{\bar{\sigma}_u/\alpha}\right)$$

$$+ \frac{2L\alpha}{\mu\underline{\sigma}_u}\left(1 + 2e\sqrt{\kappa_\sigma}\left(1 + 2\sqrt{\bar{\sigma}_u/\alpha}\right)\right)\left(2\left(\sqrt{2}DL + \sqrt{D\gamma}\right) + \sqrt{2D\sigma_l}(1 + \log T)\right). \quad (55)$$

Combining Equations (54) and (55), we obtain

$$\sum_{t=1}^{T} \eta_{x,t} \left\| \sum_{k=2}^{t} \beta_{(k+1):t}\alpha_k \epsilon_k^{\mathsf{B}} \right\| \leq \left( LD\sqrt{\frac{\eta(\alpha + \gamma)}{\mu}}(1 + \log T) \right) \mathbb{I}(\bar{\sigma}_u = 0)$$

$$+ \left( \frac{2\eta LD}{3}((t_0 - 1)^{3/2} - 1) + 2(t_0 - 2)LD\eta e\sqrt{\kappa_\sigma}\left(1 + 2\sqrt{\bar{\sigma}_u/\alpha}\right) \right.$$

$$+ \frac{2L\alpha}{\mu\underline{\sigma}_u}\left(1 + 2e\sqrt{\kappa_\sigma}\left(1 + 2\sqrt{\bar{\sigma}_u/\alpha}\right)\right)\left(2\left(\sqrt{2}DL + \sqrt{D\gamma}\right) + \sqrt{2D\sigma_l}(1 + \log T)\right) \Bigg) \mathbb{I}(\underline{\sigma}_u > 0).$$

$$\square$$

**Lemma G.4.** *For all $t \geq 1$, let $\alpha_t$, $\beta_t$, $\eta_{x,t}$, and $\epsilon_t^{\mathsf{N}}$ be defined as in Equations (3) and (4) and Lemma E.4 for bilevel optimization (see Section 4.3):*

$$\alpha_t = \frac{\alpha}{\sqrt{\alpha^2 + \sum_{k=1}^{t} \|g_{x,k} - \tilde{g}_{x,k}\|^2}}, \quad \beta_t = 1 - \alpha_t, \quad \epsilon_t^{\mathsf{N}} = \mathbb{E}_{t-1}[g_{x,t}] - \bar{\nabla}f(x_t, y_t),$$

$$\alpha_t' = \frac{\alpha}{\sqrt{\alpha^2 + \sum_{k=1}^{t} \|g_{x,k} - \tilde{g}_{x,k}\|^2 + \|g_{y,k}\|^2}}, \quad \text{and} \quad \eta_{x,t} = \frac{\eta\sqrt{\alpha_t'}}{\sqrt{t}}.$$

*Then we have*

$$\sum_{t=1}^{T} \eta_{x,t} \left\| \sum_{k=2}^{t} \beta_{(k+1):t}\alpha_k \epsilon_k^{\mathsf{N}} \right\| \leq \frac{2\eta T^{3/2} l_{g,1} l_{f,0}}{3\mu_g}\left(1 - \frac{\mu_g}{l_{g,1}}\right)^N.$$

*Proof of Lemma G.4.* By triangle inequality and Lemma E.2,

$$\sum_{t=1}^{T} \eta_{x,t} \left\| \sum_{k=2}^{t} \beta_{(k+1):t} \alpha_k \epsilon_k^{\mathtt{N}} \right\| \leq \sum_{t=1}^{T} \eta_{x,t} \sum_{k=2}^{t} \beta_{(k+1):t} \alpha_k \| \epsilon_k^{\mathtt{N}} \|$$

$$\leq \frac{l_{g,1} l_{f,0}}{\mu_g} \left( 1 - \frac{\mu_g}{l_{g,1}} \right)^{N} \sum_{t=1}^{T} \eta_{x,t} \sum_{k=2}^{t} \beta_{(k+1):t} \alpha_k$$

$$\leq \frac{l_{g,1} l_{f,0}}{\mu_g} \left( 1 - \frac{\mu_g}{l_{g,1}} \right)^{N} \sum_{t=1}^{T} \frac{\eta}{\sqrt{t}} (t-1)$$

$$\leq \frac{2\eta T^{3/2} l_{g,1} l_{f,0}}{3\mu_g} \left( 1 - \frac{\mu_g}{l_{g,1}} \right)^{N},$$

where the third inequality uses $\eta_{x,t} \leq \eta/\sqrt{t}$ and $\alpha_t, \beta_t \leq 1$. $\qquad\square$

## H  Discussion on Existing Algorithms for Minimax Optimization

Among existing algorithms for nonconvex-strongly-concave minimax optimization, TiAda [47] is the only work that attempts to be noise-adaptive. However, their convergence guarantees in the stochastic setting depend only on upper bounds of the stochastic gradient norm and the function value (e.g., Assumption 3.4, 3.5, and Theorem 3.2 in [47]), rather than the actual noise level of stochastic gradients. Consequently, TiAda does not achieve optimal convergence in terms of the dependency on stochastic gradient variance.

## I  Experimental Settings for Synthetic Experiments

For synthetic experiments, we tune hyperparameters for each baseline using a grid search and report their best results. Both the parameter $\alpha$ used in the momentum parameter estimate (3) and the base learning rates $(\eta_x, \eta_y)$ are tuned within the set $\{0.5, 1.0, 1.5, 2.0, 3.0, 4.0, 5.0\}$. We use the following parameter choices for various noise magnitude: for $\sigma = 0$, $\alpha = 2.0$, $\eta_x = 3.0$, $\eta_y = 3.0$ for `Ada-Minimax`, and $\eta_x = 4.0$, $\eta_y = 4.0$ for TiAda; for $\sigma = 20$, $\alpha = 2.0$, $\eta_x = 1.5$, $\eta_y = 1.5$ for `Ada-Minimax`, and $\eta_x = 2.0$, $\eta_y = 2.0$ for TiAda; for $\sigma = 50$, $\alpha = 3.0$, $\eta_x = 2.0$, $\eta_y = 2.0$ for `Ada-Minimax`, and $\eta_x = 2.0$, $\eta_y = 2.0$ for TiAda; for $\sigma = 100$, $\alpha = 5.0$, $\eta_x = 3.0$, $\eta_y = 3.0$ for `Ada-Minimax`, and $\eta_x = 2.5$, $\eta_y = 2.5$ for TiAda. Other hyperparameters in TiAda are set to the default choices as suggested in [47].

## J  Experimental Settings for Deep AUC Maximization

For a fair comparison, we tune hyperparameters for each baseline using a grid search and report their best results. The base learning rates $(\eta_x, \eta_y)$ are tuned within the range of $[0.001, 0.1]$. Specifically, we select $(\eta_x, \eta_y) = (0.1, 0.05)$ for SGDA, $(0.01, 0.1)$ for PDSM, $(0.1, 0.05)$ for TiAda, and $(0.01, 0.01)$ for Ada-Minimax. The exponential hyperparameters $(\alpha, \beta)$ for TiAda follow the original settings in their paper, i.e., $(0.6, 0.4)$. For Ada-Minimax, the parameters $(\alpha, \gamma)$ are tuned within $\alpha \in \{0.1, 0.5, 1.0, 2.0\}$ and $\gamma \in \{0.01, 0.1, 1.0, 2.0\}$, resulting in the optimal choice $(\alpha, \gamma) = (0.5, 0.1)$.

## K  Experiments for Hyperparameter Optimization

In this section, we consider hyperparameter optimization on the TREC text classification dataset [49], provided under the Creative Commons Attribution 4.0 License. We formulate the hyperparameter optimization problem as follows:

$$\min_{\lambda} \frac{1}{|\mathcal{D}_{\text{val}}|} \sum_{\xi \in \mathcal{D}_{\text{val}}} \mathcal{L}(\boldsymbol{w}^*(\lambda); \xi), \quad \text{s.t.} \quad \boldsymbol{w}^*(\lambda) = \arg\min_{\boldsymbol{w}} \frac{1}{|\mathcal{D}_{\text{tr}}|} \sum_{\zeta \in \mathcal{D}_{\text{tr}}} \left( \mathcal{L}(\boldsymbol{w}; \zeta) + \frac{\lambda}{2} \|\boldsymbol{w}\|^2 \right),$$

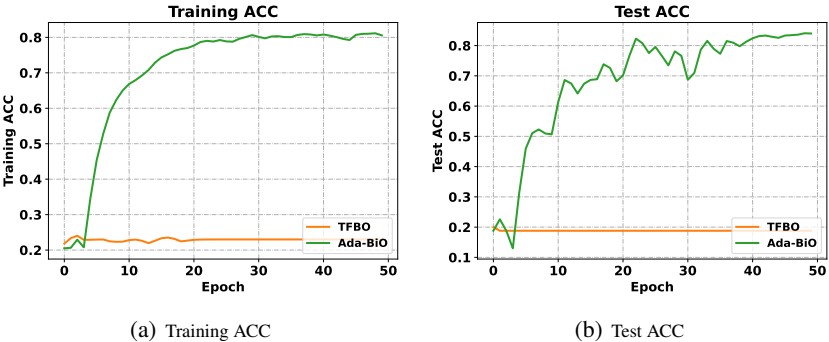

(a) Training ACC  (b) Test ACC

Figure 4: Comparison of BERT model on hyperparameter optimization.

where $\mathcal{L}(\boldsymbol{w}; \xi)$ denotes the loss function, $\boldsymbol{w}$ represents model parameters, and $\lambda$ is the regularization hyperparameter. Here, $\mathcal{D}_{\mathrm{tr}}$ and $\mathcal{D}_{\mathrm{val}}$ denote the training and validation datasets, respectively. In our experiments, we employ a BERT model with 4 self-attention layers, each comprising 4 attention heads, followed by a fully-connected layer with an output dimension of 6, corresponding to the six classification categories. The model is trained from scratch for 50 epochs. We compare our algorithm's training and test performance against the tuning-free bilevel optimization (TFBO) method proposed by [73]. For TFBO, we conduct a grid search to select optimal initial values for the upper-level learning rate $\alpha_0$, lower-level learning rate $\beta_0$, and linear system learning rate $\varphi$ within the range $[1.0 \times 10^{-5}, 10.0]$, and set them to $\{0.01, 0.1, 0.1\}$. For Ada-BiO, we similarly perform hyperparameter tuning over the parameters $(\eta_x, \eta_y, \alpha, \gamma)$ within the range $[1.0 \times 10^{-5}, 1.0]$, selecting the optimal values $(1.0 \times 10^{-5}, 0.5, 1.0, 0.1)$ for evaluation.

The training and test accuracy curves are illustrated in Figure 4. TFBO fails to converge because it is originally designed for deterministic scenarios, rendering it ineffective for stochastic settings. In contrast, Ada-BiO demonstrates rapid convergence in terms of training accuracy and consistently achieves superior test performance.

## L   Experiments for Verifying Assumptions

We empirically verify Assumption 3.2(ii), which states that the noise of the stochastic gradient satisfies $\underline{\sigma}_x \leq \|\nabla_x F(x, y; \xi) - \nabla_x f(x, y)\| \leq \bar{\sigma}_x$ with $\underline{\sigma}_x \geq 0$. Specifically, following the experimental setup for deep AUC maximization described in Section 6.2, we compute the exact gradient $\nabla_x f(x, y)$ after each training epoch by averaging the gradients over the entire validation dataset with fixed model parameters and hyperparameters. Similarly, we compute the stochastic gradient $\nabla_x F(x, y; \xi)$, but using a randomly sampled mini-batch from the validation set. We observe that the empirical maximal and minimal noise levels are $\bar{\sigma}_x = 210.71$ and $\underline{\sigma}_x = 0.21$, respectively, thus confirming that practical stochastic gradient noise is indeed bounded from both sides.

