# OpenReview forum: "Adaptive Algorithms with Sharp Convergence Rates for Stochastic Hierarchical Optimization"
_NeurIPS.cc/2025/Conference — NeurIPS 2025 poster_

### Official Review · Reviewer_kf2Z · 2025-06-23

**Clarity:** 3
**Significance:** 3
**Originality:** 2
**Rating:** 5
**Confidence:** 3

**Summary:**

This paper studies nonconvex–strongly-concave minimax and bilevel optimization problems and proposes adaptive algorithms that adjust to the gradient noise level, whether it is in a high- or low-noise regime.

**Questions:**

1. The convergence results are established for cases where the noise level (i.e., variance of the stochastic gradient) may vary over iterations, covering both low- and high-noise regimes. Does the first experiment (Figure 1) reflect such a setting? If not, it would be informative to include a comparison between TiAda and the proposed method under these conditions.

2. I was wondering whether this approach might be extendable to the nonconvex–concave settings. Since momentum-based methods are sometimes effective in maximizing concave objective, it seems possible that such an extension could work.

3. Since the theoretical results (e.g., Thm 5.6) hold with high probability, it may be beneficial to include confidence intervals in the experimental plots to better illustrate the results.

4. In [1], the convergence of the expected gradient is established in expectation, without relying on probability bounds. In contrast, the results in the present paper (e.g., Theorem 4.1) show that the (deterministic) sum of gradients converges with high probability. Can the current analysis similarly be extended to guarantee convergence in expectation, without relying on high-probability bounds? Alternatively, could the authors clarify which result provides a stronger guarantee?

[1] Xiang Li, YANG Junchi, and Niao He. Tiada: A time-scale adaptive algorithm for nonconvex minimax
optimization. In The Eleventh International Conference on Learning Representations.

**Ethical Concerns:**

["NO or VERY MINOR ethics concerns only"]

**Final Justification:**

1. My main concern was the significance of the theoretical results presented in the paper, and the authors have addressed this clearly in A.4.
2. In addition, they provided experimental results on a time-varying example, which I had hoped to see included in the paper.
3. Authors also clarified the applicability of their results to relaxed settings, which further strengthened the contribution.
Overall, I believe the clarity and completeness of the paper will be significantly improved in the revised version, and I have raised my score accordingly.

**Limitations:**

yes

**Quality:**

3

**Strengths And Weaknesses:**

Strengths:
1. The theoretical results look solid and this paper is well-organized.
2. The paper proposes an adaptive algorithm that achieves sharp convergence rates in both nonconvex–strongly-concave minimax and bilevel optimization, without requiring prior knowledge of problem-specific parameters such as smoothness constants or noise variance.
3. The method automatically adapts to different noise regimes, and the convergence results smoothly interpolate between the optimal rates for both high- and low-noise settings.

Weaknesses: The theoretical results look solid but not strong since they only provide high probability convergence rather than convergence in expectation or almost sure convergence.

---

> ### Author Rebuttal · Authors · 2025-07-29
>
> **Q1. The convergence results are established for cases where the noise level (i.e., variance of the stochastic gradient) may vary over iterations, covering both low- and high-noise regimes. Does the first experiment (Figure 1) reflect such a setting? If not, it would be informative to include a comparison between TiAda and the proposed method under these conditions.**
>
> **A1.** Thank you for your insightful question. We conducted experiments using time-varying stochastic gradient noise sampled from $\mathcal{N}(0, \sigma_t^2)$ where $\sigma_t = \sigma(1+\sin t)$ at the $t$-th iteration. We run the experiments using 5 different random seeds and the results are presented in Table 1. Both the momentum parameter $\alpha$, the base learning rates $(\eta_x, \eta_y)$, and $\gamma$ are tuned over the set $\\{0.1, 0.5, 1.0, 2.0, 3.0, 4.0, 5.0\\}$. We use the following parameter settings for different noise magnitudes:
> - For $\sigma = 20$: Ada-Minimax: $\alpha = 5.0$, $\eta_x = 1.0$, $\eta_y = 1.0$; TiAda: $\eta_x = 4.0$, $\eta_y = 4.0$. Run for $T=1000$ iterations.
> - For $\sigma = 30$: Ada-Minimax: $\alpha = 5.0$, $\eta_x = 1.0$, $\eta_y = 1.0$; TiAda: $\eta_x = 2.0$, $\eta_y = 2.0$. Run for $T=2500$ iterations.
>
> **Table 1: Gradient norm $\\|\nabla\Phi(x_t)\\|$ evaluated at the last step.**
>
> | Method       | $\sigma = 20$                  | $\sigma = 30$                  |
> |--------------|--------------------------------|---------------------------------|
> | TiAda        | $0.4451_{\pm 0.4218}$          | $0.1796_{\pm 0.1429}$           |
> | Ada-Minimax  | $0.0418_{\pm 0.0390}$          | $0.1153_{\pm 0.0873}$           |
>
>
> &#8203;
>
> **Q2. I was wondering whether this approach might be extendable to the nonconvex–concave settings. Since momentum-based methods are sometimes effective in maximizing concave objective, it seems possible that such an extension could work.**
>
> **A2.** If we further assume that "$y \in \mathcal{Y} \subseteq \mathbb{R}^{d_y}$ and $\mathcal{Y}$ is a convex and bounded set with diameter $D \geq 0$" as stated in [1, Assumption 4.2], and modify line 5 of our Algorithm 1 to $y_{t+1} = \mathcal{P}\_{\mathcal{Y}}(y_t + \eta_{y,t} g_{y,t})$, and also replace our current convergence measure with the gradient of the Moreau envelope (see Definition 3.5 and Lemma 3.6 in [1]), then it becomes possible to extend our algorithmic framework to the nonconvex-concave setting.
>
> We outline this as a high-level technical idea and leave the detailed theoretical analysis for future work. In particular, Theorem D.2 (the full version of Theorem 4.10) in [1] shows that the choices of $\eta_x$ and $\eta_y$ depend on the noise level $\sigma$. The requirement for prior knowledge of $\sigma$ and other problem-dependent parameters could potentially be removed by employing our adaptive parameter choices defined in Eqs. (3) and (4), possibly with adjusted powers in the denominators to asymptotically match the orders of $\eta_x$ and $\eta_y$ specified in Theorem D.2 of [1].
>
> &#8203;
>
> **Q3. Since the theoretical results (e.g., Thm 5.6) hold with high probability, it may be beneficial to include confidence intervals in the experimental plots to better illustrate the results.**
>
> **A3.** We have conducted the experiments using 5 different random seeds. The hyperparameter settings and experimental details are included in Appendices I and J. See Tables 2 and 3 for details.
>
> **Table 2: Gradient norm $\\|\nabla\Phi(x_t)\\|$ evaluated at the last step.**
>
> | Method       | $\sigma = 0$               | $\sigma = 20$                | $\sigma = 50$                | $\sigma = 100$               |
> |--------------|-----------------------------|-------------------------------|-------------------------------|-------------------------------|
> | TiAda        | $0.0193_{\pm 0.0000}$      | $0.1657_{\pm 0.1052}$        | $0.5437_{\pm 0.5332}$        | $0.9394_{\pm 0.8541}$        |
> | Ada-Minimax  | $0.0163_{\pm 0.0000}$      | $0.0555_{\pm 0.0319}$        | $0.1198_{\pm 0.1095}$        | $0.1644_{\pm 0.1391}$        |
>
> **Table 3: The final training/test AUC with 5 runs.**
>
> | Method       | Training AUC               | Test AUC                 |
> |--------------|-----------------------------|-------------------------------|
> | SGDA  | $0.7435_{\pm 0.0424}$      | $0.6970_{\pm 0.0251}$     |
> | PDSM  | $0.7765_{\pm 0.0050}$      | $0.7133_{\pm 0.0072}$     |
> | TiAda        | $0.6127_{\pm 0.0138}$      | $0.6152_{\pm 0.0010}$      |
> | Ada-Minimax  | $0.9284_{\pm 0.0252}$      | $0.7331_{\pm 0.0196}$     |
>
> &#8203;
>
> **Q4. In [2], the convergence of the expected gradient is established in expectation, without relying on probability bounds. In contrast, the results in the present paper (e.g., Theorem 4.1) show that the (deterministic) sum of gradients converges with high probability. Can the current analysis similarly be extended to guarantee convergence in expectation, without relying on high-probability bounds? Alternatively, could the authors clarify which result provides a stronger guarantee?**
>
> **A4.** We would like to clarify that our high-probability result is stronger than the in-expectation result in [2]. Notably, [2] assumes almost surely bounded stochastic gradient noise implicitly: their [2, Assumption 3.4] implies $\\|\nabla_zF(x,y;\xi) - \nabla_zf(x,y)\\|\leq \\|\nabla_zF(x,y;\xi)\\| + \\|\nabla_zf(x,y)\\| \leq 2G$, where $G$ is the upper bound on the stochastic gradient norm. This indicates that the noise is almost surely bounded by $2G$, and thus their analysis relies on a light-tailed noise assumption, similar to ours.
>
> In fact, a high-probability bound always implies an in-expectation bound; see Lemma A.3 in [3] for details. By applying Lemma A.3 in [3], we can derive an in-expectation bound of $\frac{4}{\sqrt{\log(2/\delta)}}\cdot\text{RHS} = \widetilde{O}(1/\sqrt{T} + \sqrt{\bar{\sigma}_x}/T^{1/4})$, where RHS denotes the right-hand side of the bound in our Theorem 4.1. This expectation bound is comparable to that of [2] up to a $\text{polylog}(1/\delta)$ factor in the worst case. However, our bound is noise-adaptive, whereas their bound in [2, Theorem 3.2] depends on the stochastic gradient norm bound $G$.
>
> Conversely, while an in-expectation bound in [2] can be converted into a high-probability bound using Markov's inequality, this approach results in a looser dependency on the failure probability $\delta$, specifically yielding a bound that scales with $O(1/\delta)$. In contrast, our analysis yields a high-probability bound that depends only on $O(\log(1/\delta))$, and is therefore much tighter.
>
> &#8203;
>
> [1] Lin, Tianyi, Chi Jin, and Michael Jordan. "On gradient descent ascent for nonconvex-concave minimax problems." In International conference on machine learning, pp. 6083-6093. PMLR, 2020.
>
> [2] Li, Xiang, Junchi Yang, and Niao He. "TiAda: A Time-scale Adaptive Algorithm for Nonconvex Minimax Optimization." In The Eleventh International Conference on Learning Representations, 2023.
>
> [3] Shalev-Shwartz, Shai, and Shai Ben-David. Understanding machine learning: From theory to algorithms. Cambridge university press, 2014.

---

> > ### Comment · Reviewer_kf2Z · 2025-08-01
> >
> > I thank the authors for their thorough response. I will raise my score to 5. I encourage the authors to include experiments involving time-varying gradient noise in the revised version of the paper, and to highlight more clearly the significance of the result discussed in A.4

---

> > > ### Author Response · Authors · 2025-08-01
> > >
> > > We thank reviewer kf2Z for the insightful review and constructive comments. In the revised paper, we will include experiments involving time-varying gradient noise and provide a clearer discussion of A.4.

---

### Official Review · Reviewer_59Af · 2025-06-28

**Clarity:** 3
**Significance:** 3
**Originality:** 3
**Rating:** 4
**Confidence:** 3

**Summary:**

This paper introduces new adaptive stochastic optimization algorithms for hierarchical problems (specifically minimax and bilevel optimization) that automatically adjust to the noise in the data. The authors identify that existing methods for stochastic convex optimization and hierarchical optimization do not achieve optimal convergence across different noise regimes unless the noise level is known in advance. In response, the paper proposes two algorithms (dubbed Ada-Minimax and Ada-BiO) that leverage a momentum normalization technique combined with adaptive step-size tuning. These methods guarantee sharp convergence rates of $\tilde{O}(1/\sqrt{T} + \sqrt{\bar{\sigma}}/T^{1/4})$ for the gradient norm, matching known lower bounds and thus achieving optimal statistical performance in both low-noise and high-noise settings. Notably, the algorithms require no prior knowledge of the noise distribution or magnitude, handling both well-specified and misspecified noise models seamlessly.

**Questions:**

NA.

**Ethical Concerns:**

["NO or VERY MINOR ethics concerns only"]

**Final Justification:**

I received author's response and decide to maintain my current score.

**Limitations:**

yes.

**Paper Formatting Concerns:**

some equations are out of the line space. An arrangement of the equation format would be good.

**Quality:**

3

**Strengths And Weaknesses:**

Strengths:

1. The paper provides a novel theoretical result by achieving adaptive convergence rates in hierarchical stochastic optimization. It is the first work to guarantee optimal rates across different noise regimes without tuning to noise level, addressing a clear gap in the literature.

2. Each major claim is supported either by a theorem or an experiment. The authors prove their convergence rate results under clear assumptions and also include a comparison to existing methods to highlight where those fail to adapt. The presence of both theory and experiments strengthens confidence that the contributions are sound and not merely of theoretical interest.

3. Despite the theoretical nature, the paper demonstrates strong practical performance. Empirical results on a range of tasks show the new algorithms outperform traditional baselines (both adaptive and non-adaptive) in convergence speed and stability.

Weaknesses:

1. Due to the technical depth of the contribution, some parts of the paper are dense. The theoretical sections (e.g. proofs and algorithm analysis) involve intricate analysis of two-level optimization with time-varying parameters. This could make it challenging for non-experts to follow all details.

2. The adaptive guarantees are proven under certain assumptions (e.g. the lower-level problem is strongly convex or concave, smoothness of objectives, and bounded noise variance). These conditions limit the scope of the results. I understand that these assumptions are standard in this field. However, it somehow makes the theoretical analysis deviate from the realistic scenarios.

---

> ### Author Rebuttal · Authors · 2025-07-29
>
> **Q1. Due to the technical depth of the contribution, some parts of the paper are dense. The theoretical sections (e.g. proofs and algorithm analysis) involve intricate analysis of two-level optimization with time-varying parameters. This could make it challenging for non-experts to follow all details.**
>
> **A1.** Thank you for your suggestions. In Algorithms 1 and 2, we update $x_t$ using normalized SGD with momentum and $y_t$ using AdaGrad-Norm, with momentum parameters $\alpha_t$ and learning rates $\eta_{x,t}$ and $\eta_{y,t}$ defined in Eqs. (3) and (4). These parameter choices are motivated by the goal of estimating variance through differences between independently sampled gradients, thus enabling sharp and adaptive convergence rates without requiring prior knowledge of the noise level (or other problem-dependent parameters).
>
> For the theoretical analysis of Theorems 4.1 and 4.2 (corresponding to Algorithms 1 and 2), we begin by analyzing the adaptive normalized SGD with momentum (Algorithm 3) in the nonconvex stochastic single-level optimization setting, as described in Section 5.1. In Theorem 5.6, we establish a sharp noise-adaptive convergence rate of $\widetilde{O}(1/\sqrt{T} + \sqrt{\bar{\sigma}}/T^{1/4})$. A key component of this result involves applying concentration inequalities and linear programming techniques to derive tight bounds for the momentum deviation $\\|m_t - \nabla F(x_t)\\|$ (Lemma 5.4) and the momentum parameters (Lemma 5.5). The upper-level analyses for both minimax and bilevel problems then follow as straightforward extensions of this single-level framework.
>
> Next, we provide a unified analysis of the lower-level estimation error applicable to both minimax and bilevel optimization in Lemma 5.7 (Section 5.2). This result is critical for controlling the bias in (hyper)gradient estimation. Specifically, it extends the AdaGrad-Norm analysis developed for convex settings in [1] to account for the shift induced by $x_t$, while incorporating our novel adaptive parameter choices from Eqs. (3) and (4). Note that setting $g = -f$ in the bilevel problem reduces it to the minimax formulation.
>
> Finally, by combining the upper-level analysis framework from Section 5.1 with the lower-level estimation error bounds developed in Section 5.2 (i.e., bounds on the gradient and hypergradient estimation bias), we derive Theorems 4.1 and 4.2.
>
> In the revised version, we will reorganize the paper and include more intuition and high-level explanations of the theoretical proofs and algorithm analysis in Sections 4 and 5 to improve readability and make the paper easier to follow.
>
> **Q2. The adaptive guarantees are proven under certain assumptions (e.g. the lower-level problem is strongly convex or concave, smoothness of objectives, and bounded noise variance). These conditions limit the scope of the results. I understand that these assumptions are standard in this field. However, it somehow makes the theoretical analysis deviate from the realistic scenarios.**
>
> **A2.** These assumptions are indeed standard in the fields of minimax and bilevel optimization; see [2, 3, 4, 5, 6] and the assumptions therein. In fact, many practical applications, such as meta-learning, naturally satisfy these assumptions, including strong convexity of the lower-level problem as the lower-level variable often represents the last layer of neural networks [3, 7].
>
> &#8203;
>
> [1] Attia, Amit, and Tomer Koren. "SGD with AdaGrad stepsizes: Full adaptivity with high probability to unknown parameters, unbounded gradients and affine variance." In International Conference on Machine Learning, pp. 1147-1171. PMLR, 2023.
>
> [2] Li, Xiang, Junchi Yang, and Niao He. "TiAda: A Time-scale Adaptive Algorithm for Nonconvex Minimax Optimization." In The Eleventh International Conference on Learning Representations, 2023.
>
> [3] Ji, Kaiyi, Junjie Yang, and Yingbin Liang. "Bilevel Optimization: Convergence Analysis and Enhanced Design." In International Conference on Machine Learning, pp. 4882-4892. PMLR, 2021.
>
> [4] Ghadimi, Saeed, and Mengdi Wang. "Approximation methods for bilevel programming." arXiv preprint arXiv:1802.02246 (2018).
>
> [5] Chen, Tianyi, Yuejiao Sun, and Wotao Yin. "Closing the gap: Tighter analysis of alternating stochastic gradient methods for bilevel problems." Advances in Neural Information Processing Systems 34 (2021): 25294-25307.
>
> [6] Hong, Mingyi, Hoi-To Wai, Zhaoran Wang, and Zhuoran Yang. "A two-timescale stochastic algorithm framework for bilevel optimization: Complexity analysis and application to actor-critic." SIAM Journal on Optimization 33, no. 1 (2023): 147-180.
>
> [7] Raghu, Aniruddh, Maithra Raghu, Samy Bengio, and Oriol Vinyals. "Rapid Learning or Feature Reuse? Towards Understanding the Effectiveness of MAML." In The Eighth International Conference on Learning Representations, 2020.

---

### Official Review · Reviewer_TL2T · 2025-07-01

**Clarity:** 2
**Significance:** 3
**Originality:** 3
**Rating:** 4
**Confidence:** 4

**Summary:**

This paper presents novel adaptive algorithms for stochastic hierarchical optimization, focusing on nonconvex-strongly-concave minimax and nonconvex-strongly-convex bilevel problems. The proposed methods achieve state-of-the-art convergence rates without requiring prior knowledge of gradient noise levels, overcoming a key limitation in existing approaches. The technical innovation combines momentum normalization for upper-level variables with a carefully designed adaptive stepsize scheme for both levels, providing the first theoretically guaranteed noise-adaptive solution for these challenging optimization problems. Experimental evaluations on both synthetic benchmarks and real-world deep learning tasks demonstrate the superior performance of our approach compared to existing methods.

**Questions:**

Question1: The challenges section should clarify the algorithmic goals (e.g., noise-adaptive convergence? hyperparameter robustness?) and explicitly link them to the listed challenges. For example:

1.1 Why is "update balance" critical for the goal? Does imbalance cause divergence or slow convergence?

1.2 How does "randomness dependency" hinder theoretical guarantees (e.g., broken Martingale conditions)?

1.3 Please contrast with single-level optimization to highlight the unique hierarchical difficulties.

Question2: The authors state that the minimax problem (1) is a special case of the bilevel problem (2) when g=−f. However, the paper subsequently develops separate algorithms for each class. Could the authors clarify:

2.1 Why a unified bilevel algorithm cannot naturally handle the minimax case by setting g=−f?

2.2 Are there fundamental differences between the two problem classes that necessitate distinct algorithmic treatments (e.g., convergence guarantees, hyperparameter tuning, or practical performance)?

Question3: Regarding Theorems 1 and 2, I notice that both convergence results contain the parameter \eta, while the corresponding Algorithms 1 and 2 actually employ two distinct step sizes (\eta_{x} for upper-level variables and \eta_{y} for lower-level variables). Could the authors please clarify what \eta specifically represents in each theorem statement. A clear explanation of this notation choice would significantly improve the theoretical rigor and readability of the paper.

**Ethical Concerns:**

["NO or VERY MINOR ethics concerns only"]

**Final Justification:**

I thank the authors for their thorough response, I wil maintain my score.

**Limitations:**

yes

**Quality:**

3

**Strengths And Weaknesses:**

Strengths:
The paper makes significant algorithmic contributions by developing the first theoretically guaranteed noise-adaptive methods for stochastic hierarchical optimization problems. The novel combination of momentum normalization for upper-level variables with adaptive stepsizes for both levels represents an important technical advancement and could inspire future research in the field.

Weaknesses:
The paper contains several unclear key formulations, for example, the 'Main Challenges' section fails to clearly articulate the fundamental objectives of the algorithm design (e.g., achieving noise-adaptive convergence, ensuring hyperparameter robustness). This omission makes it difficult to evaluate whether the subsequently listed challenges are indeed the most critical barriers to these objectives.

---

> ### Author Rebuttal · Authors · 2025-07-29
>
> **Q1. The challenges section should clarify the algorithmic goals (e.g., noise-adaptive convergence? hyperparameter robustness?) and explicitly link them to the listed challenges. For example: 1) Why is "update balance" critical for the goal? Does imbalance cause divergence or slow convergence? 2) How does "randomness dependency" hinder theoretical guarantees (e.g., broken Martingale conditions)? 3) Please contrast with single-level optimization to highlight the unique hierarchical difficulties.**
>
> **A1.** The update balance is critical to establish guarantees for the lower-level estimation error, see Lemma 5.7 and its proof in Appendix C for details. Technically, this balance is captured by the term $\eta_{x,t}^2 / \eta_{y,t}$, and we aim to ensure that $\sum_{k=1}^{t} \eta_{x,k}^2 / \eta_{y,k} = \widetilde{O}(1)$ to provide tight upper bounds for the lower-level estimation error. This condition is essential for guaranteeing convergence in both minimax and bilevel optimization problems. If $\eta_{x,t}$ and $\eta_{y,t}$ are not properly chosen, the resulting imbalance may lead to slow convergence or even divergence in these hierarchical settings.
>
> The complex randomness dependencies arising from both the upper- and lower-level variables do hinder the theoretical guarantees in [1], unless strong assumptions such as bounded stochastic gradients and bounded function values (see Assumptions 3.4 and 3.5 in [1]) are imposed. These assumptions are essential for their proofs of [1, Theorem 3.2], as illustrated in [1, Lemmas C.3–C.5] in [1, Appendix C.2]. In contrast, our newly proposed method updates $x_t$ using normalized SGD with momentum and $y_t$ using AdaGrad-Norm, rather than applying AdaGrad-Norm to both updates, to avoid such strong assumptions. Moreover, we develop a novel high-probability analysis to address the challenges posed by statistical dependencies among the momentum parameters $\alpha_t$, $\beta_t$, and the stochastic gradient noise $\epsilon_t$, as detailed in our Lemmas 5.4 and 5.5 (see proofs in Appendix B.1).
>
> The main challenges in establishing adaptive guarantees for hierarchical optimization lie in two key aspects. First, designing adaptive algorithms for hierarchical problems requires carefully balancing the updates of the upper- and lower-level variables, which is particularly challenging without prior knowledge of the stochastic gradient noise magnitude. Second, the statistical dependencies introduced by the upper- and lower-level updates necessitate the development of a new high-probability analysis to achieve sharp and adaptive convergence rates in hierarchical settings.
>
> **Q2. The authors state that the minimax problem (1) is a special case of the bilevel problem (2) when $g=-f$. However, the paper subsequently develops separate algorithms for each class. Could the authors clarify: 1) Why a unified bilevel algorithm cannot naturally handle the minimax case by setting $g=-f$? 2) Are there fundamental differences between the two problem classes that necessitate distinct algorithmic treatments (e.g., convergence guarantees, hyperparameter tuning, or practical performance)?**
>
> **A2.** The unified algorithmic framework presented in Algorithm 2 can, in fact, handle both minimax problems (by setting $g = -f$) and bilevel problems. However, since minimax problems jointly optimize the same objective function with respect to both $x$ and $y$, we can design a simpler method as presented in Algorithm 1. Specifically, Algorithm 1 (for minimax optimization problem) directly uses the stochastic gradient $g_{x,t} = \nabla_x F(x_t, y_t; \xi_t)$ to update the momentum $m_t$ and the upper-level variable $x_t$, rather than relying on the hypergradient estimator $g_{x,t} = \bar{\nabla} f(x_t, y_t; \bar{\xi}_t)$ used in Algorithm 2 (for bilevel optimization problem).
>
> As a result, the proposed Ada-Minimax method in Algorithm 1 is fully parameter-free (see the remark below Theorem 4.1) and does not require prior knowledge of $\mu$, $L$, or $T$, which would otherwise be needed for the construction of the Neumann series in Algorithm 2 for bilevel problems.
>
> There are no fundamental differences between the two problem classes that necessitate distinct algorithmic treatments. In fact, we provide a unified analytical framework in Section 5 to establish convergence guarantees for both Algorithms 1 and 2 (see Theorems 4.1 and 4.2). The only minor distinction lies in the handling of hypergradients (rather than stochastic gradients) and the additional bias terms introduced by the Neumann series construction in Algorithm 2. Moreover, the hyperparameters that require tuning are essentially the same for both algorithms, with the exception of the parameter $N$ in the Neumann series used in Algorithm 2. In practice, we set $N = 3$ by default, which has been shown to work well in previous studies [2, 3].
>
> **Q3. Regarding Theorems 1 and 2, I notice that both convergence results contain the parameter $\eta$, while the corresponding Algorithms 1 and 2 actually employ two distinct step sizes ($\eta_x$ for upper-level variables and $\eta_y$ for lower-level variables). Could the authors please clarify what $\eta$ specifically represents in each theorem statement. A clear explanation of this notation choice would significantly improve the theoretical rigor and readability of the paper.**
>
> **A3.** Thank you for catching this issue, and we apologize for the ambiguity. In Theorems 4.1 and 4.2 (corresponding to Algorithms 1 and 2), we actually set the free parameters as $\eta_x=\eta_y=\eta$ for simplicity. It is worth noting that this condition is not necessary for establishing convergence, it only affects the universal constants in the rate.
>
> &#8203;
>
> [1] Li, Xiang, Junchi Yang, and Niao He. "TiAda: A Time-scale Adaptive Algorithm for Nonconvex Minimax Optimization." In The Eleventh International Conference on Learning Representations, 2023.
>
> [2] Ji, Kaiyi, Junjie Yang, and Yingbin Liang. "Bilevel Optimization: Convergence Analysis and Enhanced Design." In International Conference on Machine Learning, pp. 4882-4892. PMLR, 2021.
>
> [3] Hao, Jie, Xiaochuan Gong, and Mingrui Liu. "Bilevel Optimization under Unbounded Smoothness: A New Algorithm and Convergence Analysis." In The Twelfth International Conference on Learning Representations, 2024.

---

> > ### Comment · Reviewer_TL2T · 2025-08-05
> >
> > I thank the authors for their thorough response, I wil maintain my score.

---

> > > ### Author Response · Authors · 2025-08-05
> > >
> > > Thank you for your insightful review and constructive comments.

---

### Official Review · Reviewer_pyyb · 2025-07-03

**Clarity:** 3
**Significance:** 2
**Originality:** 3
**Rating:** 4
**Confidence:** 3

**Summary:**

This paper proposes novel adaptive algorithms for nonconvex-strongly-concave min-max and nonconvex-strongly-convex bilevel stochastic optimization problems. The algorithms achieve sharp convergence rates without requiring prior knowledge of the noise level. The proposed method combines momentum normalization with an adaptive stepsize scheme and shows strong empirical performance on synthetic and deep learning tasks.

**Questions:**

- I could not find the convergence rates for existing methods in this manuscript. Therefore, I could not compare the convergence rate of the proposed methods. Please provide the explicit form of convergence rates for existing methods.

- Are the experimental results in Figures 1 and 2 obtained with multiple trials? If not, please conduct the experiments using multiple trials.

- In Section 6.1, the authors insist that Ada-Mimimax consistently outperformed TiAda. However, I could not confirm it from Figure 1. Is the performance gap significant?

- In Algorithms 1 and 2, the update y_t does not use momentum while the update of x_t does. Could you provide the reason?

**Ethical Concerns:**

["NO or VERY MINOR ethics concerns only"]

**Final Justification:**

I reviewed the response from the authors and decide to maintain my current score.

**Limitations:**

yes

**Quality:**

3

**Strengths And Weaknesses:**

Strength:
This paper proposes an adaptive stepsize scheme for bilevel stochastic optimization problems and provides the theoretical analysis deriving the convergence rate. The originality and quality of these points are sufficient.

Weakness:
The experimental evaluation of the proposed method is not significant.

---

> ### Author Rebuttal · Authors · 2025-07-29
>
> **Q1. I could not find the convergence rates for existing methods in this manuscript. Therefore, I could not compare the convergence rate of the proposed methods. Please provide the explicit form of convergence rates for existing methods.**
>
> **A1.** Thank you for your suggestion. We have included comparison tables for adaptive algorithms in minimax and bilevel optimization in Tables 1 and 2, respectively, where the assumptions and convergence rates of existing methods and our proposed methods are explicitly presented.
>
> **Table 1: Comparison of Adaptive Algorithms for Minimax Optimization.**
>
> | Method        | Setting                                      | Assumptions                                         | High Probability | Complexity                                                                 |
> |---------------|-----------------------------------------------|-----------------------------------------------------|------------------|----------------------------------------------------------------------------|
> | TiAda         | Deterministic [1, Theorem 3.1]  | Assumptions 3.1 to 3.3 in [1]          |                  | $O(1/\sqrt{T})$                                                            |
> | TiAda         | Stochastic [1, Theorem 3.2]     | Assumptions 3.1 to 3.6 in [1]          | ❌               | $O(\text{poly}(G) \cdot (T^{\frac{\alpha-1}{2}} + T^{-\frac{\alpha}{2}} + T^{\frac{\beta-1}{2}} + T^{-\frac{\beta}{2}}))^{1}$ |
> | Ada-Minimax   | Deterministic & Stochastic (Theorem 4.1, this work) | Assumptions 3.1 and 3.2 in this work          | ✅               | $\widetilde{O}(1/\sqrt{T} + \bar{\sigma}_x^{1/4}/T^{3/8} + \sqrt{\bar{\sigma}_x}/T^{1/4})$                      |
>
> 1. $G$ denotes the upper bound on the stochastic gradient norm, and $\alpha, \beta$ satisfy $0 < \beta < \alpha < 1$.
>
>
> **Table 2: Comparison of Adaptive Algorithms for Bilevel Optimization.**
>
> | Method     | Setting                                           | Assumptions                                        | High Probability | Complexity                                                                 |
> |------------|----------------------------------------------------|----------------------------------------------------|------------------|----------------------------------------------------------------------------|
> | S-TFBO     | Deterministic [4, Theorem 2]       | Assumptions 1 to 3 in [4]           |                  | $\widetilde{O}(1/\sqrt{T})$                                                |
> | Ada-Bio    | Deterministic & Stochastic (Theorem 4.2, this work) | Assumptions 3.3 and 3.4 in this work             | ✅               | $\widetilde{O}(1/\sqrt{T} + \sigma\_{g,1}^{1/4}/T^{3/8} + (\sqrt{\bar{\sigma}\_{\phi}}+\sqrt{\sigma\_{g,1}})/T^{1/4})$ |
>
> &#8203;
>
> **Q2. Are the experimental results in Figures 1 and 2 obtained with multiple trials? If not, please conduct the experiments using multiple trials.**
>
> **A2.** We have conducted the experiments using 5 different random seeds. The hyperparameter settings and experimental details are included in Appendices I and J. See Tables 3 and 4 for details.
>
> **Table 3: Gradient norm $\\|\nabla\Phi(x_t)\\|$ evaluated at the last step.**
>
> | Method       | $\sigma = 0$               | $\sigma = 20$                | $\sigma = 50$                | $\sigma = 100$               |
> |--------------|-----------------------------|-------------------------------|-------------------------------|-------------------------------|
> | TiAda        | $0.0193_{\pm 0.0000}$      | $0.1657_{\pm 0.1052}$        | $0.5437_{\pm 0.5332}$        | $0.9394_{\pm 0.8541}$        |
> | Ada-Minimax  | $0.0163_{\pm 0.0000}$      | $0.0555_{\pm 0.0319}$        | $0.1198_{\pm 0.1095}$        | $0.1644_{\pm 0.1391}$        |
>
> **Table 4: The final training/test AUC with 5 runs.**
>
> | Method       | Training AUC               | Test AUC                 |
> |--------------|-----------------------------|-------------------------------|
> | SGDA  | $0.7435_{\pm 0.0424}$      | $0.6970_{\pm 0.0251}$     |
> | PDSM  | $0.7765_{\pm 0.0050}$      | $0.7133_{\pm 0.0072}$     |
> | TiAda        | $0.6127_{\pm 0.0138}$      | $0.6152_{\pm 0.0010}$      |
> | Ada-Minimax  | $0.9284_{\pm 0.0252}$      | $0.7331_{\pm 0.0196}$     |
>
> &#8203;
>
> **Q3. In Section 6.1, the authors insist that Ada-Mimimax consistently outperformed TiAda. However, I could not confirm it from Figure 1. Is the performance gap significant?**
>
> **A3.** In the regime where $\sigma$ is large (e.g., $\sigma = 50, 100$), Ada-Minimax converges faster than TiAda, with a significant performance gap. When $\sigma$ is smaller or there is no noise (e.g., $\sigma = 0, 20$), the gap is smaller or the two algorithms perform similarly.
>
> In the revised version of the paper, we will tone down this statement to: "Ada-Minimax outperforms in the large variance regime and performs comparably to TiAda in the small variance regime."
>
> &#8203;
>
> **Q4. In Algorithms 1 and 2, the update $y_t$ does not use momentum while the update of $x_t$ does. Could you provide the reason?**
>
> **A4.** Since the lower-level problem is strongly concave in minimax optimization and strongly convex in bilevel optimization, using momentum for the update of $y_t$ does not help improve the convergence rate. However, updating $x_t$ without momentum necessitates either imposing additional assumptions [1] or using large batch sizes to achieve sharp convergence rates, as shown in prior works [2,3]. Specifically, [2] and [3] require batch sizes of $M = \Theta(\max(1, \kappa \sigma^2 \epsilon^{-2}))$ (see Theorem 4.5 in [2]) and $B = O(\epsilon^{-1})$ (see Section 4.2 in [3]) to achieve $O(\epsilon^{-4})$ convergence rate for nonconvex-strongly-concave minimax problems and $\widetilde{O}(\epsilon^{-4})$ for nonconvex-strongly-convex bilevel problems, respectively. These requirements are often impractical in real-world applications.
>
> In contrast, incorporating momentum with normalization in the update of $x_t$ not only provably eliminates the need for large batch sizes on nonconvex upper-level objectives in both problem settings, but also simplifies the analysis of the lower-level estimation error. In particular, the shift in the lower-level optimal solution, $\\|y_{t+1}^\*-y_t^\*\\|$, can be bounded by $L\eta_{x,t}/\mu$ due to the normalization step, which is critical in establishing the bound in Lemma 5.7 (see proofs in Appendix C for details).
>
> &#8203;
>
> [1] Li, Xiang, Junchi Yang, and Niao He. "TiAda: A Time-scale Adaptive Algorithm for Nonconvex Minimax Optimization." In The Eleventh International Conference on Learning Representations, 2023.
>
> [2] Lin, Tianyi, Chi Jin, and Michael Jordan. "On gradient descent ascent for nonconvex-concave minimax problems." In International conference on machine learning, pp. 6083-6093. PMLR, 2020.
>
> [3] Ji, Kaiyi, Junjie Yang, and Yingbin Liang. "Bilevel Optimization: Convergence Analysis and Enhanced Design." In International Conference on Machine Learning, pp. 4882-4892. PMLR, 2021.
>
> [4] Yang, Yifan, Hao Ban, Minhui Huang, Shiqian Ma, and Kaiyi Ji. "Tuning-free bilevel optimization: New algorithms and convergence analysis." In The Thirteenth International Conference on Learning Representations, 2025.

---

> > ### Comment · Reviewer_pyyb · 2025-08-04
> >
> > Thank you for the detailed response. I will maintain my score. I also expect that the authors incorporate the contents of their response (especially the responses to Q2 and Q3) into their manuscript or supplementary material.

---

> ### Author Response · Authors · 2025-08-04
>
> Thank you for your insightful review and constructive feedback. We will incorporate our responses to Q1-Q3 into the revised paper.

---

### Decision · Program_Chairs · 2025-09-17

**Decision:**

Accept (poster)

**Comment:**

This paper presents new adaptive algorithms for nonconvex–strongly-concave min-max problems and nonconvex–strongly-convex bilevel stochastic optimization. The proposed methods achieve sharp convergence rates without requiring prior knowledge of the noise level. By integrating momentum normalization with an adaptive step-size scheme, the approach demonstrates strong empirical performance on both synthetic benchmarks and deep learning tasks.

From a technical perspective, the paper combines momentum normalization for the upper-level variables with a carefully designed adaptive step-size scheme for both levels, yielding the first theoretically guaranteed noise-adaptive solution for these challenging optimization problems. I believe the paper merits acceptance. Nonetheless, it could be further strengthened by improving the empirical evaluation and presentation in the final version. In particular, I encourage the authors to address the reviewers’ feedback in the final version of the paper, with particular emphasis on:

1-Improving the presentation of the key challenges in the analysis.

2-Discussing the limitations of the analysis.

3-Including experiments that evaluate performance under varying noise levels.